# Twisted S-duality

**Surya Raghavendran[1] and Philsang Yoo[2]**

**1** Department of Mathematics, Yale University, 219 Prospect St,
Floors 7-9, New Haven, CT 06511

**2** Department of Mathematical Sciences & Research Institute of Mathematics,
Seoul National University, 1 Gwanak-ro, Gwanak-gu,
Seoul 08826, Republic of Korea

## Abstract

We study an $SL(2,\mathbb{Z})$ symmetry of a variant of BCOV theory in three complex dimensions. Using conjectural descriptions of twists of superstrings in terms of topological strings, we argue that this action can be thought of as a version of S-duality that preserves an $SU(3)$-invariant twist of type IIB supergravity. We analyze how $SL(2,\mathbb{Z})$ acts on various deformations of the holomorphic-topological twist of 4-dimensional $\mathcal{N}=4$ supersymmetric gauge theory, which come from residual supertranslations and superconformal symmetries, and are of relevance to geometric Langlands theory and gauge-theoretic constructions of the Yangian.

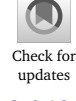

# 1  Introduction

S-duality was originally suggested as a strong-weak duality of 4-dimensional gauge theory [1, 2]: it says that a gauge theory with a given gauge group is equivalent to another gauge theory with its Langlands dual group as the gauge group and with the coupling constant inverted. Soon it was realized that gauge theory with $\mathcal{N} = 4$ supersymmetry is a better context to realize the suggestion [3]. Although the original duality was suggested as a $\mathbb{Z}/4$-symmetry, it is then natural to extend it to SL$(2,\mathbb{Z})$-symmetry as both the coupling constant and the theta angle are acted on nontrivially.

In the 1990s, it was discovered that S-duality can be extended to the context of string theory [4]. In particular, in [5] a conjecture was made that type IIB string theory has SL$(2,\mathbb{Z})$-symmetry. After the conception of M-theory [6], this SL$(2,\mathbb{Z})$-symmetry was found to admit a manifestation as the symmetries of a torus factor of a M-theory background [7,8].

From a mathematical perspective, one rather remarkable application of S-duality of 4-dimensional $\mathcal{N} = 4$ gauge theory is the work of Kapustin and Witten [9] where they argue that a version of the geometric Langlands correspondence is a special case of S-duality. Given that a special application of S-duality leads to such rich mathematics, it is natural to wonder what mathematical marvels the general phenomenon can detect. On the other hand, the stringy origin of S-duality has made such questions somewhat inaccessible to mathematicians. This leads to a natural question of how to mathematically understand its original context, find new examples of S-dual pairs, and accordingly make new mathematical conjectures.

In this paper, we propose an answer to the above question – we propose a mathematical definition of S-duality as an action on a particular twist of type IIB supergravity. We remark that twisted supergravity is entirely contained in the massless sector of the physical theory, and that S-duality of massless IIB supergravity was physically understood as early as the 90s. In addition to mathematical rigor, the main merit of the formalism we develop here is that it allows for

easier calculation of how S-duality acts on further deformations of twists of supersymmetric gauge theories. This allows for recovery of old conjectures and formulation of new ones from a unified perspective: we demonstrate this point in Section 5.

The idea of twisting in the context of supergravity and string theory was formalized in a paper of Costello and Li [10]. This is somewhat similar to the idea of twisting supersymmetric field theory [11] in that one extracts a sector that is a holomorphic-topological field theory. It is now well-known that the idea of twisting supersymmetric field theories has proven highly useful for mathematical applications, with mirror symmetry being a prominent example. On this note, the present paper illustrates that twisted supergravity has similarly rich mathematical implications. Specifically, it provides a natural framework for analyzing an often infinite-dimensional space of deformations of twisted supersymmetric field theories, each of which allows for the study of associated invariants.

## 1.1 Summary

Now let us explain the contents of our paper in a bit more detail. In Section 2, we begin by recalling some features of topological string theory, which provides a convenient framework for discussing closed string field theory, open string field theory, and their relations, all in terms of categorical data. The closed string field theories we will consider all admit descriptions in terms of *BCOV theory* – the closed string field theory associated to the B-model. We recall slight variants of BCOV theory such as *minimal BCOV theory* and a further variant thereof, which on flat space simply enlarges the space of fields by a copy of the constant sheaf. The key result in this section is Theorem 2.20 which shows that after this modification, minimal BCOV theory in three complex dimensions has a natural action of $SL(2,\mathbb{Z})$.

Next, in Section 3 we explain the conjectural descriptions of twists of IIA and IIB supergravity and superstrings. Our discussion is guided by an attempt to unify the perspectives of topological string theory and the work of Costello and Li [10] on twisted supergravity. In particular, there is a twist of type IIB defined on manifolds of the form $M^4 \times X^3$ where $M^4$ is a symplectic 4-manifold and $X^3$ is a Calabi–Yau three-fold. The $SL(2,\mathbb{Z})$ symmetry of our modification of minimal BCOV theory in three complex dimensions induces an $SL(2,\mathbb{Z})$ symmetry of this twist. It is this symmetry that we call *twisted S-duality*.

Next, in Section 4 we provide justification that the symmetry of the twist of type IIB we have identified is in fact a manifestation of S-duality. Physically, S-duality of type IIB can be understood via a relationship between type IIB on a circle and M-theory on a two-torus; starting from the latter theory, one can compactify on one circle fiber and T-dualize along the other to obtain type IIB on a circle. This chain of equivalences is summarized in the following diagram.

$$
\begin{array}{ccc}
\overset{\mathrm{SL}(2,\mathbb{Z})}{\curvearrowright} & & \overset{\mathrm{SL}(2,\mathbb{Z})}{\curvearrowright} \\
\mathrm{M}[S^1_{\mathrm{M}} \times S^1_r \times M^9] \xrightarrow[\simeq]{\mathrm{red}_M} \mathrm{IIA}[S^1_r \times M^9] \xrightarrow[\simeq]{\mathbf{T}} \mathrm{IIB}[S^1_{1/r} \times M^9]
\end{array}
$$

Figure 1: S-duality of type IIB string theory from M-theory.

The $SL(2,\mathbb{Z})$ action on the source is induced by a spacetime symmetry whereas the $SL(2,\mathbb{Z})$ action on the target is the aforementioned S-duality group. We show that the same diagram holds at the level of twists. We construct a map $\Phi_{11d \to \mathrm{IIB}}$ from a twist of 11d supergravity on $T^2 \times \mathbb{R}^5 \times X^2$ to the aforementioned twist of type IIB, placed on $\mathbb{R}^4 \times \mathbb{C}^\times \times X^2$. Theorem 4.9 shows that this map is $SL(2,\mathbb{Z})$-equivariant, as expected from physical considerations.

Along the way, we make an unusual number of remarks as we make conjectures of less precise form; this is due to lack of development of the framework and working out each precisely should be regarded as a research project on its own. Examples include Remark 4.5 on what we call M2 cohomology, a conjectural deformation of the de Rham complex of a $G_2$ manifold based on [12], or Remark 4.10 on a perspective on the Gromov–Witten/Donaldson–Thomas correspondence [13] from our framework.

Finally, in Section 5, we study possible consequences of our definition of twisted S-duality on D-brane gauge theory by way of the closed-open map from the closed string states to deformations of D-brane gauge theory introduced in Subsection 2.2. Then deformations of a twisted supersymmetric field theory, viewed as an open string field theory associated to a twist of superstrings, admit their preimages under the closed-open map in the space of closed string fields. By looking at how S-duality acts on those closed string fields, we can check which deformations of D-brane gauge theory are S-dual to one another.

The first example of interest is what we call HT(A)-deformation and HT(B)-deformation of Kapustin's holomorphic-topological twist of 4d $\mathcal{N} = 4$ which are S-dual to each other. Consider such a deformed theory on the spacetime of the form $\Sigma \times C$ where $\Sigma, C$ are both Riemann surfaces. Taking compactification along $C$ yields the B-model with target $\mathrm{Bun}_G(C)_{\mathrm{dR}}$ and B-model with target $\mathrm{Flat}_G(C)$, respectively; the identification in terms of global derived stacks is based on the work of the second author with Chris Elliott [14]. By taking the category of boundary conditions and identifying them via the proposed S-duality, we would obtain the de Rham geometric Langlands correspondence for $G = \mathrm{GL}_n$, modulo the subtlety of the nilpotent singular support condition whose physical meaning is addressed in the sequel [15] (see Subsection 5.2).

**Remark 1.1.** Note that our proposal for finding a physical context for geometric Langlands correspondence is crucially different from the one by Kapustin and Witten [9], which is revisited in a paper [16] from a mathematical perspective. The fundamental point of departure is that we consider a torus in the topological direction of the M-theory background and hence the dilaton field is topological. In particular, the gauge theory also only has topological dependence on the coupling constant. For this reason, our analysis doesn't offer any deep insights regarding strong-weak duality. However, for the purpose of getting the de Rham geometric Langlands correspondence, our proposal is a lot more direct and transparent in its nature.

Moreover, our twisted S-duality map can be applied to arbitrary deformations of a gauge theory, producing new infinite family of dual pairs of deformations of a gauge theory. Indeed, some new conjectures are identified: in Subsection 5.3 on superconformal deformations and Omega backgrounds and Subsection 5.4 on relating supergroup 4-dimensional Chern–Simons theory and a certain category of difference modules. Again, one may find surprising that we can identify nontrivial conjectures in this manner, but it is possible because we work in a sector where the dependence on a coupling constant is topological in nature.

Finally, proposed relations between S-duality of type IIB string theory and S-duality for topological string theory were made long before the present paper. Important works in this direction include the paper by Nekrasov, Ooguri, and Vafa [17] and the paper on topological M-theory by Dijkgraaf, Gukov, Neitzke, and Vafa [18]. An implication of such proposals is that the Gromov–Witten/Donaldson–Thomas correspondence [13] can be viewed as a kind of S-duality between the A-model and B-model topological strings in complex dimension three. In Remark 4.10, we suggest a way in which the correspondence of [13] can be realized in our framework, as a kind of S-duality relating two different twists of type IIB.

## 1.2 Conventions

When we describe a field theory, we work with the BV formalism in a perturbative setting. We work with $\mathbb{Z}/2$-grading but may write $\mathbb{Z}$-grading in a way that it gives an expected cohomological degree when it can. By tensor product of the space of fields, we mean the one of nuclear spaces as, for example, explained in [19, Appendix 2]. The crucial property we use is that for manifolds $M, N$, one has $C^\infty(M) \otimes C^\infty(N) = C^\infty(M \times N)$ as well as a similar statement for sections of vector bundles on $M$ and $N$. Algebras are complexified unless otherwise mentioned; in particular, we see $\mathrm{GL}(N)$ when physicists expect to see $\mathrm{U}(N)$.

# 2 Topological strings and BCOV theory

A goal of this paper is to provide a mathematical description of how S-duality acts on certain supersymmetry-protected sectors of type IIB string theory and supergravity. In this section, we wish to establish a mathematical context for discussing such protected sectors in terms of topological string theory.

In Subsection 2.1, we start with a brief mathematical treatment of topological string theory. Although this plays a mostly organizational role for our paper, this setting allows us to rigorously describe many maneuvers familiar to string theorists in terms of data attached to a Calabi–Yau category. In particular, open string field theory and closed string field theory both arise as natural moduli problems determined by categorical data.

Then we move on to reviewing the relation between open and closed sectors in Subsection 2.2; in particular, we discuss that further deformations of gauge theories living on D-branes can be identified with closed string fields in the topological string.

In Subsection 2.3, we introduce a modification of BCOV theory in dimension 2 and 3 as it plays a major role in our later discussion. Moreover, we prove that there is a hidden $\mathrm{SL}(2,\mathbb{Z})$ symmetry on BCOV theory, which we later argue is compatible with expectations for S-duality of a twist of type IIB supergravity (see Theorem 4.9).

## 2.1 Topological string theory

A large portion of topological string theory can be understood in terms of axiomatic 2-dimensional extended topological quantum field theory (TQFT). In fact, for our purposes we will think of topological string theory as such. While we do not use it in an essential way, the language can serve as a useful organizational device. A fully-extended 2d TQFT attaches to a point, a category, and natural moduli problems associated to this category can be viewed as the equations of motion for two separate classical field theories. The first of these is open string field theory and will be discussed in Subsection 2.1.2. The second of these is closed string field theory as will be discussed in Subsection 2.1.3.

### 2.1.1 Review of 2d TQFT

Let us provide a brief review of 2-dimensional TQFT as relevant for our purpose. For more details, refer to the original articles [20, 21]. A 2-dimensional TQFT is a symmetric monoidal functor $Z\colon \mathrm{Bord}_2 \to \mathrm{DGCat}$. Here, $\mathrm{Bord}_2$ is a 2-category whose objects are 0-manifolds, 1-morphisms between objects are 1-manifolds with boundary the given 0-manifolds, and 2-morphisms are 2-manifolds with corresponding boundaries and corners, and $\mathrm{DGCat}$ denotes the 2-category of small DG categories with colimit-preserving functors. In fact, everything should be considered in an $\infty$-categorical context, but we suppress any mention of it as the discussion in this section is mostly motivational.

By the cobordism hypothesis, a fully extended framed 2d TQFT is determined by a fully dualizable object of DGCat, and an object of DGCat is known to be fully dualizable if and only if it is smooth and proper.[1] Additionally, we often wish to consider oriented theories, which are likewise determined by Calabi–Yau categories. In other words, a smooth proper Calabi–Yau category determines an oriented 2-dimensional extended TQFT [20,21]. Moreover, by only considering 2-manifolds where each connected component has at least one outgoing boundary component, one can similarly consider a version of oriented 2d TQFT determined by a not necessarily proper Calabi–Yau category [21, Section 4.2]. In what follows, we will freely remove any compactness assumption on spaces at the price of working with the latter version.

Given a DG category $\mathcal{C}$, one can consider its Hochschild chains $\mathrm{Hoch}_\bullet(\mathcal{C})$ and Hochschild homology $\mathrm{HH}_\bullet(\mathcal{C})$. It is a well-known fact that Hochschild chains admit an action of a circle $S^1$. Now, note that in terms of the 2d framed TQFT $Z_\mathcal{C}$ determined by a smooth proper DG category $\mathcal{C}$, one has $\mathrm{Hoch}_\bullet(\mathcal{C}) = Z_\mathcal{C}(S^1_{\mathrm{cyl}})$, namely, what $Z_\mathcal{C}$ assigns to a circle $S^1_{\mathrm{cyl}}$ with the cylinder framing. This is the 2-framing induced from the canonical framing of $S^1$. From this perspective, the fact that one may rotate such a circle without changing the framing is responsible for the $S^1$-action on Hochschild chains.

On the other hand, one may also consider Hochschild cochains $\mathrm{Hoch}^\bullet(\mathcal{C})$ and Hochschild cohomology $\mathrm{HH}^\bullet(\mathcal{C})$ of a DG category $\mathcal{C}$. Reasoning similar to the above explains why Hochschild cochains admit the structure of an $\mathbb{E}_2$-algebra. Indeed, by considering $Z_\mathcal{C}(S^1_{\mathrm{ann}})$ for a circle $S^1_{\mathrm{ann}}$ with the annulus framing, since a pair of pants diagram can be drawn in a way that respects the framing, $Z_\mathcal{C}(S^1_{\mathrm{ann}})$ is seen to have the desired algebra structure.

For a smooth proper Calabi–Yau DG category $\mathcal{C}$, or equivalently, an oriented 2d TQFT $Z_\mathcal{C}$ determined by it, these two spaces are identified up to a shift, in which case the two structures are combined to yield a BV algebra structure. Physically, we may think of such an identification between Hochschild chains and Hochschild cochains as a state-operator correspondence.

In our setting, we are interested in a particular type of topological string theory. Let $M$ be a (compact) symplectic manifold of real dimension $2m$ and let $X$ be a (compact) Calabi–Yau manifold of complex dimension $n$ such that $2m + 2n = 10$. Then by a *mixed A-B topological string theory with target $M \times X$*, which we succinctly name as a topological string theory on $M_A \times X_B$, we mean the 2-dimensional oriented TQFT determined by the Calabi–Yau 5-category

$$\mathrm{Fuk}(M) \otimes \mathrm{Perf}(X),$$

where $\mathrm{Fuk}(M)$ refers to a Fukaya category of $M$ and $\mathrm{Perf}(X)$ is the DG category of perfect complexes of $X$.[2] This Calabi–Yau 5-category should be thought of as the category of D-branes for the topological string theory on $M_A \times X_B$.

**Note 2.1.** We do not specify which version of Fukaya category we are considering here – the above generality is only used for narrative purposes. The primary example of relevance for us is when $M = T^*S^1$ as appearing in Subsection 4.2, which admits an explicit description. In view of that, let us note that when the symplectic manifold $M$ is of the form $M = T^*N$, the corresponding wrapped Fukaya category is, roughly speaking, equivalent to the category of modules over the algebra of chains $C_\bullet(\Omega_*N)$ where $\Omega_*N$ is the based loop space [23].

---

[1] For the definition of a DG category being smooth and proper, see, for instance, [22]. It is also known that for a quasi-compact quasi-separated scheme $X$, the DG category $\mathrm{Perf}(X)$ of perfect complexes on $X$ is smooth and proper if and only if $X$ is smooth and proper in the classical sense, respectively.

[2] The 2-category DGCat of dg categories is a symmetric monoidal category. For instance, we have $\mathrm{Perf}(X) \otimes \mathrm{Perf}(Y) \simeq \mathrm{Perf}(X \times Y)$.

### 2.1.2 Topological open string field theory

We now turn to describing the open string field theory of a topological string theory. Given a topological string theory described by a smooth Calabi–Yau $d$ category $\mathcal{C}$, we may think of an object $\mathcal{F} \in \mathcal{C}$ as describing a locus on spacetime on which open strings can end. The dynamics of open strings stretched from the support of $\mathcal{F}$ to itself are described by a field theory called the *D-brane gauge theory* or *world-volume theory*.

Before specializing to the examples of primary interest, let us briefly describe a nonperturbative characterization of such field theories, utilizing language from the work of Brav and Dyckerhoff [24]. Though we will primarily work perturbatively in the main body of this paper, this characterization serves as a useful organizational device.

Given a smooth Calabi–Yau $d$ category, one may consider its moduli of objects $\mathcal{M}_{\mathcal{C}}^{\mathrm{obj}}$. This is a prestack associated to $\mathcal{C}$ whose $U$-points are given by $\mathcal{M}_{\mathcal{C}}^{\mathrm{obj}}(U) := \mathrm{Hom}_{\mathrm{DGCat}}(\mathcal{C}, \mathrm{Perf}\, U)$; we think of this as defining a sort of universal open string field theory. In the work [24], it is shown that the Calabi–Yau structure on $\mathcal{C}$ equips $\mathcal{M}_{\mathcal{C}}^{\mathrm{obj}}$ with a $(2-d)$-shifted symplectic structure. Now given a point $\mathcal{F} \in \mathcal{M}_{\mathcal{C}}^{\mathrm{obj}}$, we may consider the $(-1)$-shifted tangent complex $\mathbb{T}_{\mathcal{F}}[-1]\mathcal{M}_{\mathcal{C}}^{\mathrm{obj}}$ – this is an $L_\infty$-algebra. Furthermore, it is also proven in [24] that there is an equivalence $\mathbb{T}_{\mathcal{F}}[-1]\mathcal{M}_{\mathcal{C}}^{\mathrm{obj}} \cong \mathbb{R}\mathrm{Hom}_{\mathcal{C}}^{\bullet}(\mathcal{F}, \mathcal{F})$ as $L_\infty$-algebras where the $L_\infty$-structure on the right hand side comes from skew-symmetrizing the natural $A_\infty$-structure. Points of the moduli of objects correspond to left-proper objects of $\mathcal{C}$; for such objects, the $(2-d)$-shifted symplectic structure equips $\mathbb{T}_{\mathcal{F}}[-1]\mathcal{M}_{\mathcal{C}}^{\mathrm{obj}}$ with an odd pairing and defines the worldvolume theory of $\mathcal{F}$ as a perturbative BV theory.

Let us explicate this in the example of the B-model on a Calabi–Yau variety $X$ of odd dimension $d$. The corresponding Calabi–Yau category is the dg category $\mathrm{Coh}(X)$ of coherent sheaves on $X$. The general story in the previous paragraph tells us that for a D-brane $\mathcal{F} \in \mathrm{Coh}(X)$, the algebra of open string states ending on it is given by the DG algebra $\mathbb{R}\mathrm{Hom}_{\mathrm{Coh}(X)}^{\bullet}(\mathcal{F}, \mathcal{F})$. Moreover, as the space of fields of a field theory is a local entity, we consider the sheaf of DG algebras $\mathbb{R}\underline{\mathrm{Hom}}_{\mathrm{Coh}(X)}(\mathcal{F}, \mathcal{F})$. A special case of interest for us is when we wrap $N$ coincident D-branes along a complex submanifold $Y$, meaning that $\mathcal{F}$ is the trivial vector bundle of rank $N$ over a subvariety $Y$ of $X$. In this case, the above sheaf describes the sheaf of holomorphic sections of $\wedge^\bullet N_{X/Y}$ where $N_{X/Y}$ is the normal bundle of $Y$ in $X$. In the approach to the BV formalism we utilize, spaces of fields are locally free over smooth functions – therefore, we consider the Dolbeault resolution

$$\mathcal{E} = \Omega^{0,\bullet}(Y, \wedge^\bullet N_{X/Y}) \otimes \mathfrak{gl}(N)[1].$$

From the fact that $\mathrm{Coh}(X)$ is a Calabi–Yau category, this space $\mathcal{E}$ has an induced structure of a classical BV theory with a symplectic pairing of degree $2-d$. Again we obtain a field theory in a $\mathbb{Z}$-graded sense exactly when $d = 3$. For other $d$, by regrading one may still be able to get a $\mathbb{Z}$-graded theory. Note that if $X$ were even-dimensional, then the space $\mathcal{E}$ behaves like a field theory except that the symplectic pairing would be of even degree.

**Example 2.2.** Consider topological string theory on $\mathbb{C}_B^5$.

- Consider $\mathbb{C}^2 \subset \mathbb{C}^5$. Computing as above yields

$$\mathcal{E}_{\mathrm{D3}}^{\mathrm{Hol}} = \Omega^{0,\bullet}(\mathbb{C}^2)[\varepsilon_1, \varepsilon_2, \varepsilon_3] \otimes \mathfrak{gl}(N)[1],$$

  where $\varepsilon_1, \varepsilon_2, \varepsilon_3$ are odd variables that can be understood as parametrizing the directions normal to $\mathbb{C}^2$ in $\mathbb{C}^5$ and hence describing transverse fluctuations of the brane (See [15, Section 4] for more detail on this perspective and its consequences). This is the holomorphic twist of 4-dimensional $\mathcal{N} = 4$ supersymmetric gauge theory with gauge group $\mathrm{GL}(N)$ with a certain choice of a twisting datum.

- Consider $\mathbb{C}^5 \subset \mathbb{C}^5$. Computing as above yields

$$\mathcal{E}_{\text{D9}}^{\text{Hol}} = \Omega^{0,\bullet}(\mathbb{C}^5) \otimes \mathfrak{gl}(N)[1].$$

A result of Baulieu [25] identifies this as a holomorphic twist of 10-dimensional $\mathcal{N} = 1$ supersymmetric gauge theory with gauge group GL($N$).

Next, let us consider a symplectic manifold $M$ of dimension $2k$ with odd $k$ as a target for A-type topological string theory. Then a D-brane should be given by a Lagrangian $L$ of $M$. In this case, we should similarly compute its (derived) endomorphism algebra in the Fukaya category. In the current paper, we are mostly interested in the case of $M = \mathbb{R}^{2k}$ for which $N$ coinciding D-branes on $L = \mathbb{R}^k \subset \mathbb{R}^{2k}$ yield a theory described by

$$\mathcal{E} = \Omega^\bullet(\mathbb{R}^k) \otimes \mathfrak{gl}(N)[1].$$

Again if $k$ were even, then we would have gotten an even symplectic pairing on $\mathcal{E}$.

**Example 2.3.** Let $L$ be a 3-manifold and take $M = T^*L$. Then we obtain Chern–Simons theory

$$\mathcal{E} = \Omega^\bullet(L) \otimes \mathfrak{gl}(N)[1],$$

as argued by [26].

In general, given a tensor product $\mathcal{C}_1 \otimes \mathcal{C}_2$ of Calabi–Yau categories, and an object $\mathcal{F}_1 \otimes \mathcal{F}_2 \in \mathcal{C}_1 \otimes \mathcal{C}_2$, we may compute

$$\mathbb{R}\underline{\text{Hom}}_{\mathcal{C}_1 \otimes \mathcal{C}_2}(\mathcal{F}_1 \otimes \mathcal{F}_2, \mathcal{F}_1 \otimes \mathcal{F}_2) \cong \mathbb{R}\underline{\text{Hom}}_{\mathcal{C}_1}(\mathcal{F}_1, \mathcal{F}_1) \otimes \mathbb{R}\underline{\text{Hom}}_{\mathcal{C}_2}(\mathcal{F}_2, \mathcal{F}_2).$$

Again, if it is an odd-dimensional Calabi–Yau category, as is for a topological string theory on $M_A \times X_B$, it would give a field theory. That is, given a D-brane in an arbitrary mixed A-B topological string theory, the corresponding D-brane gauge theory is described by a combination of the above two classes of examples with an odd symplectic pairing.

**Example 2.4.** Having $N$ D3-branes on $\mathbb{R}^2 \times \mathbb{C} \subset \mathbb{R}^4_A \times \mathbb{C}^3_B$ leads to

$$\mathcal{E}_{\text{D3}}^{\text{HT}} = \Omega^\bullet(\mathbb{R}^2) \otimes \Omega^{0,\bullet}(\mathbb{C})[\varepsilon_1, \varepsilon_2] \otimes \mathfrak{gl}(N)[1],$$

which is the holomorphic-topological twist of 4d $\mathcal{N} = 4$ gauge theory with gauge group GL($N$).

**Example 2.5.** Having $N$ D4 branes on $\mathbb{R}^3 \times \mathbb{C} \subset \mathbb{R}^6_A \times \mathbb{C}^2_B$ leads to

$$\mathcal{E}_{\text{D4}}^{\text{H}_1\text{T}_3} = \Omega^\bullet(\mathbb{R}^3) \otimes \Omega^{0,\bullet}(\mathbb{C})[\varepsilon] \otimes \mathfrak{gl}(N)[1],$$

which is a holomorphic-topological twist of 5d $\mathcal{N} = 2$ gauge theory with gauge group GL($N$).

### 2.1.3 Topological closed string field theory

The next aim is to understand closed string field theory and its structures [27,28] in the context of topological string theory. Once again, we begin by hinting at a more general framework for narrative purposes; one can take our discussion of BCOV theory in Subsection 2.1.4 as the starting point of a mathematical discussion of closed string field theory.

Let $Z$ be a topological string theory in the above sense. Naively, one may think of the space of closed string states as $Z(S^1)$. However, in the physical theory, the worldsheet theory is a 2-dimensional conformal field theory coupled to 2-dimensional gravity; in particular, the space of closed string states should be invariant under the group Diff($S^1$) of diffeomorphisms of $S^1$

acting by reparametrizing the boundary components of the worldsheet. In our topological setting, this amounts to taking the $S^1$-invariant space $Z(S^1)^{S^1}$. In a categorical setting, for a Calabi–Yau DG category $\mathcal{C}$, we define its cyclic cochain complex to be $\mathrm{Cyc}^\bullet(\mathcal{C}) := \mathrm{Hoch}^\bullet(\mathcal{C})^{S^1}$. An expectation is that there is a natural way to equip it with the structure of a Poisson degenerate field theory in the sense of [29] (at least when a topological string theory $Z$ is local in nature, for example, a B-model).

Here the space $V^{S^1}$ of homotopy $S^1$-invariants for a cochain complex $(V, d_V)$ with an action of $S^1$, or equivalently a cochain map $C_\bullet(S^1) \to \mathrm{End}(V)$, may be modeled by

$$V^{S^1} = (V[\![t]\!], d_V + tB),$$

where $t$ is of cohomological degree 2 and $B \colon V \to V$ is an operator of cohomological degree $-1$, which is the image of the fundamental class $[S^1]$ under the map $C_\bullet(S^1) \to \mathrm{End}(V)$. In our case where $V = \mathrm{Hoch}^\bullet(\mathcal{C})$, the operator $B$ is precisely Connes's $B$ operator and we obtain that the shifted cyclic cochains $\mathrm{Cyc}^\bullet(\mathcal{C})[1]$ have the structure of an $L_\infty$-algebra. Moreover, the Calabi–Yau structure of $\mathcal{C}$ yields a cyclic $L_\infty$-algebra structure on the shifted cyclic cochains $\mathrm{Cyc}^\bullet(\mathcal{C})[1]$. The cyclic $L_\infty$-structure may be thought of as coming from a realization of $\mathrm{Cyc}^\bullet(\mathcal{C})[1]$ as the $-1$ shifted tangent complex at $\mathcal{C}$ in the moduli of smooth proper Calabi–Yau $d$ categories [30].

Moreover, a folklore theorem asserts that the formal neighborhood of $\mathcal{C}$ in the moduli of smooth proper Calabi–Yau $d$-categories is odd shifted Poisson. Indeed, consider the periodic cyclic cochains $\mathrm{PCyc}^\bullet(\mathcal{C})$, which can be identified with the Tate fixed points of $\mathrm{Hoch}^\bullet(\mathcal{C})$, where the space $V^{\mathrm{Tate}, S^1}$ of Tate fixed points for the $S^1$ action on $(V, d_V)$ is modeled by

$$V^{\mathrm{Tate}, S^1} = (V(\!(t)\!), d_V + tB).$$

Using the trace pairing and the residue pairing, $\mathrm{PCyc}^\bullet(\mathcal{C})$ has a canonical symplectic structure of degree $6-2d$ where $d$ is the Calabi–Yau dimension of $\mathcal{C}$. Now from the canonical embedding $\mathrm{Cyc}^\bullet(\mathcal{C}) \to \mathrm{PCyc}^\bullet(\mathcal{C})$, the failure of the differential $B$ preserving a splitting of $\mathrm{PCyc}^\bullet(\mathcal{C})$ into $\mathrm{Cyc}^\bullet(\mathcal{C})$ and its complement induces a Poisson structure on $\mathrm{Cyc}^\bullet(\mathcal{C})$ of degree $2d-5$. Note that it is only for $d=3$ that naturally is equipped with a $\mathbb{P}_0$-structure in a $\mathbb{Z}$-graded sense.

All these combined, if one can find a local model $\mathcal{E}$ for $\mathrm{Cyc}^\bullet(\mathcal{C})[2]$, then it would have the structure of a degenerate field theory by construction.

**Example 2.6.** Consider a (compact) Calabi–Yau 5-fold $X$ and the topological string theory determined by $\mathcal{C} = \mathrm{Coh}(X)$. We want to identify the corresponding closed string field theory. For more details, we refer the reader to Subsection 2.1.4.

We know $Z_{\mathcal{C}}(S^1)$ is identified with Hochschild cochains. By the HKR theorem (and the Kontsevich formality [31] for an $L_\infty$-equivalence after shifting), the Hochschild cochain complex is identified with the space of polyvector fields $\mathrm{PV}(X) = \bigoplus_{i,j} \mathrm{PV}^{i,j}(X)$ with the differential $\bar{\partial}$ (and the Lie bracket $[-,-]_{\mathrm{SN}}$, the Schouten–Nijenhuis bracket), where $\mathrm{PV}^{i,j}(X) = \Omega^{0,j}(X, \wedge^i T_X)$. Moreover, one finds $B = \partial \colon \mathrm{PV}^{i,j}(X) \to \mathrm{PV}^{i-1,j}(X)$. This leads to a description of closed string field theory of the topological string theory associated to $\mathrm{Coh}(X)$ as

$$\mathcal{E}(X) = (\mathrm{PV}(X)[\![t]\!][2]; \bar{\partial} + t\partial, [-,-]_{\mathrm{SN}}).$$

Finally, from the symplectic pairing on $\mathrm{PV}(X)(\!(t)\!)$ given by

$$\omega(f(t)\mu, g(t)\nu) = (\mathrm{Res}_{t=0} f(t)g(-t)) \cdot \mathrm{Tr}(\mu\nu),$$

it induces the kernel $(\partial \otimes 1)\delta_{\mathrm{diag}}$ for the shifted Poisson structure on $\mathcal{E}(X)$.

In fact, more is true in this example. As explained in the original paper [32], by a highly nontrivial $L_\infty$-quasi-isomorphism, one can find another description of this degenerate field theory that admits a description in terms of a local action functional, as discussed in Subsection 2.1.4.

**Note 2.7.** Hochschild (co)homology of a Fukaya category Fuk($M$) on a general symplectic manifold $M$ is known to be closely related to quantum cohomology QH($M$) [33,34]. In particular, it is very hard to imagine a local cochain model which encodes all the non-perturbative information. Our interest will be restricted to the case where we consider $M_A \times X_B$ with $M = T^*N$: then Note 2.1 suggests that the closed string field theory should be in terms of $C_\bullet(LN)$ and $\mathcal{E}(X)$, where $LN$ is the free loop space and $C_\bullet(LN)$ is identified as the space of Hochschild (co)chains of $C_\bullet(\Omega_*N)$-mod. This will be used in Subsection 3.2.

Finally, one should note that by definition a closed string field gives a deformation of the category one started with. For example, consider $\mathcal{C} = \mathrm{Coh}(X)$ for a smooth scheme $X$. Then first-order deformations of the category are classified by the second Hochschild cohomology $\mathrm{HH}^2(\mathrm{Coh}(X)) = \mathrm{HH}^2(X)$. By the HKR theorem, we have that $\mathrm{HH}^2(X) \cong H^0(X, \wedge^2 T_X) \oplus H^1(X, T_X) \oplus H^2(X, \mathcal{O}_X)$; $H^1(X, T_X)$ classifies geometric deformations of $X$, $H^0(X, \wedge^2 T_X)$ deformations of $\mathcal{O}_X$ as an associative algebra, and $H^2(X, \mathcal{O}_X)$ gerbal deformations of $\mathrm{Coh}(X)$. See, for instance, [35] for a concrete description of the deformed category.

As another example, we think of $W \in \mathrm{HH}^0(\mathrm{Coh}(X)) \cong H^0(X, \mathcal{O}_X)$ giving a deformation as a $\mathbb{Z}/2$-graded category, by introducing a degree 2 parameter $\beta$; a folklore theorem says that the resulting category can be identified as a matrix factorization category $\mathrm{MF}(W)$.

Moreover, the degree 2 element of the cyclic cohomology, or equivalently, degree 0 element of the associated closed string field theory, precisely gives a first-order deformation of $\mathcal{C}$ as a Calabi–Yau category. In other words, those deformations corresponding to the second Hochschild cohomology respect the Calabi–Yau structure if they define classes of the second cyclic cohomology.

### 2.1.4 BCOV theory

In this subsection we discuss BCOV theory as the main example of closed string field theory. We will discuss and study its variants in Subsection 2.3.

BCOV theory was originally introduced in the seminal work of Bershadsky–Cecotti–Ooguri–Vafa [36] under the name of Kodaira–Spencer theory of gravity for a Calabi–Yau 3-fold, as a string field theory for the topological B-model that describes deformations of complex structures. It was later generalized by Costello and Li [32] for an arbitrary Calabi–Yau manifold.

Let us briefly review the set-up. Let $X$ be a $d$-dimensional Calabi–Yau manifold with a holomorphic volume form $\Omega_X$. Consider the space $\mathrm{PV}^{i,j}(X) = \Omega^{0,j}(X, \wedge^i T_X)$, where $T_X$ is the holomorphic tangent bundle of $X$; this is a locally free resolution of the sheaf of holomorphic $i$-polyvector fields on $X$. We let $\mathrm{PV}(X) = \bigoplus_{i,j} \mathrm{PV}^{i,j}(X)$ where the summand $\mathrm{PV}^{i,j}(X)$ lives in cohomological degree $i + j$; accordingly, we write $|\mu| = i + j$ for $\mu \in \mathrm{PV}^{i,j}(X)$. Then $\mathrm{PV}(X)$ is a commutative differential graded algebra with the differential $\bar{\partial}$ and the product being the wedge product. Contracting with $\Omega_X$ yields an identification

$$(-) \vee \Omega_X \colon \ \mathrm{PV}^{i,j}(X) \cong \Omega^{d-i,j}(X).$$

This allows us to transfer the operator $\partial$ on $\Omega^{\bullet,\bullet}(X)$ to yield operator $\partial$ on $\mathrm{PV}^{\bullet,\bullet}(X)$. The result $\partial \colon \mathrm{PV}^{i,j}(X) \to \mathrm{PV}^{i-1,j}(X)$ is the divergence operator with respect to a holomorphic volume form $\Omega_X$; it is a second-order differential operator of cohomological degree $-1$ so that

$$[\mu, \nu]_{\mathrm{SN}} := (-1)^{|\mu|-1} \left( \partial(\mu\nu) - (\partial\mu)\nu - (-1)^{|\mu|}\mu(\partial\nu) \right),$$

defines a Poisson bracket of degree $-1$. Here the sign factor is necessary to precisely match with the Schouten–Nijenhuis bracket, explaining the notation. One can check $\partial$ is a shifted

derivation of $[-,-]_{SN}$ in that

$$\partial[\mu, \nu]_{SN} = [\partial \mu, \nu]_{SN} + (-1)^{|\mu|-1}[\mu, \partial \nu]_{SN},$$

holds. It also satisfies a shifted version of graded Lie algebra axioms in that the following hold:

$$[\alpha, \beta]_{SN} = -(-1)^{(|\alpha|-1)(|\beta|-1)}[\beta, \alpha]_{SN},$$
$$[\alpha, [\beta, \gamma]_{SN}]_{SN} = [[\alpha, \beta]_{SN}, \gamma]_{SN} + (-1)^{(|\alpha|-1)(|\beta|-1)}[\beta, [\alpha, \gamma]_{SN}]_{SN}.$$

In fact, $(PV(X)[1], \bar{\partial}, [-,-]_{SN})$ is a differential graded Lie algebra. We also consider the trace map

$$\text{Tr}: PV_c(X) \to \mathbb{C}, \qquad \text{given by} \qquad \mu \mapsto \int_X (\mu \vee \Omega_X) \wedge \Omega_X,$$

where $PV_c(X)$ stands for compactly supported sections. Note that the trace map is non-trivial only on $PV_c^{d,d}(X)$.

Consider $PV(X)[2]$ so that the summand $PV^{1,1}(X)$ is in degree 0. This part of the fundamental fields of the theory govern deformations of the complex structure on $X$, which is the reason why it was originally called Kodaira–Spencer gravity. Consider the Schouten–Nijenhuis bracket and the shifted Poisson structure given by the kernel $(\partial \otimes 1)\delta_{\text{diag}}$. Then this immediately gives the free part of the action functional of BCOV theory as $\frac{1}{2}\text{Tr}(\mu\bar{\partial}\partial^{-1}\mu)$, which is precisely the form as discussed by the original work [36].

As motivated in Subsection 2.1.3 and discussed in [32], we use $\mathcal{E}_{\text{BCOV}}(X)$ to denote the resolution of the subcomplex of divergence-free polyvector fields $(\ker \partial)(X) \subset PV(X)[2]$ given by the sheaf of cochain complexes $(PV(X)[\![t]\!][2], Q = \bar{\partial} + t\partial)$ where $t$ is a parameter of cohomological degree 2. We have the $t$-linear extension of the Schouten–Nijenhuis bracket that we still denote by $[-,-]_{SN}$. The shifted Poisson structure $\{-,-\}$ given by the kernel $(\partial \otimes 1)\delta_{\text{diag}}$ equip $\mathcal{E}_{\text{BCOV}}(X)$ with the structure of a Poisson degenerate BV theory in the sense of [29] – this is precisely the closed string field theory of the topological B-model as discussed in Subsection 2.1.3.

Moreover, it is shown by Costello–Li [32, Section 6] that the DG Lie algebra $(PV(X)[\![t]\!][2], Q = \bar{\partial} + t\partial, [-,-]_{SN})$ is quasi-isomorphic to another $L_\infty$-algebra on $PV(X)[\![t]\!][2]$ that is determined by a vector field of the form $Q + \{I_{\text{BCOV}}, -\}$. Here the "interaction term" $I_{\text{BCOV}}$ is a local functional $I_{\text{BCOV}}$ of cohomological degree $6-2d$ on $\mathcal{E}_{\text{BCOV}}(X)$ that satisfies the classical master equation $QI + \frac{1}{2}\{I, I\} = 0$. To define it, we first introduce the notation $\langle - \rangle_0 \colon \text{Sym}(PV(X)[\![t]\!]) \to PV(X)$ for the descendent integral given by

$$\langle t^{k_1}\mu_1, t^{k_2}\mu_2, \cdots, t^{k_n}\mu_n \rangle_0 = \left( \int_{\overline{\mathcal{M}}_{0,n}} \psi_1^{k_1} \cdots \psi_n^{k_n} \right) \mu_1 \cdots \mu_n = \binom{n-3}{k_1, k_2, \cdots, k_n} \mu_1 \cdots \mu_n.$$

Then one can succinctly write

$$I_{\text{BCOV}}(\mu) = \text{Tr}\langle e^\mu \rangle_0.$$

To explain the point of introducing this more complicated $L_\infty$-structure, recall that for an ordinary non-degenerate perturbative BV theory $\mathcal{E}$, an action functional on $\mathcal{E}$ and $L_\infty$-structure on $\mathcal{E}$ is interchangeable, as any vector field is Hamiltonian in the formal moduli setting. On the other hand, BCOV theory is a Poisson degenerate BV theory, so its $L_\infty$-structure doesn't necessarily come from an action functional. However, the above claim exhibits that the $L_\infty$-structure indeed comes from an action functional, which is strictly more data than a mere $L_\infty$-structure. Henceforth, by BCOV theory we mean the sheaf of cochain complexes $(PV(X)[\![t]\!][2], Q = \bar{\partial} + t\partial)$ together with the shifted Poisson structure $\{-,-\}$ induced from the kernel $(\partial \otimes 1)\delta_{\text{diag}}$ as well as $L_\infty$-structure induced from $\{I_{\text{BCOV}}, -\}$.

## 2.2 Coupling between open and closed sectors

We now discuss the relations between the closed string sector (or supergravity) and open string sector (or D-brane gauge theory). We discuss how an element of the space of closed string states yields a deformation of D-brane gauge theory in Subsection 2.2.1 and how conversely having D-brane yields a closed string field as a flux sourced by the brane in Subsection 2.2.2.

### 2.2.1 Closed-open map

This subsection is based on [10, Subsection 7.2]. For more details, the readers are advised to refer to the original paper.

Consider the topological closed string field theory described by $\mathcal{E}_{\text{BCOV}}(X)$ on a Calabi–Yau manifold $X$. As we have learned, when we consider D-branes of twisted type IIB string theory, we have a D-brane gauge theory living on them. Physically, whenever we have a BRST closed element of a closed string field theory, it yields a deformation of the D-brane gauge theory since those theories are coupled. This construction is implemented via the closed-open map. In our setting of twisted string theory or topological string theory, this can be understood in a conceptual way because we can consider the category of boundary conditions for the B-model, namely, $\text{Coh}(X)$. The closed-open map then codifies the idea that a deformation of a category should induce deformations of the endomorphisms of every object. The following theorem can be understood as a closed-open map with the universal target $\text{Coh}(X)$.

**Theorem 2.8.** [31, 37] Let $X$ be a Calabi–Yau manifold. There is an equivalence of $L_\infty$-algebras $(\text{PV}(X)[\![t]\!][1], \bar{\partial} + t\partial, [-,-]_{\text{SN}}) \to \text{Cyc}^\bullet(\text{Coh}(X))[1]$.

This $L_\infty$-morphism is complicated to describe but fortunately we don't need to keep track of the higher maps for our purpose and will explicitly describe the map after taking cohomology.

The theorem succinctly encodes the coupling information between the closed string field theory and D-brane gauge theory. To see this, first note that for a D-brane $\mathcal{F} \in \text{Coh}(X)$, one always has a map $\text{Hoch}^\bullet(X) \to \text{Hoch}^\bullet(\mathbb{R}\underline{\text{Hom}}_{\text{Coh}(X)}(\mathcal{F}, \mathcal{F}))$. Identifying $\text{Hoch}^\bullet(X)$ with $\text{PV}(X)$ via the HKR theorem, the Calabi–Yau structure equips $\text{PV}(X)$ with $\partial$ and $\text{Hoch}^\bullet(\mathbb{R}\underline{\text{Hom}}_{\text{Coh}(X)}(\mathcal{F}, \mathcal{F}))$ with the Connes $B$-operator in a compatible way, yielding a map of cochain complexes $\text{PV}(X)[\![t]\!] \to \text{Cyc}^\bullet(\mathbb{R}\underline{\text{Hom}}_{\text{Coh}(X)}(\mathcal{F}, \mathcal{F}))$. Note that a cyclic cohomology class gives a first-order deformation as an $A_\infty$-algebra with a trace pairing, which precisely gives a deformation of the gauge theory one would construct out of $\mathbb{R}\underline{\text{Hom}}_{\text{Coh}(X)}(\mathcal{F}, \mathcal{F})$ for an odd Calabi–Yau manifold $X$. Then, the main content of the theorem is that with care about higher maps, this can be done in such a way that respects formal deformation theory.

As mentioned, our main interest is when $X$ is a flat space and after we take cohomology. For example, consider $X = \mathbb{C}^5$ and D3 branes on $\mathbb{C}^2 \subset \mathbb{C}^5$ so that we obtain $\text{Hom}^\bullet_{\text{Coh}(\mathbb{C}^5)}(\mathcal{O}_{\mathbb{C}^2}, \mathcal{O}_{\mathbb{C}^2}) \cong \mathcal{O}(\mathbb{C}^2)[\varepsilon_1, \varepsilon_2, \varepsilon_3] = \mathcal{O}(\mathbb{C}^{2|3})$. Then after taking cohomology the map of interest is from the canonical map $\text{PV}_{\text{hol}}(\mathbb{C}^5) \to \text{PV}_{\text{hol}}(\mathbb{C}^{2|3})$. Here $\text{PV}_{\text{hol}} = \oplus_k \text{PV}_{\text{hol}}^k$ with $\text{PV}_{\text{hol}}^k = \text{PV}_{\text{hol}}^{k,0} \subset \text{PV}^{k,0}$ is the space of holomorphic polyvector fields, that is, consisting of those which are in the kernel of $\bar{\partial}$. The map is the identity map on $\mathbb{C}^2$, whereas for the normal coordinates $w_1, w_2, w_3$ of $\mathbb{C}^2 \subset \mathbb{C}^5$, the map is given by a Fourier transform $w_i \mapsto \partial_{\varepsilon_i}$ and $\partial_{w_i} \mapsto \varepsilon_i$. Recall that having a formal parameter $t$ together with additional differential $t\partial$ amounts to considering $\ker \partial$ in our formulation. In our model of twisted supergravity theory, we will use a modified BCOV theory (see Subsection 2.3) with additional $\ker \partial \subset \text{PV}^{d,\bullet}(X)$, which should get sent to zero for a degree reason.

In other words, a first-order deformation of a D-brane gauge theory, which should be described by an element of a cyclic cohomology class, can be represented by a closed string state as desired. Note that the number of D-branes does not matter because Hochschild cohomology

is Morita-invariant and $\mathfrak{gl}(N)$ is Morita-trivial. This is compatible with the expectation that a deformation given by a closed string state should work for arbitrary $N$ in a uniform way.

**Remark 2.9.** Nothing in this discussion crucially depends on the fact that $X$ is of dimension 5. Indeed, for our main application, we will consider a theory on $\mathbb{R}^4_A \times \mathbb{C}^3_B$ which corresponds to the case of $X = \mathbb{C}^3$.

### 2.2.2 Boundary states and fields sourced by D-branes

Just as in the physical string theory, branes in the topological string theory also source certain closed string fields that interact with the closed string sector. Mathematically, this means that fixing a D-brane should yield an element of the space of closed string states. Here, we will derive a procedure for computing such an element by examining some constraints on couplings between open and closed string field theories that are forced upon us from TQFT axiomatics.

Consider a mixed A-B topological string theory on $M \times X$ with category of D-branes $\mathcal{C}$ and fix a D-brane $\mathcal{F} \in \mathcal{C}$. To first order, a coupling between the D-brane gauge theory of $\mathcal{F}$ and the closed string field theory is given by a pairing

$$\mathbb{R}\underline{\mathrm{Hom}}_{\mathcal{C}}(\mathcal{F}, \mathcal{F}) \otimes \mathrm{Cyc}^\bullet(X) \to \mathbb{C}.$$

We may equivalently view this as an $S^1$-invariant map

$$\mathbb{R}\underline{\mathrm{Hom}}_{\mathcal{C}}(\mathcal{F}, \mathcal{F}) \otimes \mathrm{Hoch}^\bullet(X) \to \mathbb{C},$$

and using the identification between Hochschild chains and Hochschild cochains afforded by the Calabi–Yau structure of $\mathcal{C}$, we have a map

$$\mathbb{R}\underline{\mathrm{Hom}}_{\mathcal{C}}(\mathcal{F}, \mathcal{F}) \otimes \mathrm{Hoch}_\bullet(X) \to \mathbb{C}.$$

Now this latter map is exactly what the TQFT $Z_{\mathcal{C}}$ assigns to a world-sheet depicting a scattering process where the endpoints of an open string, which are labeled by the brane $\mathcal{F}$, fuse to yield a closed string, which then annihilates with another closed string. This is depicted in the left-hand side of the figure below, and the endpoints of the open string are depicted in blue.

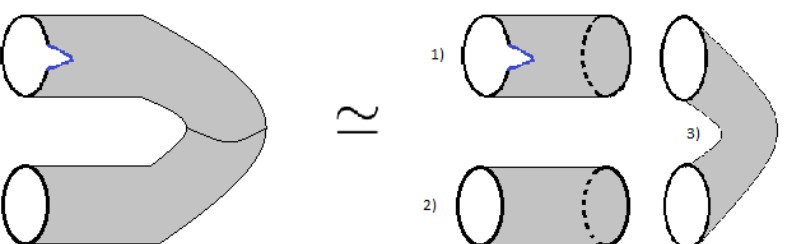

Now, from the functoriality of $Z_{\mathcal{C}}$ with respect to compositions of cobordisms, we may compute $Z_{\mathcal{C}}$ of the left-hand side above, by computing $Z_{\mathcal{C}}$ of each of the pieces of the right-hand side above and composing them appropriately. Accordingly, we have that:

- Applying $Z_{\mathcal{C}}$ to the cobordism labeled 1) above yields a map

$$\partial_{\mathrm{st}} : \mathbb{R}\underline{\mathrm{Hom}}_{\mathcal{C}}(\mathcal{F}, \mathcal{F}) \to \mathrm{Hoch}_\bullet(X).$$

  We suggestively denote the image of $\mathrm{Id}_{\mathcal{F}}$ by $\mathrm{ch}(\mathcal{F})$; this is a mathematical codification of the *boundary state* associated to a boundary condition [38].

- Applying $Z_{\mathcal{C}}$ to the cobordism labeled 2) above yields the identity map

$$\mathrm{Id}\colon \mathrm{Hoch}_{\bullet}(X) \to \mathrm{Hoch}_{\bullet}(X).$$

- Applying $Z_{\mathcal{C}}$ to the cobordism labeled 3) above yields a pairing

$$\mathrm{Tr}\colon \mathrm{Hoch}_{\bullet}(X) \otimes \mathrm{Hoch}_{\bullet}(X) \to \mathbb{C}.$$

Now composing the above, we see that the cobordism on the left-hand side above yields a map

$$\mathbb{R}\underline{\mathrm{Hom}}_{\mathcal{C}}(\mathcal{F},\mathcal{F}) \otimes \mathrm{Hoch}_{\bullet}(X) \to \mathrm{Hoch}_{\bullet}(X) \otimes \mathrm{Hoch}_{\bullet}(X) \to \mathbb{C},$$
$$\mathrm{Id}_{\mathcal{F}} \otimes \alpha \mapsto \mathrm{Tr}(\mathrm{ch}(\mathcal{F}) \otimes \alpha).$$

Finally, appealing to the Calabi–Yau structure of $\mathcal{C}$ once more and letting $\Omega$ denote the isomorphism $\mathrm{Hoch}^{\bullet}(X) \cong \mathrm{Hoch}_{\bullet}(X)$, we see that the desired coupling must be a map

$$\mathbb{R}\underline{\mathrm{Hom}}_{\mathcal{C}}(\mathcal{F},\mathcal{F}) \otimes \mathrm{Hoch}^{\bullet}(X) \to \mathbb{C},$$
$$\mathrm{Id}_{\mathcal{F}} \otimes \mu \mapsto \mathrm{Tr}(\mathrm{ch}(\mathcal{F}) \otimes \Omega(\mu)).$$

For concreteness, let us explicate the above in the case of the topological B-model with target a Calabi–Yau 5-fold $X$, that is, $\mathcal{C} = \mathrm{Coh}(X)$. In this case the HKR theorem gives us isomorphisms $\mathrm{Hoch}_{\bullet}(X) = \Omega^{-\bullet}(X)$, $\mathrm{Hoch}^{\bullet}(X) \cong \mathrm{PV}^{\bullet,\bullet}(X)$, the map $\partial_{\mathrm{st}}$ sends $\mathrm{Id}_{\mathcal{F}}$ to the ordinary Chern character of $\mathcal{F}$, the map $\Omega$ is given by contracting with the Calabi–Yau form $\Omega_X$, and the pairing $\mathrm{Tr}$ is given by wedging and integrating. In sum, the above composition is the map

$$\mathbb{R}\underline{\mathrm{Hom}}_{\mathrm{Coh}(X)}(\mathcal{F},\mathcal{F}) \otimes \mathrm{PV}^{\bullet,\bullet}(X) \to \mathbb{C},$$
$$\mathrm{Id}_{\mathcal{F}} \otimes \mu \mapsto \int_X \mathrm{ch}(\mathcal{F}) \wedge (\mu \vee \Omega_X).$$

In particular, when $X = \mathbb{C}^5$ and $\mathcal{F} = \mathcal{O}_Y^N$ where $Y \subset \mathbb{C}^5$ is a subvariety of complex dimension $k$, then one has $\mathrm{ch}(\mathcal{F}) = N\delta_Y$, where $\delta_Y$ denotes the $(5-k,5-k)$-form with distributional coefficients corresponding to the usual $\delta$-function supported on $Y$. Thus, the right-hand side of the above functional becomes $N\int_Y \mu \vee \Omega_X$.

Now let us incorporate the action of $C_{\bullet}(S^1)$ on $\mathrm{Hom}(\mathrm{PV}^{\bullet,\bullet}(X),\mathbb{C})$ coming from rotating the closed string, in order to get a $\partial$-invariant functional. Explicitly, the action is given by precomposing with $\partial^{-1}$. The image of a map under this action will necessarily be $S^1$-invariant; thus, the desired functional introduced by the presence of the brane $\mathcal{F} = \mathcal{O}_Y^N$ is given by

$$I(\mu) = N \int_Y \partial^{-1}\mu \vee \Omega_X.$$

We note that this is only non-zero on the $(4-k,k)$-component of $\mu$.

Let us see how considering a D-brane $\mathcal{F} = \mathcal{O}_Y^N$ and its coupling in the above sense modifies the equations of motion. For this purpose, we work with the formulation of BCOV theory without the interaction term; thus, the output of our derivation will be a linear approximation to the actual field sourced by a D-brane. The terms in the action functional involving the $\mu^{4-k,k}$ term are

$$\int_X \left(\mu^{k,4-k}\,\bar{\partial}\,\partial^{-1}\mu^{4-k,k} \vee \Omega_X\right) \wedge \Omega_X + N \int_Y \partial^{-1}\mu^{4-k,k} \vee \Omega_X$$
$$= -\int_X \left(\bar{\partial}\,\mu^{k,4-k}\,\partial^{-1}\mu^{4-k,k} \vee \Omega_X\right) \wedge \Omega_X + N \int_Y \partial^{-1}\mu^{4-k,k} \vee \Omega_X,$$

where in the second line we have integrated by parts. Varying with respect to $\partial^{-1}\mu^{4-k,k}$ yields the equations of motion

$$\bar{\partial}\,\mu^{k,4-k} \vee \Omega_X = N\delta_Y\,.$$

Here we should think of $\mu^{k,4-k}$ satisfying the equations of motion as the field sourced by $\mathcal{O}_Y^N$ on $X$.

This motivates the following definition:

**Definition 2.10.** Let $X$ be a Calabi–Yau variety and $\mathcal{F} \in \mathrm{Coh}(X)$ be a coherent sheaf. *A flux density sourced by $\mathcal{F}$ on $X$* is a choice of a representative $F_{\mathcal{F}}$ for a class in $\mathrm{Hoch}^\bullet(X) \cong \mathrm{PV}(X)$ trivializing the Chern character $\mathrm{ch}(\mathcal{F}) \vee \Omega_X^{-1}$ and satisfying $\partial F_{\mathcal{F}} = 0$.

The only example we care about in this paper is the following:

**Example 2.11.** Let $X = \mathbb{C}^d$, $Y = \mathbb{C}^k \subset \mathbb{C}^d$, and $\mathcal{F} = \mathcal{O}_Y^N$. In this case, one can additionally ask for the gauge fixing condition $\bar{\partial}^* F_{\mathcal{F}} = 0$. It turns out that

$$\bar{\partial} F_{\mathcal{F}} \vee \Omega_X = N\delta_Y\,, \qquad \partial F_{\mathcal{F}} = 0\,, \qquad \bar{\partial}^* F_{\mathcal{F}} = 0\,,$$

uniquely characterize a representative, which is the so-called *Bochner–Martinelli kernel* [39].

This can be generalized to the case when the normal bundle $N_{X/Y}$ is trivial.

## 2.3 Further points on BCOV theory

In this subsection, we introduce minimal BCOV theory as well as further variants of minimal BCOV theory in dimensions 2 and 3, and discuss a hidden $\mathrm{SL}(2,\mathbb{Z})$ symmetry on the modified BCOV theory on a Calabi–Yau 3-fold.

### 2.3.1 Modified BCOV theories

Recall that BCOV theory is equipped with a shifted Poisson structure from the kernel $(\partial \otimes 1)\delta_{\mathrm{diag}}$. Note that the Poisson structure pairs components of $\mathrm{PV}^{i,\bullet}(X)$ with $\mathrm{PV}^{d-i-1,\bullet}(X)$. This suggests that $\oplus_{i \leq d-1}\mathrm{PV}^{i,\bullet}(X)$ are the propagating fields while fields in $\mathrm{PV}^{d,\bullet}(X)$ and any summand of $\mathcal{E}_{\mathrm{BCOV}}(X)$ nonconstant in $t$ are certain background fields. One would like to discard the nonpropagting fields, but since the differential $\bar{\partial} + t\partial$ increases degree in $t$ the result would not be closed under the differential. This motivates the following definition.

**Definition 2.12.** *Minimal BCOV theory* $\mathcal{E}_{\mathrm{mBCOV}}(X) \subset \mathcal{E}_{\mathrm{BCOV}}(X)$ is defined by asking the space of fields to be given by the sheaf of subcomplexes

$$\mathcal{E}_{\mathrm{mBCOV}}(X) = \left( \bigoplus_{i+k \leq d-1} t^k\,\mathrm{PV}^{i,\bullet}(X), \overline{\partial} + t\partial \right),$$

and setting $I_{\mathrm{mBCOV}} = I_{\mathrm{BCOV}}|_{\mathcal{E}_{\mathrm{mBCOV}}}$.

This is minimal in the sense that it is the smallest subcomplex containing the propagating fields.

**Example 2.13** (Minimal BCOV theory for Calabi–Yau surfaces)**.** The space $\mathcal{E}_{\mathrm{mBCOV}}(X)$ of fields of the minimal BCOV theory on a Calabi–Yau surface $X$ is

$$\begin{array}{ccc}
\underline{-2} & \underline{-1} & \underline{0} \\
\mathrm{PV}^{0,\bullet}(X) & & \\
& \mathrm{PV}^{1,\bullet}(X) \xrightarrow{\ t\partial\ } t\,\mathrm{PV}^{0,\bullet}(X) &
\end{array}\ ,$$

and the action functional $I_{\text{mBCOV}}$ is

$$I_{\text{mBCOV}}(\alpha^0, \mu^1, t\mu^0) = \frac{1}{2}\operatorname{Tr}\left\langle \alpha^0\mu^1\mu^1 e^{t\mu^0}\right\rangle_0 = \frac{1}{2}\sum_{k\geq 0}\operatorname{Tr}\left(\alpha^0\mu^1\mu^1\left(\mu^0\right)^k\right),$$

where $\alpha^0 \in \text{PV}^{0,\bullet}(X)$ and $(\mu^1, t\mu^0) \in \text{PV}^{1,\bullet}(X) \oplus t\,\text{PV}^{0,\bullet}(X)$.

**Example 2.14** (Minimal BCOV theory for Calabi–Yau 3-folds). One has the space of fields

$$
\begin{array}{ccccc}
\underline{-2} & \underline{-1} & \underline{0} & \underline{1} & \underline{2} \\[4pt]
\text{PV}^{0,\bullet}(X) & & & & \\[8pt]
& \text{PV}^{1,\bullet}(X) \xrightarrow{\ t\partial\ } t\,\text{PV}^{0,\bullet}(X) & & & \\[8pt]
& & \text{PV}^{2,\bullet}(X) \xrightarrow{\ t\partial\ } t\,\text{PV}^{1,\bullet}(X) \xrightarrow{\ t\partial\ } t^2\,\text{PV}^{0,\bullet}(X) &
\end{array}
\quad,
$$

with the action functional given by

$$I_{\text{mBCOV}} = \operatorname{Tr}\left\langle\left(\alpha^0\mu^1\eta^2 + \frac{1}{6}\mu^1\mu^1\mu^1 + \frac{1}{2}\alpha^0\alpha^0\eta^2\left(t\eta^1\right) + \frac{1}{2}\alpha^0\alpha^0\mu^1\eta^2\left(t^2\eta^0\right)\right)e^{t\mu^0}\right\rangle_0$$

$$= \sum_{k\geq 0}\operatorname{Tr}\left(\alpha^0\mu^1\eta^2\left(\mu^0\right)^k + \frac{1}{6}\mu^1\mu^1\mu^1\left(\mu^0\right)^k\right)$$

$$+ \operatorname{Tr}\left\langle\left(\frac{1}{2}\alpha^0\alpha^0\eta^2\left(t\eta^1\right) + \frac{1}{2}\alpha^0\alpha^0\mu^1\eta^2\left(t^2\eta^0\right)\right)e^{t\mu^0}\right\rangle_0,$$

where $\alpha^0 \in \text{PV}^{0,\bullet}(X)$, $(\mu^1, t\mu^0) \in \text{PV}^{1,\bullet}(X) \oplus t\,\text{PV}^{0,\bullet}(X)$, and

$$(\eta^2, t\eta^1, t^2\eta^0) \in \text{PV}^{2,\bullet}(X) \oplus t\,\text{PV}^{1,\bullet}(X) \oplus t^2\,\text{PV}^{0,\bullet}(X).$$

Now, we would like to introduce modifications of minimal BCOV theory for $d \geq 2$. The idea is to consider $\text{PV}^{d,\bullet}(X)$ instead of $\left(\bigoplus_{i+k=d-1} t^k\,\text{PV}^{i,\bullet}(X), \overline{\partial} + t\partial\right)$ and the pullback of minimal BCOV theory along the natural cochain map

$$\partial:\ \text{PV}^{d,\bullet}(X) \to \left( \text{PV}^{d-1,\bullet}(X) \longrightarrow t\,\text{PV}^{d-2,\bullet}(X) \longrightarrow \cdots \longrightarrow t^{d-1}\,\text{PV}^{0,\bullet}(X) \right).$$

Equivalently, recall the isomorphism of cochain complexes

$$\left(\bigoplus_{i+k=d-1} t^k\,\text{PV}^{i,\bullet}(X), \overline{\partial} + t\partial\right) \cong \left(\bigoplus_{i+k=d-1} t^k\Omega^{d-i,\bullet}(X), \overline{\partial} + t\partial\right),$$

induced by contracting the volume form; the right hand side is a resolution of the sheaf of holomorphic, closed 1-forms. Under this isomorphism, the intended modification amounts to replacing this complex with a resolution of the sheaf of holomorphic functions. As we will discuss in subsection 3.2.2, this modification can be interpreted as replacing a theory whose fields are curvatures with a theory whose fields are connections.

Note that $\text{PV}^{d,\bullet}(X)$ should live in the same cohomological degree as $\text{PV}^{d-1,\bullet}(X)$, whose parity is different from the usual one for an element of $\text{PV}^{d,\bullet}(X)$.

To motivate the formal definition for the free part or shifted symplectic/Poisson structure for $d = 2$, recall that if we have usual BV theory described by a shifted $L_\infty$-algebra $\mathcal{E}$ together with a $(-1)$-shifted symplectic pairing $\omega$, then the corresponding action functional would be

$$S(\phi) = S_{\text{free}}(\phi) + I(\phi) = \int \frac{1}{2}\omega(\phi, \ell_1\phi) + \sum_{k\geq 2}\frac{1}{(k+1)!}\omega(\phi, \ell_k(\phi, \phi, \cdots, \phi)), \qquad \phi \in \mathcal{E}.$$

For the free part, if we formally insert things with $\omega(-,-)$ understood as $\mathrm{Tr}((-) \wedge \partial^{-1}(-))$, then for the original (minimal) BCOV functional

$$S_{\mathrm{free}}(\mu) = \mathrm{Tr}\left(\frac{1}{2}\mu \wedge \overline{\partial} \partial^{-1}\mu\right), \qquad \mu \in \mathcal{E}_{\mathrm{mBCOV}}(X),$$

its pullback becomes of the form

$$S(\alpha,\beta) = \mathrm{Tr}\left(\alpha \wedge \overline{\partial}\beta\right), \qquad \alpha \in \mathrm{PV}^{0,\bullet}(X), \quad \beta \in \mathrm{PV}^{2,\bullet}(X).$$

Hence we "derived" the following definition.

**Definition 2.15.** Let $X$ be a Calabi–Yau surface. *Minimal BCOV theory on $X$ with potentials* $\widetilde{\mathcal{E}}_{\mathrm{mBCOV}}(X)$ is the $\mathbb{Z}/2$-graded BV theory with

- underlying space of fields given by the sheaf of cochain complexes

$$\widetilde{\mathcal{E}}_{\mathrm{mBCOV}}(X) = \mathrm{PV}^{0,\bullet}(X)[2] \oplus \mathrm{PV}^{2,\bullet}(X)[1],$$

- odd symplectic structure given by the map $\mathrm{Tr}$, and

- an action functional given by

$$\widetilde{I}_{\mathrm{mBCOV}}(\alpha,\beta) = \frac{1}{2}\mathrm{Tr}(\alpha\partial\beta\partial\beta),$$

where $\alpha \in \mathrm{PV}^{0,\bullet}(X)$ and $\beta \in \mathrm{PV}^{2,\bullet}(X)$.

**Note 2.16.** Since the theory is non-degenerate theory, one may equivalently encode the datum in terms of shifted $L_\infty$-structure. Then we have $\ell_1 = \overline{\partial}$, $\ell_2 = [-,-]$, and $\ell_{n\geq 3} = 0$, where $[-,-]$ is defined in terms of the Schouten–Nijenhuis bracket as follows:

(1) For $\alpha \in \mathrm{PV}^{0,\bullet}(X)$ and $\beta \in \mathrm{PV}^{2,\bullet}(X)$, we define

$$\begin{aligned}
[\alpha,\beta] &:= (-1)^{|\alpha|-1}\partial[\alpha,\beta]_{\mathrm{SN}} &&\in \mathrm{PV}^{0,\bullet}(X),\\
[\beta,\alpha] &:= \partial[\beta,\alpha]_{\mathrm{SN}} &&\in \mathrm{PV}^{0,\bullet}(X),
\end{aligned}$$

where $|\cdot|$ denotes degree as an element in $\mathrm{PV}(X)$.

(2) For $\beta, \tilde{\beta} \in \mathrm{PV}^{2,\bullet}(X)$, we define

$$[\beta,\tilde{\beta}] := (-1)^{|\beta|}[\partial\beta,\tilde{\beta}]_{\mathrm{SN}} = [\beta,\partial\tilde{\beta}]_{\mathrm{SN}} \in \mathrm{PV}^{2,\bullet}(X).$$

(3) The bracket between elements of $\mathrm{PV}^{0,\bullet}(X)$ vanishes.

If we apply the same logic for a Calabi–Yau 3-fold, we obtain the following definition.

**Definition 2.17.** Let $X$ be a Calabi–Yau 3-fold. *Minimal BCOV theory on $X$ with potentials* $\widetilde{\mathcal{E}}_{\mathrm{mBCOV}}(X)$ is the Poisson degenerate BV theory with:

- underlying space of fields given by the sheaf of cochain complexes

$$\mathrm{PV}^{0,\bullet}(X)[2] \oplus (\mathrm{PV}^{1,\bullet}(X)[1] \to t\,\mathrm{PV}^{0,\bullet}(X)) \oplus \mathrm{PV}^{3,\bullet}(X),$$

- shifted symplectic structure between $\mathrm{PV}^{0,\bullet}$ and $\mathrm{PV}^{3,\bullet}$ given by $\mathrm{Tr}$ and shifted Poisson structure given by the kernel $(\partial \otimes 1)\delta_{\mathrm{diag}}$ on $(\mathrm{PV}^{1,\bullet}(X)[1] \to t\,\mathrm{PV}^{0,\bullet}(X))$, and

- an action functional given by

$$\widetilde{I}_{\mathrm{mBCOV}}(\alpha,\mu,t\,\nu,\gamma) = \sum_{k\geq 0} \mathrm{Tr}\left(\alpha\mu\partial\gamma\,\nu^k + \frac{1}{6}\mu\mu\mu\,\nu^k\right),$$

where $\alpha \in \mathrm{PV}^{0,\bullet}(X)$, $(\mu, t\,\nu) \in \mathrm{PV}^{1,\bullet}(X) \oplus t\,\mathrm{PV}^{0,\bullet}(X)$, and $\gamma \in \mathrm{PV}^{3,\bullet}(X)$.

It is obvious by construction that both functionals satisfy the classical master equation. We denote by the map from minimal BCOV theory with potential to minimal BCOV theory $\Phi\colon \widetilde{\mathcal{E}}_{\mathrm{mBCOV}}(X) \to \mathcal{E}_{\mathrm{mBCOV}}(X)$.

**Remark 2.18.** Even if $d > 3$, the above algorithm gives a perfectly reasonable definition of meaningful Poisson degenerate BV theory. On the other hand, the definition doesn't deserve the name of minimal BCOV theory with potentials as we explain now.

To get a feel for the necessary ingredients for the definition deserving the name, it helps to identify the physical meaning of this modification. As explained in detail in Subsection 3.2.2, our replacement of $\left(\bigoplus_{i+k=d-1} t^k\,\mathrm{PV}^{i,\bullet}(X), \overline{\partial} + t\partial\right)$ by $\mathrm{PV}^{d,\bullet}$ amounts to a choice of a potential for certain Ramond–Ramond (RR) field strengths when $d = 5$. This suggests that one should think of the minimal BCOV theory as twisted supergravity theory without any choice of RR forms and our definition as with a choice of a potential, hence the name. On the other hand, if $d > 3$, then for the democratic formulation of superstring/supergravity theory, one needs to make more choices of potentials. For $d = 2$, we made a choice of potential in degree 1, and for $d = 3$, we did in degree 2. However, when $d = 4, 5$, one should make a choice of a potential in two summands, say in degree $d_1$ and in degree $d_2$ such that $d_1 + d_2 \neq d - 1$. In the case of $d = 5$, the last condition exactly corresponds to the fact that one cannot simultaneously make a choice of potentials for field strengths that are electro-magnetic dual to each other. On the contrary, for $d = 4$, one can make such choices without any problem and obtain a non-degenerate $\mathbb{Z}/2$-graded BV theory as a result.

### 2.3.2 An SL$(2,\mathbb{Z})$ symmetry of BCOV theory with potential

The goal of this subsection is to argue that BCOV theory with potential on a Calabi–Yau 3-fold $X$ should admit an action of SL$(2,\mathbb{Z})$ in a model without any descendants. Exhibiting an action respecting an $L_\infty$-structure seems nontrivial so we hope to revisit the topic in the future. On the other hand, what we argue below is enough for all practical applications we have in mind.

Note that minimal BCOV theory with potential has underlying cochain complex

$$\mathrm{PV}^{3,\bullet}(X) \oplus (\mathrm{PV}^{1,\bullet}(X)[1] \to t\,\mathrm{PV}^{0,\bullet}(X)) \oplus \mathrm{PV}^{0,\bullet}(X)[2],$$

with a nontrivial $L_\infty$-bracket. To remove descendants, we just take cohomology with respect to $t\partial$ to replace $(\mathrm{PV}^{1,\bullet}(X)[1] \to t\,\mathrm{PV}^{0,\bullet}(X))$ by $\ker\partial \subset \mathrm{PV}^{1,\bullet}(X)[1]$ to obtain

$$\widetilde{\mathcal{E}}^{\circ}_{\mathrm{mBCOV}} := \mathrm{PV}^{3,\bullet}(X) \oplus (\ker\partial \subset \mathrm{PV}^{1,\bullet}(X)[1]) \oplus \mathrm{PV}^{0,\bullet}(X)[2].$$

Then it becomes DG Lie algebra with $\ell_1 = \overline{\partial}$ and $\ell_2 = [-,-]$ defined by

(1) For $\alpha \in \mathrm{PV}^{0,\bullet}(X)$ and $\gamma \in \mathrm{PV}^{3,\bullet}(X)$, we define

$$[\alpha,\gamma] := (-1)^{|\alpha|-1}\partial[\alpha,\gamma]_{\mathrm{SN}} \quad \in \ker\partial \subset \mathrm{PV}^{1,\bullet}(X),$$
$$[\gamma,\alpha] := \partial[\gamma,\alpha]_{\mathrm{SN}} \qquad\qquad \in \ker\partial \subset \mathrm{PV}^{1,\bullet}(X).$$

(2) For $\mu \in \ker\partial \subset \mathrm{PV}^{1,\bullet}(X)$ and $\gamma \in \mathrm{PV}^{3,\bullet}(X)$, we define

$$[\mu,\gamma] := (-1)^{|\mu|-1}[\mu,\gamma]_{\mathrm{SN}} \quad \in \mathrm{PV}^{3,\bullet}(X),$$
$$[\gamma,\mu] := [\gamma,\mu]_{\mathrm{SN}} \qquad\qquad \in \mathrm{PV}^{3,\bullet}(X).$$

(3) All other brackets are the Schouten–Nijenhuis bracket.

Now we define an action of $\mathrm{SL}(2,\mathbb{Z})$ on $\widetilde{\mathcal{E}}^\circ_{\mathrm{mBCOV}}$. The following definition is motivated by a construction in [40].

**Definition 2.19.** Let $(X, \Omega_X)$ be a Calabi–Yau 3-fold with a holomorphic volume form $\Omega_X$. Recall the presentation of $\mathrm{SL}(2,\mathbb{Z})$ as

$$\mathrm{SL}(2,\mathbb{Z}) = \left\langle S = \begin{pmatrix} 0 & -1 \\ 1 & 0 \end{pmatrix}, \ T = \begin{pmatrix} 1 & 1 \\ 0 & 1 \end{pmatrix} \ \middle| \ S^4 = 1, \ (ST)^3 = S^2 \right\rangle.$$

Denoting elements of $\mathrm{Aut}(\widetilde{\mathcal{E}}^\circ_{\mathrm{mBCOV}})$ with block matrices of the form for a linear transformation on $\mathrm{PV}^{0,\bullet}(X) \oplus \left( \ker \partial \subset \mathrm{PV}^{1,\bullet}(X) \right) \oplus \mathrm{PV}^{3,\bullet}(X)$, the action is given by

$$S \mapsto \begin{pmatrix} & & -(-) \vee \Omega_X \\ & \mathrm{Id} & \\ (-) \wedge \Omega_X^{-1} & & \end{pmatrix}, \qquad T \mapsto \begin{pmatrix} \mathrm{Id} & & (-) \vee \Omega_X \\ & \mathrm{Id} & \\ & & \mathrm{Id} \end{pmatrix}.$$

That is,

- Given $\alpha \in \mathrm{PV}^{0,\bullet}(X)$,

$$S(\alpha) = \alpha \wedge \Omega_X^{-1} \in \mathrm{PV}^{3,\bullet}(X), \qquad T(\alpha) = \alpha \in \mathrm{PV}^{0,\bullet}(X).$$

- Given $\mu \in \ker \partial \subset \mathrm{PV}^{1,\bullet}(X)$,

$$S(\mu) = \mu \in \ker \partial \subset \mathrm{PV}^{1,\bullet}(X), \qquad T(\mu) = \mu \in \ker \partial \subset \mathrm{PV}^{1,\bullet}(X).$$

- Given $\gamma \in \mathrm{PV}^{3,\bullet}(X)$

$$S(\gamma) = -\alpha \in \mathrm{PV}^{0,\bullet}(X), \qquad T(\gamma) = \gamma + \alpha \in \mathrm{PV}(X),$$

  where $\alpha$ is such that $\gamma = \alpha \wedge \Omega_X^{-1}$.

In fact, this action is an action on a DG Lie algebra and extends to the one of $\mathrm{SL}(2,\mathbb{Z})$:

**Theorem 2.20.** Let $(X, \Omega_X)$ be a Calabi–Yau 3-fold. Then $S$ and $T$ generate an action of $\mathrm{SL}(2,\mathbb{Z})$ on the DG Lie algebra $\widetilde{\mathcal{E}}^\circ_{\mathrm{mBCOV}}(X)$.

*Proof.* First, we need to check $S^4 = \mathrm{Id}$ and $(ST)^3 = S^2$; the first is apparent and the second follows from noting $S|_{\ker \partial \subset \mathrm{PV}^{1,\bullet}(X)} = \mathrm{Id} = T|_{\ker \partial \subset \mathrm{PV}^{1,\bullet}(X)}$ and the following chain

$$\alpha \xrightarrow{\ T\ } \alpha \xrightarrow{\ S\ } \gamma \xrightarrow{\ T\ } \gamma + \alpha \xrightarrow{\ S\ } -\alpha + \gamma \xrightarrow{\ T\ } \gamma \xrightarrow{\ S\ } -\alpha \ ,$$

where $\alpha \in \mathrm{PV}^{0,\bullet}(X)$ and $\gamma = \alpha \wedge \Omega_X^{-1} \in \mathrm{PV}^{3,\bullet}(X)$. Now we argue that $S$ and $T$ preserve the DG Lie structure on $\widetilde{\mathcal{E}}^\circ_{\mathrm{mBCOV}}(X)$. For $S$, we have the following cases;

- Let $\alpha \in \mathrm{PV}^{0,\bullet}(X)$ and $\tilde{\alpha} \in \mathrm{PV}^{0,\bullet}(X)$. In this case, $S([\alpha, \tilde{\alpha}]) = 0 = [S(\alpha), S(\tilde{\alpha})]$. This also proves the claim for the bracket between $\gamma, \tilde{\gamma} \in \mathrm{PV}^{3,\bullet}(X)$.

- Let $\alpha \in \mathrm{PV}^{0,\bullet}(X)$ and $\mu \in \ker \partial \subset \mathrm{PV}^{1,\bullet}(X)$. Then

$$S\left([\alpha, \mu]\right) = S\left([\alpha, \mu]_{\mathrm{SN}}\right) = [\alpha, \mu]_{\mathrm{SN}} \wedge \Omega_X^{-1}.$$

On the other hand, we have that

$$[S(\alpha), S(\mu)] = [\alpha \wedge \Omega_X^{-1}, \mu] = [\alpha \wedge \Omega_X^{-1}, \mu]_{\mathrm{SN}}.$$

These two agree because $[-, \mu]_{\mathrm{SN}}$ intertwines the identification

$$(-) \wedge \Omega_X^{-1} \colon \ \mathrm{PV}^{0,\bullet}(X) \cong \mathrm{PV}^{3,\bullet}(X).$$

This also proves the claim for the bracket between $\gamma \in \mathrm{PV}^{0,\bullet}(X)$ and $\mu \in \ker \partial \subset \mathrm{PV}^{1,\bullet}(X)$.

- Let $\alpha \in \mathrm{PV}^{0,\bullet}(X)$ and $\gamma \in \mathrm{PV}^{3,\bullet}(X)$, say $\gamma = \tilde{\alpha} \wedge \Omega_X^{-1}$ for $\tilde{\alpha} \in \mathrm{PV}^{0,\bullet}(X)$. Then

$$S([\alpha,\gamma]) = (-1)^{|\alpha|-1} S\left(\partial [\alpha, \tilde{\alpha} \wedge \Omega_X^{-1}]_{\mathrm{SN}}\right) = (-1)^{|\alpha|-1} \partial [\alpha, \tilde{\alpha} \wedge \Omega_X^{-1}]_{\mathrm{SN}}.$$

On the other hand, we have

$$[S(\alpha), S(\gamma)] = [\alpha \wedge \Omega_X^{-1}, -\tilde{\alpha}] = -\partial [\alpha \wedge \Omega_X^{-1}, \tilde{\alpha}]_{\mathrm{SN}}.$$

One can check that if $\partial(\alpha \wedge \Omega_X^{-1}) = 0$, then both terms vanish. Suppose $\partial(\alpha \wedge \Omega_X^{-1}) \neq 0$. Then the first term is

$$
\begin{aligned}
(-1)^{|\alpha|-1} \partial &\left[\alpha, \tilde{\alpha} \wedge \Omega_X^{-1}\right]_{\mathrm{SN}} \\
&= (-1)^{|\alpha|-1}(-1)^{|\alpha|-1} \partial\left(\partial\left(\alpha \wedge \tilde{\alpha} \wedge \Omega_X^{-1}\right) - \partial \alpha \wedge \tilde{\alpha} \wedge \Omega_X^{-1} - (-1)^{|\alpha|} \alpha \wedge \partial\left(\tilde{\alpha} \wedge \Omega_X^{-1}\right)\right) \\
&= (-1)^{|\alpha|-1} \partial\left(\alpha \wedge \partial\left(\tilde{\alpha} \wedge \Omega_X^{-1}\right)\right) \ = \ (-1)^{|\alpha|-1} \partial\left(\alpha \wedge \iota_{\Omega_X^{-1}} \partial_{\mathrm{dR}} \tilde{\alpha}\right) \\
&= -\partial\left(\iota_{\Omega_X^{-1}}(\alpha \wedge \partial_{\mathrm{dR}} \tilde{\alpha})\right) \ = \ -\iota_{\Omega_X^{-1}}(\partial_{\mathrm{dR}}(\alpha \wedge \partial_{\mathrm{dR}} \tilde{\alpha})) \ = \ -\iota_{\Omega_X^{-1}}(\partial_{\mathrm{dR}} \alpha \wedge \partial_{\mathrm{dR}} \tilde{\alpha}),
\end{aligned}
$$

where $\partial_{\mathrm{dR}}$ is the holomorphic de Rham differential applied to $\Omega^{0,\bullet}(X)$, $\iota_{\Omega_X^{-1}} \colon \Omega(X) \to \mathrm{PV}(X)$ is the contraction with $\Omega_X^{-1}$, and $\wedge$ stands for the wedge product of forms or polyvector fields. The second term is

$$
\begin{aligned}
-\partial &\left[\alpha \wedge \Omega_X^{-1}, \tilde{\alpha}\right]_{\mathrm{SN}} \\
&= -(-1)^{|\alpha \wedge \Omega_X^{-1}|-1} \partial\left(\partial\left(\alpha \wedge \Omega_X^{-1} \wedge \tilde{\alpha}\right) - \partial\left(\alpha \wedge \Omega_X^{-1}\right) \wedge \tilde{\alpha} - (-1)^{|\alpha \wedge \Omega_X^{-1}|} \alpha \wedge \Omega_X^{-1} \wedge \partial \tilde{\alpha}\right) \\
&= (-1)^{|\alpha|} \partial\left(\partial\left(\alpha \wedge \Omega_X^{-1}\right) \wedge \tilde{\alpha}\right) \ = \ (-1)^{|\alpha|} \partial\left(\left(\iota_{\Omega_X^{-1}} \partial_{\mathrm{dR}} \alpha\right) \wedge \tilde{\alpha}\right) \\
&= (-1)^{|\alpha|} \partial\left(\iota_{\Omega_X^{-1}}(\partial_{\mathrm{dR}} \alpha \wedge \tilde{\alpha})\right) \ = \ (-1)^{|\alpha|} \iota_{\Omega_X^{-1}}(\partial_{\mathrm{dR}}(\partial_{\mathrm{dR}} \alpha \wedge \tilde{\alpha})) \ = \ -\iota_{\Omega_X^{-1}}(\partial_{\mathrm{dR}} \alpha \wedge \partial_{\mathrm{dR}} \tilde{\alpha}),
\end{aligned}
$$

where all the degrees are the usual polyvector degrees. Note that the two coincide as desired.

It remains to show that the action of $T$ preserves the DG Lie algebra structure. We prove in a similar way as follows:

- Let $\alpha \in \mathrm{PV}^{0,\bullet}(X)$ and $\mu \in \ker \partial \subset \mathrm{PV}^{1,\bullet}(X)$. Then

$$T([\alpha,\mu]) = T([\alpha,\mu]_{\mathrm{SN}}) = [\alpha,\mu]_{\mathrm{SN}} = [T(\alpha), T(\mu)]_{\mathrm{SN}} = [T(\alpha), T(\mu)].$$

- Let $\alpha \in \mathrm{PV}^{0,\bullet}(X)$ and $\gamma \in \mathrm{PV}^{3,\bullet}(X)$, say $\gamma = \tilde{\alpha} \wedge \Omega_X^{-1}$ for $\tilde{\alpha} \in \mathrm{PV}^{0,\bullet}(X)$.

$$T([\alpha,\gamma]) = [\alpha,\gamma] = [\alpha, \gamma + \tilde{\alpha}] = [T(\alpha), T(\gamma)].$$

- Let $\mu \in \ker \partial \subset \mathrm{PV}^{1,\bullet}(X)$ and $\gamma \in \mathrm{PV}^{3,\bullet}(X)$, say $\gamma = \tilde{\alpha} \wedge \Omega_X^{-1}$. Then note

$$T([\mu, \gamma]) = [\mu, \gamma] + \alpha,$$

where $[\mu, \gamma] = \alpha \wedge \Omega_X^{-1}$. On the other hand, we have

$$[T(\mu), T(\gamma)] = [\mu, \gamma + \tilde{\alpha}] = [\mu, \gamma] + [\mu, \tilde{\alpha}]_{\mathrm{SN}}.$$

Now it remains to show $\alpha = [\mu, \tilde{\alpha}]_{\mathrm{SN}}$, or equivalently, $[\mu, \tilde{\alpha} \wedge \Omega_X^{-1}] = [\mu, \tilde{\alpha}]_{\mathrm{SN}} \wedge \Omega_X^{-1}$. Note

$$
\begin{aligned}
[\mu, \tilde{\alpha} \wedge \Omega_X^{-1}] &= (-1)^{|\mu|-1}[\mu, \tilde{\alpha} \wedge \Omega_X^{-1}]_{\mathrm{SN}} = (-1)^{|\mu|-1}(-1)^{(|\mu|-1)|\tilde{\alpha}|}[\tilde{\alpha} \wedge \Omega_X^{-1}, \mu]_{\mathrm{SN}} \\
&= (-1)^{(|\mu|-1)(|\tilde{\alpha}|-1)}[\tilde{\alpha}, \mu]_{\mathrm{SN}} \wedge \Omega_X^{-1} \\
&= (-1)^{(|\mu|-1)(|\tilde{\alpha}|-1)}(-1)^{(|\mu|-1)(|\tilde{\alpha}|-1)}[\mu, \tilde{\alpha}]_{\mathrm{SN}} \wedge \Omega_X^{-1} \\
&= [\mu, \tilde{\alpha}]_{\mathrm{SN}} \wedge \Omega_X^{-1},
\end{aligned}
$$

where we again used the fact that $[-, \mu]_{\mathrm{SN}}$ intertwines the identification

$$(-) \wedge \Omega_X^{-1}: \quad \mathrm{PV}^{0,\bullet}(X) \cong \mathrm{PV}^{3,\bullet}(X).$$

$\square$

**Remark 2.21.** Note that we may in fact define an action of all of $\mathrm{SL}(2, \mathbb{C})$ – under the identification of minimal BCOV theory with potentials with a twist of type IIB supergravity established in the next section, this is a complexification of the $\mathrm{SL}(2, \mathbb{R})$ duality group of perturbative type IIB supergravity. The fields acted on nontrivially are describing components of the B-field and Ramond–Ramond 2-form of type IIB that survive the twist. Nonperturbatively, this is broken to $\mathrm{SL}(2, \mathbb{Z})$, due to charge quantization effects. A natural way to incorporate these effects would be to replace those fields, with a suitable holomorphic Deligne complex; see [41] for a related discussion. We hope to return to these points elsewhere.

## 3  Twisted supergravity

Having introduced topological string theory, we wish to argue that it indeed describes protected sectors of the physical superstring theory. Ideally, there would be a twisting procedure that takes in the physical superstring theory as an input and produces the topological string theory we work with as an output. Unfortunately, a rigorous treatment of physical string theory and a mathematical codification of such a procedure is currently out of reach.

On the other hand, Costello and Li [10] define a class of so-called *twisted supergravity backgrounds* that conjecturally have the feature that the fields of supergravity in perturbation theory around such a background map to the closed string field theory of a topological string theory.

Following the work of Costello and Li, we explain the twisting procedure for supergravity in Subsection 3.1 and moreover provide operational descriptions of twists of supergravity in Subsection 3.2. In particular, closed string field theory of topological string theory contains twisted supergravity as discussed in more detail in the latter.

Although we explain several points in a way that has not appeared in the literature before, much of the idea was already present in [10]; the reader may find the original paper a useful supplement.

## 3.1 Constructing twisted supergravity

One of the main utilities of string theory is that the low-energy dynamics of closed strings describe the dynamics of gravitons. Analogously, topological closed string field theory contains a version of gravity that governs a particular subclass of metric deformations present in physical gravity. For those topological string theories that conjecturally describe twists of superstrings, this version of gravity is known as twisted supergravity.

In this subsection, we review the construction of twisted supergravity following [10], with the goal of motivating the conjectural definitions and discussions in the next subsection. We also briefly describe the relationship between the twisting of supersymmetric field theories and supergravity, touching on the role that twisting homomorphisms for the former play in the latter context. This subsection is largely independent of the rest of the paper, and readers who are interested primarily in mathematical applications of our work may wish to skip ahead to the next subsection.

### 3.1.1 Type IIB supergravity

The construction of twisted supergravity uses a description of type II supergravity in the BV-BRST formalism. Such a description of the full theory is both unwieldy and excessive; here we will give a partial description of the theory that includes the relevant ingredients for describing the twisting procedure. For concreteness, we will discuss the construction in the setting of type IIB supergravity on $\mathbb{R}^{10}$ – the construction for type IIA supergravity is completely analogous and the generalization to an arbitrary 10-manifold is straightforward.

We will work with supergravity in the first-order formalism. Roughly speaking, one may think of the theory as a gauge theory for a 10-dimensional supersymmetry algebra. Let us begin by recalling the definition of the relevant supersymmetry algebra.

We first fix some notation. Note that the Hodge star operator acting on $\Omega^5(\mathbb{R}^{10}) = \Omega^5(\mathbb{R}^{10}; \mathbb{C})$ squares to $-1$; we will use a subscript of $\pm$ to denote the $\pm i$-eigenspace of this action. Additionally, we decorate a space of forms with the subscript cc to denote the space of such forms with constant coefficients. Recall that the Lie algebra $\mathfrak{so}(10, \mathbb{C})$ has two irreducible spin representations $S_\pm$, each of complex dimension 16. Furthermore, we have an isomorphism of $\mathfrak{so}(10, \mathbb{C})$-representations $\mathrm{Sym}^2 S_+ \cong \mathbb{C}^{10} \oplus \Omega^5_{+,\mathrm{cc}}(\mathbb{R}^{10})$; let $\Gamma^+ \colon \mathrm{Sym}^2 S_+ \to \mathbb{C}^{10}$ denote the projection.

**Definition 3.1.** The *10-dimensional $\mathcal{N} = (2,0)$ super-translation algebra* is the Lie superalgebra with underlying $\mathbb{Z}/2$-graded vector space

$$\mathcal{T}^{(2,0)} := \mathbb{C}^{10} \oplus \Pi(S_+ \otimes \mathbb{C}^2),$$

and bracket given as follows. Choose an inner product $\langle -, - \rangle$ on $\mathbb{C}^2$ and let $\{e_1, e_2\}$ denote an orthonormal basis. The bracket on odd elements is given by

$$[\psi_1 \otimes \alpha, \psi_2 \otimes \beta] = \Gamma^+(\psi_1, \psi_2)\langle \alpha, \beta \rangle.$$

The *10-dimensional $\mathcal{N} = (2,0)$ supersymmetry algebra* is the Lie superalgebra given by the semidirect product

$$\mathfrak{siso}_{\mathrm{IIB}} := \mathfrak{so}(10, \mathbb{C}) \ltimes \mathcal{T}^{(2,0)}.$$

As remarked above, we will work in the first-order formalism, where supergravity is a theory with fundamental field a $\mathfrak{siso}_{\mathrm{IIB}}$-valued connection [42]. The idea of describing gravity in this way may be unfamiliar, so let us first recall how this works in ordinary Einstein gravity. In the first-order formalism for (Euclidean) Einstein gravity on $\mathbb{R}^4$, the fundamental field is $A \in \Omega^1(\mathbb{R}^4) \otimes \mathfrak{siso}(4)$ where $\mathfrak{siso}(4) = \mathfrak{so}(4) \ltimes \mathbb{R}^4$ denotes the Poincaré Lie algebra. Decomposing this into components, we find

- The component $e \in \Omega^1(\mathbb{R}^4) \otimes \mathbb{R}^4$ is the *vielbein* and encodes the metric as $g = (e \otimes e)$ where we used the standard inner product $(-,-)$ on $\mathbb{R}^4$.

- The component $\Omega \in \Omega^1(\mathbb{R}^4) \otimes \mathfrak{so}(4)$ is the *spin connection*.

The action of the theory takes the form $S(e, \Omega) = \int_{\mathbb{R}^4} e \wedge e \wedge F_\Omega$ where $F_\Omega$ denotes the curvature of $\Omega$. Note that it is of first order – hence the name of the formulation.

Returning to the supergravity setting, the fundamental field is a $\mathfrak{siso}_{\text{IIB}}$-valued connection, which we may locally express as a $\mathfrak{siso}_{\text{IIB}}$-valued 1-form and decompose into components. This yields the following fields:

- The component $E \in \Omega^1(\mathbb{R}^{10}) \otimes \mathbb{C}^{10}$ is the vielbein which encodes the metric as above.

- The component $\Omega \in \Omega^1(\mathbb{R}^{10}) \otimes \mathfrak{so}(10, \mathbb{C})$ is the *spin connection*.

- The component $\Psi \in \Omega^1(\mathbb{R}^{10}) \otimes \Pi(S_+ \otimes \mathbb{C}^2)$ is the *gravitino*.

The theory also includes other fields such as the B-field that we have not included here [42].

Note that we have an action of the Lie algebra $C^\infty(\mathbb{R}^{10}, \mathfrak{siso}_{\text{IIB}})$ on the above space of fields. We wish to treat the theory in the BV-BRST formalism, and to do so, we take a homotopy quotient of the space of fields by the action of $C^\infty(\mathbb{R}^{10}, \mathfrak{siso}_{\text{IIB}})$ and then take the $(-1)$-shifted cotangent bundle. The resulting extended space of fields is $\mathbb{Z} \times \mathbb{Z}/2$-graded and contains the following

Table 1: Fields in type IIB supergravity in the BV-BRST formalism.

| | $-1$ | $0$ | $1$ | $2$ |
|---|---|---|---|---|
| even | $\Omega^0(\mathbb{R}^{10}) \otimes \mathbb{C}^{10}$ | $\Omega^1(\mathbb{R}^{10}) \otimes \mathbb{C}^{10}$ | $\Omega^9(\mathbb{R}^{10}) \otimes \mathbb{C}^{10}$ | $\Omega^{10}(\mathbb{R}^{10}) \otimes \mathbb{C}^{10}$ |
| even | $\Omega^0(\mathbb{R}^{10}) \otimes \mathfrak{so}(10, \mathbb{C})$ | $\Omega^1(\mathbb{R}^{10}) \otimes \mathfrak{so}(10, \mathbb{C})$ | $\Omega^9(\mathbb{R}^{10}) \otimes \mathfrak{so}(10, \mathbb{C})$ | $\Omega^{10}(\mathbb{R}^{10}) \otimes \mathfrak{so}(10, \mathbb{C})$ |
| odd | $\Omega^0(\mathbb{R}^{10}) \otimes \Pi S$ | $\Omega^1(\mathbb{R}^{10}) \otimes \Pi S$ | $\Omega^9(\mathbb{R}^{10}) \otimes \Pi S$ | $\Omega^{10}(\mathbb{R}^{10}) \otimes \Pi S$ |

where we have put $S = S_+ \otimes \mathbb{C}^2$, the $\mathbb{Z}$-grading is listed horizontally, and the $\mathbb{Z}/2$-grading vertically. We emphasize that this is a partial description of the theory that just includes fields needed in our construction. More precisely, this partial description can be viewed as a BF-type theory for a super-Cartan connection – this formal moduli problem ought to receive a map from any formal moduli problem describing a generic theory of supergravity in the first order formalism. For some more details on this topic in the BV formalism, we refer the reader to [43, Section 12.2].

We also use the following terminologies:

- The *bosonic ghost* is the field $q \in C^\infty(\mathbb{R}^{10}, \Pi S)$. This field has bidegree $(-1, 1)$ with respect to the $\mathbb{Z} \times \mathbb{Z}/2$-grading and will play a central role in the construction of twisted supergravity in the next subsection.

- The *ghost for diffeomorphisms* is the field $V \in \Omega^0(\mathbb{R}^{10}) \otimes \mathbb{C}^{10}$ and $V^* \in \Omega^{10}(\mathbb{R}^{10}) \otimes \mathbb{C}^{10}$ denotes its antifield.

- The field $\Psi^* \in \Omega^9(\mathbb{R}^{10}) \otimes \Pi S$ is the antifield to the gravitino.

On a curved spacetime $(M^{10}, g)$ where $g$ satisfies the supergravity equations of motion (i.e. Ricci flat), each entry above should be replaced with forms valued in an appropriate bundle on $M$.

**Note 3.2.** There is a nontrivial bracket between the diffeomorphism ghosts. If we replace $\mathbb{R}^{10}$ with a more general manifold, the diffeomorphism ghosts are going to be vector fields. The existence of this nontrivial bracket gives a sense in which first-order gravity is not a gauge theory, at least at face value. On the other hand, we are still going to colloquially refer to the action of $C^\infty(M; \mathfrak{siso}_{\mathrm{IIB}})$ as gauge transformations.

In addition to the bracket mentioned in the remark, the action includes the following terms:

- $\int_{\mathbb{R}^{10}} V^*[q, q]$, where $[-, -]$ denotes the bracket of $\mathfrak{siso}_{\mathrm{IIB}}$ extended $C^\infty(\mathbb{R}^{10})$-linearly.

- $\int_{\mathbb{R}^{10}} \Psi^* \nabla_g q$, where $\nabla_g$ is a metric connection on the trivial spinor bundle.

Note that varying the action functional with respect to $V^*$ yields the equation of motion $[q, q] = 0$. Therefore, it makes sense to take $q$-cohomology. Further, varying with respect to $\Psi^*$ yields the equation of motion $\nabla_g q = 0$ so the bosonic ghost must be covariantly constant. Below we will use the subscript "cov" to refer to being covariantly constant on a possibly curved spacetime $(M, g)$.

### 3.1.2 Twisting supergravity

Let us now describe the construction of twisted supergravity. Afterwards, we describe some analogies with phenomena in supersymmetric and non-supersymmetric gauge theory to help orient the readers.

**Definition 3.3.** *Twisted supergravity* on $M = M^n$ by a supercharge $Q \in \Pi S$ is supergravity in perturbation theory around a solution to the equations of motion where the bosonic ghost $q$ is required to be the covariantly constant scalar $d_Q \in C^\infty_{\mathrm{cov}}(M; \Pi S)[1]$. We say the twist is an *H-invariant twist* if $Q \in \Pi S$ is invariant under $H \subset \mathrm{Spin}(n)$.

That is, twisting supergravity simply amounts to choosing a particular vacuum around which to do perturbation theory. In that regard, it may be helpful to think of the following analogy with choosing a vacuum on the Coulomb branch of a supersymmetric gauge theory:

Table 2: Analogy between supersymmetric gauge theory vacua and supergravity twists.

| supersymmetric gauge theory | supergravity |
|---|---|
| $G$ gauge group | $\mathfrak{g} = \mathfrak{siso}_{\mathrm{IIB}}$ supersymmetry algebra |
| $\phi \in C^\infty(M, \mathfrak{g})$ scalar in vector multiplet | $q \in C^\infty(M; \Pi S)[1]$ bosonic ghost |
| **Coulomb branch** | **twisted supergravity** |
| $\phi_0 \in \mathfrak{g}$ or $\phi_0 \in C^\infty_{\mathrm{flat}}(M, \mathfrak{g})$ | $Q \in \Pi S \subset \mathfrak{siso}_{\mathrm{IIB}}$ or $d_Q \in C^\infty_{\mathrm{cov}}(M; \Pi S)[1]$ |
| ask $\phi = \phi_0$ | ask $q = d_Q$ |
| broken gauge group | broken SUSY algebra |
| $\mathrm{Stab}_G(\phi_0)$ | $\mathrm{Stab}_{\mathfrak{siso}_{\mathrm{IIB}}}(Q) := H^\bullet(\mathfrak{siso}_{\mathrm{IIB}}, Q)$ |

Here the subscript "flat" means that we take flat sections of a connection, on the background $G$-bundle, induced by the metric on $M$.

It is not essential that the gauge theory be supersymmetric to make the above analogy. To clarify this, let us provide a different analogy:

Table 3: Analogy between super group gauge theory vacua and supergravity twists.

| gauge theory with super gauge group | supergravity |
|---|---|
| $G = \mathrm{GL}(N\vert N)$ gauge group $\phi \in C^\infty(M, \Pi\mathfrak{gl}(N))[1]$ bosonic ghost | $\mathfrak{g} = \mathfrak{siso}_{\mathrm{IIB}}$ supersymmetry algebra $q \in C^\infty(M; \Pi S)[1]$ bosonic ghost |
| **twisted gauge theory** $\phi_0 \in \Pi\mathfrak{gl}(N) \subset \mathfrak{gl}(N\vert N)$ or $\phi_0 \in C^\infty_{\mathrm{flat}}(M, \Pi\mathfrak{gl}(N))$ ask $\phi = \phi_0$ | **twisted supergravity** $Q \in \Pi S \subset \mathfrak{siso}_{\mathrm{IIB}}$ or $d_Q \in C^\infty_{\mathrm{cov}}(M; \Pi S)[1]$ ask $q = d_Q$ |
| broken gauge group $\mathrm{Stab}_G(\phi_0)$ | broken SUSY algebra $\mathrm{Stab}_{\mathfrak{siso}_{\mathrm{IIB}}}(Q) := H^\bullet(\mathfrak{siso}_{\mathrm{IIB}}, Q)$ |

Note that one cannot assign a non-zero vacuum expectation value to a fermionic element. However, if we have a fermionic component of an algebra, then its ghost is fermionic in the cohomological grading as well. This gives a bosonic ghost which can admit a non-zero vacuum expectation value so we can ask a field to be at that vacuum.

The upshot is that a twist of supergravity by $d_Q$ has residual supersymmetry action of $H^\bullet(\mathfrak{siso}_{\mathrm{IIB}}, Q)$. Moreover, by construction of supergravity theory, $H^\bullet(\mathfrak{siso}_{\mathrm{IIB}}, Q)$ should arise as fields of twisted supergravity. We intend to revisit this idea in future work.

### 3.1.3 Gravitational backgrounds from twisting homomorphisms

In this subsection we wish to relate twisted supergravity with the familiar procedure for twisting supersymmetric field theories. The main claim is that from a twisting homomorphism of a supersymmetric field theory, one can construct a twisted supergravity background such that coupling the supersymmetric field theory to the given background has the effect of twisting the supersymmetric field theory. That a world-volume theory of D-branes on a curved (non-twisted) supergravity background should be twisted was already known [44].

Let us begin by outlining the general procedure. Given a square zero supercharge $Q$ for a supersymmetric field theory in dimension $n$, let us consider the stabilizer subgroup $G(Q) := \mathrm{Stab}_{\mathrm{Spin}(n) \times G_R}(Q)$ of $\mathrm{Spin}(n) \times G_R$, that is, the largest subgroup for which $Q$ is invariant. In practice, there exists a subgroup $H \subset \mathrm{Spin}(n)$ and a homomorphism $\rho : H \to G_R$ such that its graph is the group $G(Q)$. In this case, we call $\rho$ a twisting homomorphism. The theory obtained by twisting with $Q$ can then be defined on manifolds whose structure group is contained in $H$.

Now, let us restrict to the setting of world-volume theories of D-branes on type IIB string theory, namely, those with maximal supersymmetry. In this case, it is known that as we consider $D_{2k-1}$-brane on $M^n = M^{2k}$, the R-symmetry group is $G_R = \mathrm{Spin}(10-2k)$. Suppose the structure group of $M$ is contained in $H$ and let $V$ denote the vector representation of the R-symmetry group $G_R$. We construct a supergravity background with the data of $(M, \rho, V)$ as follows. Fixing a spin structure on $M$ and using the assumption that the structure group of $M$ is $H$, let $P$ denote the $H$-reduction of the spin frame bundle of $M$. The claim is that the desired supergravity background is the 10-manifold $X = \mathrm{Tot}(P \times_H V)$, the total space of the $V$-bundle associated to $P$ via $\rho$, together with a Calabi–Yau metric and a covariantly constant spinor. The existence of covariantly constant spinors is guaranteed by the Calabi–Yau structure if it exists, but as far as we are aware, it must be shown on a case-by-case basis that such a 10-manifold is Calabi–Yau. Finally, $X$ should be completed along $M$ to yield the background $X^\wedge$, as D-branes

do not know the global geometry.

We now carry out this procedure for the case of the Kapustin (or holomorphic-topological) twist of 4-dimensional $\mathcal{N} = 4$ gauge theory to identify the supergravity background. The twist will be invariant under a subgroup $\mathrm{Spin}(2) \times \mathrm{Spin}(2) \hookrightarrow \mathrm{Spin}(4)$; we identify this subgroup as $H = U(1) \times U(1)$ and think of a spin representation of $U(1)$ to be of weight $\frac{1}{2}$. Then the Kapustin twisting homomorphism is given by

$$
\rho\colon\ U(1) \times U(1) \to SU(4)\,, \qquad (\lambda, \mu) \mapsto \begin{pmatrix} \lambda^{1/2}\mu^{1/2} & & & \\ & \lambda^{-1/2}\mu^{1/2} & & \\ & & \lambda^{1/2}\mu^{-1/2} & \\ & & & \lambda^{-1/2}\mu^{-1/2} \end{pmatrix}.
$$

In the notation of our general procedure above, we have $H = U(1) \times U(1)$ so our twisted theory can be formulated on $M = \Sigma \times C$ where the $\Sigma$ and $C$ are Riemann surfaces. Here, $V$ is the vector representation of $\mathrm{Spin}(6)$. Note that its complexification $V_{\mathbb{C}}$, under the isomorphism $\mathrm{Spin}(6, \mathbb{C}) \cong SL(4, \mathbb{C})$, is isomorphic to $\wedge^2 \mathbb{C}^4$ where $\mathbb{C}^4$ is the fundamental representation of $SL(4, \mathbb{C})$. To determine $X$ let us first decompose $V_{\mathbb{C}}$ as a representation of $\mathrm{im}\,\rho$. By the functoriality of restriction, we have that

$$
\begin{aligned}
\mathrm{Res}_{SL(4,\mathbb{C})}^{\mathrm{im}\,\rho} \wedge^2 \mathbb{C}^4 &\cong \wedge^2 \mathrm{Res}_{SL(4,\mathbb{C})}^{\mathrm{im}\,\rho} \mathbb{C}^4 \\
&\cong \wedge^2 \left( \mathbb{C}_{(\frac{1}{2},\frac{1}{2})} \oplus \mathbb{C}_{(-\frac{1}{2},\frac{1}{2})} \oplus \mathbb{C}_{(\frac{1}{2},-\frac{1}{2})} \oplus \mathbb{C}_{(-\frac{1}{2},-\frac{1}{2})} \right) \\
&\cong \mathbb{C}_{(0,1)} \oplus \mathbb{C}_{(1,0)} \oplus \mathbb{C}_{(0,0)} \oplus \mathbb{C}_{(0,0)} \oplus \mathbb{C}_{(-1,0)} \oplus \mathbb{C}_{(0,-1)}\,.
\end{aligned}
$$

The representation $\wedge^2 \mathbb{C}^4$ admits a real structure given by the Hodge star operator $\star$ which is a complex conjugate linear map satisfying $\star^2 = \mathrm{Id}$ on $\wedge^2 \mathbb{C}^4$. If we write a complex basis of $\mathbb{C}^4$ as $e_1, e_2, e_3, e_4$, then the six real basis elements are given by

$$
\begin{cases} e_1 \wedge e_2 + e_3 \wedge e_4\,, \\ \sqrt{-1}(e_1 \wedge e_2 - e_3 \wedge e_4)\,, \end{cases} \qquad \begin{cases} e_1 \wedge e_3 + e_4 \wedge e_2\,, \\ \sqrt{-1}(e_1 \wedge e_3 - e_4 \wedge e_2)\,, \end{cases} \qquad \begin{cases} e_1 \wedge e_4 + e_2 \wedge e_3\,, \\ \sqrt{-1}(e_1 \wedge e_4 - e_2 \wedge e_3)\,. \end{cases}
$$

Now from the $U(1)$-weights of $e_i$, one can see that the left two pairs of them make vector representations while the third pair is the trivial representation. Choosing a spin structure on each of $\Sigma$ and $C$ and letting $P$ denote $K_{\Sigma}^{1/2} \oplus K_C^{1/2}$, we have that $X = T^*(\Sigma \times C) \times \mathbb{C}$. Let $X^\wedge$ denote $T^*_{\mathrm{form}}(\Sigma \times C) \times \mathbb{C}$ where the subscript is used to denote the formal neighborhood of the zero section. Since $\Sigma \times C$ is Kähler, a result of Kaledin [45] gives that $T^*_{\mathrm{form}}(\Sigma \times C)$ is hyperkähler, and hence Calabi–Yau.

Then $X^\wedge$ evidently is Calabi–Yau as well. Thus, we have constructed the desired gravitational background. As a consistency check, one observes that sections of the normal bundle of $\Sigma \times C$ in $X$ have spins that agree with those of the six adjoint scalars of 4-dimensional $\mathcal{N} = 4$ gauge theory after performing the Kapustin twist, that is, four 1-forms and two scalars.

Finally, let us discuss the effect of twisting datum, which is used to equip the twisted supersymmetric field theory with a $\mathbb{Z}$-grading. The twisting datum of interest is $\alpha\colon U(1) \to SU(4)$ given by $\lambda \mapsto \mathrm{diag}(\lambda, \lambda^{-1}, \lambda^{-1}, \lambda)$. Introducing the weight under the image as a superscript, now $\wedge^2 \mathbb{C}^4$ as a representation of $H \times U(1)$ is given by

$$
\begin{aligned}
\wedge^2 \mathbb{C}^4 &\cong \wedge^2 \left( \mathbb{C}_{(\frac{1}{2},\frac{1}{2})}^{(1)} \oplus \mathbb{C}_{(-\frac{1}{2},\frac{1}{2})}^{(-1)} \oplus \mathbb{C}_{(\frac{1}{2},-\frac{1}{2})}^{(-1)} \oplus \mathbb{C}_{(-\frac{1}{2},-\frac{1}{2})}^{(1)} \right) \\
&\cong \mathbb{C}_{(0,1)}^{(0)} \oplus \mathbb{C}_{(1,0)}^{(0)} \oplus \mathbb{C}_{(0,0)}^{(2)} \oplus \mathbb{C}_{(0,0)}^{(-2)} \oplus \mathbb{C}_{(-1,0)}^{(0)} \oplus \mathbb{C}_{(0,-1)}^{(0)}\,.
\end{aligned}
$$

This suggests that the cotangent direction of $T^*(\Sigma \times C)$ doesn't acquire a nontrivial cohomological degree but the constant direction does. In sum, the supergravity background which

yields Kapustin twist of 4d $\mathcal{N} = 4$ gauge theory on D3 branes as a $\mathbb{Z}$-graded theory has to be $X = T^*(\Sigma \times C) \times \mathbb{C}[2]$. The nontrivial cohomological shift in $\mathbb{C}[2]$ is essential for its application to geometric Langlands theory, specifically to capture the singular support condition (see [15]).

## 3.2 Describing twisted supergravity

### 3.2.1 Definition of closed string field theory and twisted supergravity

In the previous subsection, we discussed how one constructs twisted supergravity in some generality with the focus on twists of type IIB theory. However, given the complexity of 10-dimensional supergravity theories, working out a description of the twisted theory from first principles should be regarded as a difficult research question in and of itself. Instead, we take a conjectural description of Costello and Li [10] as our starting point and make the following definitions of twisted closed string field theory and twisted supergravity.

**Definition 3.4.**

- The closed string field theory for the SU(5)-invariant twist of type IIB superstring theory on a Calabi–Yau 5-fold $X^5$ is $\mathrm{IIB_{cl}}[X_B^5] := \mathcal{E}_{\mathrm{BCOV}}(X^5)$.

- The closed string field theory for the SU(3)-invariant twist of type IIB superstring theory on $T^*N \times X^3$, where $N$ is a 2-manifold and $X^3$ is a Calabi–Yau 3-fold, is $\mathrm{IIB_{cl}}[(T^*N)_A \times X_B^3] := C_\bullet(LN)^{S^1} \otimes_{H^\bullet(BS^1)} \mathcal{E}_{\mathrm{BCOV}}(X^3)$. Here $LN$ is the free loop space of $N$ (see Note 2.7).

- The closed string field theory for the SU(4)-invariant twist of type IIA superstring theory on $T^*N \times X^4$, where $N$ is a 1-manifold and $X^4$ is a Calabi–Yau 4-fold, is $\mathrm{IIA_{cl}}[(T^*N)_A \times X_B^4] := C_\bullet(LN)^{S^1} \otimes_{H^\bullet(BS^1)} \mathcal{E}_{\mathrm{BCOV}}(X^4)$.

- The closed string field theory for the SU(2)-invariant twist of type IIA superstring theory on $T^*N \times X^2$, where $N$ is a 3-manifold and $X^2$ is a Calabi–Yau surface, is $\mathrm{IIB_{cl}}[(T^*N)_A \times X_B^2] := C_\bullet(LN)^{S^1} \otimes_{H^\bullet(BS^1)} \mathcal{E}_{\mathrm{BCOV}}(X^2)$.

**Note 3.5.** The twist of type II superstring theory on a 10-manifold $(T^*N)_A \times X_B$ gives $C_\bullet(LN)^{S^1} \otimes_{H^\bullet(BS^1)} \mathcal{E}_{\mathrm{BCOV}}(X)$ as long as $N$ is a symplectic manifold and $X$ is a Calabi–Yau manifold, but it does depend on whether it is of type IIA or IIB to see which backgrounds are allowed.

**Remark 3.6.**

- These definitions when the spacetime manifolds are flat are provided as a conjectural description of a twist of string theory in [10]. At the moment, it seems impossible to make a precise mathematical definition of string theory even in that generality and hence the conjecture isn't mathematically posed. Therefore, we decide to take a generalization of their conjectures as definitions and hence the starting point of our mathematical discussion.

- On the other hand, one may consider the closed string field theory for an even more general background. As explained in Subsection 2.1.3, nominally, one would hope to recover these as $\mathrm{Cyc}^\bullet(\mathcal{C})[2]$ for a topological string theory described by the Calabi–Yau category $\mathcal{C}$, that is, the cyclic $L_\infty$-algebra structure on $\mathrm{Cyc}^\bullet(\mathcal{C})[1]$ from the Calabi–Yau structure together with a shifted Poisson structure. Indeed, the above definitions are all models for $\mathrm{Cyc}^\bullet(\mathrm{Fuk}_W(T^*N) \otimes \mathrm{Coh}(X))$. On the other hand, on a space not of the form

$M = T^*N$, there is no local model for the A-model that captures the non-perturbative effects. Because we don't need such a general situation in the following discussion, we are content with the above definition.

Moreover, supergravity is supposed to be a theory of low-energy limit of closed string field theory where we don't see non-propagating fields in the B-model and non-perturbative information in the A-model. This suggests the following definitions which were also stated as conjectures in [10] (modulo our modification of using $\widetilde{\mathcal{E}}_{\mathrm{mBCOV}}$ instead of $\mathcal{E}_{\mathrm{mBCOV}}$). Note that we discussed in Section 3.1 the constructions underlying these conjectural definitions.

**Definition 3.7.**

- Twists of type IIB supergravity on $M_A \times X_B$ for a symplectic $(8-4n)$-manifold $M$ and a Calabi–Yau $(2n+1)$-fold $X$ are of the form $\mathrm{IIB}_{\mathrm{SUGRA}}[M_A \times X_B] := (\Omega^\bullet(M), d) \otimes \widetilde{\mathcal{E}}_{\mathrm{mBCOV}}(X)$.

- Twists of type IIA supergravity on $M_A \times X_B$ for a symplectic $(10-4n)$-manifold $M$ and a Calabi–Yau $2n$-fold $X$ are of the form $\mathrm{IIA}_{\mathrm{SUGRA}}[M_A \times X_B] := (\Omega^\bullet(M), d) \otimes \widetilde{\mathcal{E}}_{\mathrm{mBCOV}}(X)$.

We will sometimes refer to the twist of IIB on $X_B^5$ as *minimal twists*.

**Remark 3.8.** As the second version of the present paper was being written, several checks of the above "conjectures" have been performed. We summarize them here. First, the free limit of the SU(5) and SU(4) twists of type IIB and IIA supergravity on flat space have been computed using pure spinor superfield techniques in [46]. The answers there-in differ from our conjectural definitions by copies of the constant sheaf on the holomorphic directions. The interactions of the SU(4) twist of type IIA can also be computed via dimensional reduction of the minimal twist of eleven-dimensional supergravity [47] – see Remark 4.6 for more. Taking into account these extra copies of the constant sheaf, one sees that the interacting theory has a description where $\widetilde{\mathcal{E}}_{\mathrm{mBCOV}}$ is replaced with a different potential theory. Such potential theories are studied in generality in a paper by the authors [48].

**Remark 3.9.** We anticipate that for a given twist of type IIA/B supergravity, there should be a map of field theories to the closed string field theory for the corresponding twist of type IIA/B superstring theory. This map is clear in cases where the A-model directions are flat – it is induced from the natural map $\widetilde{\mathcal{E}}_{\mathrm{mBCOV}}(X) \to \mathcal{E}_{\mathrm{BCOV}}(X)$. However, in cases where closed string field theory contains non-flat A-model directions, defining this map seems more subtle and contingent on, for example, a de Rham model for the $S^1$ equivariant string topology of $N$. In Subsection 4.2.2 below, we will describe such a map in the case where $N = \mathbb{R}^2 \times S^1$, after restricting to zero modes along $S^1$. This particular case will be enough for our current purposes, and we leave the general case for future work.

**Remark 3.10.** From the above conjectural descriptions, the main difference between twists of closed string field theory and supergravity theory is exactly given by background fields of the BCOV theory. Then it is a natural question to ask how to interpret those backgrounds fields. Partly motivated by this question, in a paper of the second author with W. He, S. Li, and X. Tang [49], it is suggested that those background fields should be understood as symmetry algebra in the BV framework. In some special case, the corresponding current observables turn out to yield infinitely many mutually commuting Hamiltonians of a dispersionless integrable hierarchy.

From now on, we consider the $\mathrm{SL}(2, \mathbb{Z})$ action on $\widetilde{\mathcal{E}}_{\mathrm{mBCOV}}(X)$ that is extended to an $\mathrm{SL}(2, \mathbb{Z})$ action on $\mathrm{IIB}_{\mathrm{SUGRA}}[M_A \times X_B]$ in a way that is linear over $\Omega^\bullet(M)$. We denote the endomorphisms corresponding to the generators $S, T \in \mathrm{SL}(2, \mathbb{Z})$ by $\mathbb{S}, \mathbb{T}$ respectively, and refer to those endomorphisms as *twisted S-duality*.

### 3.2.2 Comparison with physical supergravity theory

Aspects of the above construction may be evocative of a familiar feature from other theories with (higher) gauge fields such as Maxwell theory or physical type II supergravity theories. In such theories, one may consider charged objects as we have done here; such objects source a flux, or field strength. In this subsection, which is independent of other parts of the paper, we explain the analogy with physical supergravity theory and in particular introduce RR forms and RR field strengths in the context of twisted supergravity.

The analogy is summarized in the following table:

Table 4: Comparison of charges in Maxwell theory, type IIB supergravity, and twisted IIB supergravity.

| | Maxwell theory on $M^4$ | type IIB supergravity on $X^{10}$ | $\text{IIB}_{\text{SUGRA}}[\mathbb{C}_B^5]$ |
|---|---|---|---|
| extended objects on worldvolume | particle of charge $q$ $W \subset M^4$ | $N$ $D_{2k-1}$-branes $Y \subset X^{10}$ | $N$ $D_{2k-1}$-branes $\mathbb{C}^k \subset \mathbb{C}^5$ |
| charged under | gauge field $A \in \Omega^1(M^4)$ | RR $2k$-form $C^{(2k)} \in \Omega^{2k}(X^{10})$ | RR $2k$-form $\mathcal{C}^{(2k)} \in \Omega^{k,k}(\mathbb{C}^5)$ where $\mathcal{C}^{(2k)} := (\partial^{-1}\mu^{4-k,k}) \vee \Omega$ for $\mu^{4-k,k} \in \text{PV}^{4-k,k}(\mathbb{C}^5)$ |
| flux or field strength | $F = dA \in \Omega^2(M^4)$ | $G^{(2k+1)} = dC^{(2k)} \in \Omega^{2k+1}$ | $d\mathcal{C}^{(2k)} \in \Omega^{2k+1}$ $= (\text{Id} + \bar{\partial}\,\partial^{-1})\mu^{4-k,k} \vee \Omega$ |
| duality | $F \longleftrightarrow *F$ | $G^{(10-2k-1)} = *G^{(2k+1)}$ | "$\mathcal{G}^{(10-2k-1)} = *\mathcal{G}^{(2k+1)}$" |
| choice of potentials | | if $G^{(2k+1)} = dC^{(2k)}$ $G^{(10-2k-1)} := *G^{(2k+1)}$ | if $\mathcal{G}^{(2k+1)} = \bar{\partial}\partial^{-1}\mu^{4-k,k} \vee \Omega$ $\mathcal{G}^{(10-2k-1)} = \mu^{k,4-k} \vee \Omega$ |
| modified action | $\int_{M^4} F \wedge *F$ $+q\int_W d^{-1}F + \cdots$ | $\int_{X^{10}} G^{(2k+1)} \wedge G^{(10-2k-1)}$ $+N\int_Y d^{-1}G^{(2k+1)} + \cdots$ | $\int_{\mathbb{C}^5} \mathcal{G}^{(2k+1)} \wedge \mathcal{G}^{(10-2k-1)}$ $+N\int_{\mathbb{C}^k} d^{-1}\mathcal{G}^{(2k+1)} + \cdots$ |
| equation for sourced flux or field strength | $d*F = q\delta_W$ | $dG^{(10-2k-1)} = N\delta_Y$ | $\bar{\partial}\mu^{k,4-k} \vee \Omega_{\mathbb{C}^5} = N\delta_{\mathbb{C}^k}$ |

Note that unlike Maxwell theory where we treat $A$ as a fundamental field and its field strength $F$ as a derived notion, in the supergravity setting we regard the field strength itself as an element of the space of fields and call it the flux sourced by the brane. This may seem unusual, but is in accordance with the democractic formulation of string theory where the Ramond–Ramond (RR) field strengths $G^{(2k+1)}$ satisfying the constraint $G^{(2k+1)} = *G^{(10-2k-1)}$ are taken to be fundamental fields and one is free to choose potentials for exactly half of them [50]. That is, when we use $G^{(2k+1)} = dC^{(2k)}$, the RR field strength $G^{(10-2k-1)}$ doesn't admit a description in terms of $C^{(10-2k-2)}$, but it is defined as $G^{(10-2k-1)} := *G^{(2k+1)}$. Physicists sometimes do not a priori make any choice and say that RR $2k$-form $C^{(2k)}$ is electro-magnetic dual to RR $(10-2k-2)$-form $C^{(10-2k-2)}$. In particular, the RR 5-form field strength $G^{(5)}$ is self-dual – this constraint must be dealt with carefully in discussing charge quantization. Once a choice of RR $2k$-form $C^{(2k)}$ is made one can add the term $\int_{Y^{2k}} C^{(2k)}$ to the action functional.

It is then colloquial to say that the $D_{2k-1}$-brane is electrically charged under RR $2k$-form and and magnetically charged under the RR $(10-2k-2)$-form though for $k \neq 2$ only the electric field is part of the space of fields. In particular, we see that D3 branes are both electrically and magnetically charged under the RR 4-form.

In our twisted supergravity setting, in order for the democratic formulation to be respected, one should make a definition of $\mathcal{G}^{(2k+1)}$ and $\mathcal{G}^{(10-2k-1)}$ simultaneously when we choose a potential:

**Definition 3.11.** For $k \neq 2$, we define a pair of *Ramond–Ramond field strengths* $\mathcal{G}^{(2k+1)}$ and $\mathcal{G}^{(10-2k-1)}$

(1) *with the choice of a Ramond–Ramond $2k$-form* $\mathcal{C}^{(2k)} = \partial^{-1}\mu^{4-k,k} \vee \Omega_{\mathbb{C}^5}$ *as a potential*

$$\mathcal{G}^{(2k+1)} = \bar{\partial}\,\partial^{-1}\mu^{4-k,k} \vee \Omega_{\mathbb{C}^5}, \quad \text{and} \quad \mathcal{G}^{(10-2k-1)} = \mu^{k,4-k} \vee \Omega_{\mathbb{C}^5},$$

(2) *with the choice of a Ramond–Ramond $(10-2k-2)$-form as a potential* to be

$$\mathcal{G}^{(2k+1)} = \mu^{4-k,k} \vee \Omega_{\mathbb{C}^5}, \quad \text{and} \quad \mathcal{G}^{(10-2k-1)} = \bar{\partial}\,\partial^{-1}\mu^{k,4-k} \vee \Omega_{\mathbb{C}^5}.$$

Note that with this definition and interpretation, the free part of the action of BCOV theory, $\int_X (\mu\,\bar{\partial}\,\partial^{-1}\mu \vee \Omega) \wedge \Omega$, is precisely of the form as expected from type IIB supergravity on $X^{10}$.

**Remark 3.12.** The flux or field strength can be integrated on a sphere linking the support of the charged object to yield the charge. Mathematically, the charge is often a characteristic class for the bundle carrying the gauge field, and charge quantization amounts to an integrality condition on the characteristic class [51].

# 4 Twisted S-duality for $\text{IIB}_{\text{SUGRA}}[M_A^4 \times X_B^3]$

In Definition 2.19, we have described an action of $\text{SL}(2,\mathbb{Z})$ on minimal BCOV theory $\widetilde{\mathcal{E}}_{\text{mBCOV}}(X^3)$ with potentials in dimension 3. This extends by the identity to an action on $\text{IIB}_{\text{SUGRA}}[M_A^4 \times X_B^3]$ – we call this $\text{SL}(2,\mathbb{Z})$ action *twisted S-duality*. The goal of this section is to justify why this action is deserving of such a name.

As described in the introduction, one way physicists think of S-duality of type IIB string theory comes from the fact that type IIB on a circle is equivalent to M-theory on a torus. Via this equivalence, S-duality is realized as a spacetime symmetry – it comes from the action of the mapping class group of a torus. This situation is neatly summarized in Figure 1.

In this section, we establish a comparison between M-theory on a torus and type IIB on a circle that holds at the level of twists. Explicitly, in Subsection 4.1, we introduce the maximal twist of eleven-dimensional supergravity. This is a holomorphic-topological theory that can be defined on products of the form $M \times X$ where $M$ is a $G_2$ manifold and $X$ is a hyperkähler surface. Combined with T-duality map constructed in Subsection 4.2, we then have a map from this theory placed on $\mathbb{R}^5 \times T^2 \times X$ to $\text{IIB}_{\text{SUGRA}}[\mathbb{R}_A^4 \times (\mathbb{C}^\times \times X)_B]$. The construction of this map is a direct implementation of the physical equivalence that holds at the level of twists – it is defined to be a composition of a map gotten from dimensional reduction and T-duality. Note that the source and target of this map carry actions of $\text{SL}(2,\mathbb{Z})$ by the usual mapping class group and twisted S-duality respectively. We show in Subsection 4.3 that this map is equivariant.

## 4.1 A $G_2 \times \text{SU}(2)$-invariant twist of 11d supergravity

The main claim of this subsection is that there is a shadow of the usual relation between 11-dimensional supergravity and type IIA supergravity even after a twist.

We learned the following definition of a twist of 11-dimensional supergravity theory from Costello in the context of his paper on the subject [52].

**Definition 4.1.** The $G_2 \times \mathrm{SU}(2)$-*invariant twist of 11-dimensional supergravity* on $M \times X$, where $M$ is a $G_2$-manifold and $X$ is a hyperkähler surface, is a BV theory described as follows: the space of fields with its shifted $L_\infty$-algebra structure is

$$11\mathrm{d}[M_A \times X_B] := (\Omega^\bullet(M) \otimes \Omega^{0,\bullet}(X)[1]; \ \ell_1 = d_M \otimes 1 + 1 \otimes \bar\partial_X, \ \ell_2 = \{-,-\}, \ \ell_n = 0 \text{ for } n \geq 3),$$

where $\{-,-\}$ denotes the Poisson bracket with respect to the holomorphic volume form $\Omega_X$ on $X$ canonically extended to the entire space, together with an odd invariant pairing given by wedging and integrating against the holomorphic volume form on $X$.

**Remark 4.2.** This definition was initially proposed as a conjectural description of a twist of 11d supergravity in [52]. Since writing the first version of this paper, several checks of this conjecture have ben performed. The free limit has been derived from physical eleven-dimensional supergravity in [53], and the arguments there show that fields in the twist describe certain components of the C-field and gravitino in the physical theory. The interactions have been derived from a minimal twist of 11d supergravity studied by the first author in [47]; this argument together with the method for twisting the interacting physical theory introduced in [54] amount to a proof of Costello's conjecture. Moreover, an $\Omega$-deformed version of this theory has appeared in several holographic discussions [52, 55–65].

The following lemma plays a crucial role in establishing our claim, and is essentially proved in [52, Section 13]. For the statement, let us introduce 5-dimensional BV theory on $S^1 \times X$ where $X$ is a Calabi–Yau 2-fold, given by a truncation of the 11-dimensional supergravity theory. That is, the space of fields is $\Omega^\bullet(S^1) \otimes \Omega^{0,\bullet}(X)[1]$ and the other structures are defined in the same way as before. By the formality of $S^1$, we think of its reduction to 4-dimensional theory where the space of fields is $\mathcal{E}_{5\mathrm{d}\to 4\mathrm{d}}(X) = \Omega^{0,\bullet}(X)[\varepsilon][1]$ from the identification $H^\bullet(S^1) \cong \mathbb{C}[\varepsilon]$. Then we have the action functional of the 4-dimensional theory given by

$$S_{5\mathrm{d}\to 4\mathrm{d}}(A, B) = \int (B\bar\partial A) \wedge \Omega_X + \frac{1}{2}\int (B\{A, A\}) \wedge \Omega_X, \quad \text{where} \quad B\varepsilon \in \Omega^{0,\bullet}(X)\varepsilon, \ A \in \Omega^{0,\bullet}(X)[1].$$

**Lemma 4.3.** There is an equivalence of $\mathbb{Z}/2$-graded BV theories between the above 4-dimensional theory and 2-dimensional minimal BCOV theory with potential.

*Proof.* Recall that the minimal BCOV theory with potential on $X$ is defined by the space of fields being $\widetilde{\mathcal{E}}_{\mathrm{mBCOV}}(X) = \mathrm{PV}^{0,\bullet}(X)[2] \oplus \mathrm{PV}^{2,\bullet}(X)[1]$ and the action functional is

$$S(\alpha, \beta) = \int (\alpha\bar\partial\beta \vee \Omega_X) \wedge \Omega_X + \frac{1}{2}\int (\alpha\partial\beta\partial\beta \vee \Omega_X) \wedge \Omega_X,$$

where $\alpha \in \mathrm{PV}^{0,\bullet}(X)[2]$, $\beta \in \mathrm{PV}^{2,\bullet}(X)[1]$. We define a quasi-isomorphism of cochain complexes

$$\mathcal{E}_{5\mathrm{d}\to 4\mathrm{d}}(X) \to \widetilde{\mathcal{E}}_{\mathrm{mBCOV}}(X),$$

given by

$$\Omega^{0,\bullet}(X)\varepsilon \to \mathrm{PV}^{0,\bullet}(X), \qquad \Omega^{0,\bullet}(X) \to \mathrm{PV}^{2,\bullet}(X),$$
$$B\varepsilon \mapsto B, \qquad\qquad\qquad A \mapsto A \wedge \Pi_X.$$

Moreover, the equivalence of these two theories immediately follows as one notes

$$\{A, A\} = \Pi_X(\partial A \partial A) = (\partial(A \wedge \Pi_X) \wedge \partial(A \wedge \Pi_X)) \vee \Omega_X.$$

$\square$

The following corollary demonstrates that the usual relationship between 11-dimensional supergravity and type IIA supergravity holds at the level of twisted theories. Let us fix a $G_2$ structure on $M^6 \times S^1$. For instance, $M^6$ may be a Calabi–Yau 3-fold.

**Corollary 4.4.** There is an equivalence of DG Lie algebras

$$\mathrm{red}_M \colon \ 11\mathrm{d}\left[(M^6 \times S^1)_A \times X_B^2\right] \to \mathrm{IIA}_{\mathrm{SUGRA}}\left[M_A^6 \times X_B^2\right].$$

*Proof.* This follows from the above lemma and definition because the map is the identity in the direction of $M^6$.

$\square$

**Remark 4.5.** Note that as defined, the theory $11\mathrm{d}[M_A^7 \times X_B]$ does not appear to depend on a $G_2$ structure on $M^7$. In this remark, we wish to posit a potential modification of the conjectural description of [52]; this modification will include some nonperturbative effects and will depend on the $G_2$ structure.

Consider the spacetime of the form $T^*N \times Y$, where $Y$ is a Calabi–Yau 3-fold, and note that there is a map $\mathrm{IIA}_{\mathrm{SUGRA}}[(T^*N)_A \times Y_B] \to \mathrm{IIA}_{\mathrm{cl}}[(T^*N)_A \times Y_B]$. The target of this map in particular captures certain non-perturbative effects such as string states in the A-model with nontrivial winding modes.

One can ask for an M-theoretic lift of these non-perturbative effects, i.e., an $L_\infty$-algebra $\mathcal{M}[(T^*N \times S^1)_A \times Y_B]$ such that the square below is cocartesian:

$$
\begin{array}{ccc}
\mathcal{M}[(T^*N \times S^1)_A \times Y_B] & \dashrightarrow{\;\cong\;} & \mathrm{IIA}_{\mathrm{cl}}[(T^*N)_A \times Y_B] \\
\uparrow & & \uparrow \\
11\mathrm{d}[(T^*N \times S^1)_A \times Y_B] & \xrightarrow{\;\cong\;} & \mathrm{IIA}_{\mathrm{SUGRA}}[(T^*N)_A \times Y_B]
\end{array}.
$$

Physically, string states with nontrivial winding mode in IIA should correspond to certain configurations of M2 branes in M-theory. BPS configurations for an M2 brane in M-theory on a $G_2$ manifold are afforded by associative 3-folds in the G2. Therefore, we might expect that upon replacing $T^*N \times S^1$ with an arbitrary $G_2$ manifold $M^7$, the description of $\mathcal{M}[M_A^7 \times Y_B]$ ought to involve a deformation of the de Rham complex of $M^7$ as an $\mathbb{E}_3$-algebra, where the deformed product involves counts of associative 3-folds [12]. Comparison with the usual curve counting invariants in the context of topological string theory is summarized in the following table:

Table 5: Comparison of counting invariants in topological string theory and twisted M-theory.

| | (A-model) topological string | twisted M-theory |
|---|---|---|
| spacetime $= \mathbb{R}^4 \times M$ | $M$ : CY 3-fold | $M$ : $G_2$ manifold |
| extended objects | F1 string $\Sigma$ | M2 brane $S$ |
| supersymmetric cycles | $J$-holomorphic curves in $M$ | associative 3-folds in $M$ |
| theory on an extended object | A-model on $\Sigma$ with target $M$ | 3d theory on $S$ with target $M$ |
| states/observables | quantum cohomology QH$(M)$ | M2 cohomology M2$(M)$[3] |
| | counting $J$-holomorphic curves | counting associative 3-folds |
| structure at cochain level | $\mathbb{E}_2$-algebra | $\mathbb{E}_3$-algebra |

Relations between the interactions of 11d supergravity, both twisted and untwisted, and 2-shifted Poisson structures are identified in [54], and relations between the $\Omega$-deformed $G_2 \times \mathrm{SU}(2)$ twist of 11d supergravity and enumerative invariants of $G_2$ manifolds are developed in [66, 67].

**Remark 4.6.** Since the reduction of 11d supergravity on any circle should yield IIA supergravity, it is natural to ask if reducing M-theory on a circle $S^1 \subset X$ yields a twist of IIA supergravity theory and if so which one. For simplicity, let us consider a flat space $11\mathrm{d}[\mathbb{R}_A^7 \times (\mathbb{C} \times \mathbb{C}^\times)_B]$ with $S^1 \subset \mathbb{C}^\times$. To answer this question, we choose $\mathrm{SU}(3) \subset G_2$ with which we write $\mathbb{R}^7 \cong \mathbb{R} \times \mathbb{C}_{\mathrm{CY}}^3$ so that $\mathbb{C}_{\mathrm{CY}}^3$ is thought of having a corresponding Calabi–Yau structure. Then we expect that such a reduction yields $\mathrm{IIA}[\mathbb{R}_A^2 \times (\mathbb{C}_{\mathrm{CY}}^3 \times \mathbb{C})_B]$ where $\mathbb{C}$ is equipped with a linear superpotential.

One way to see this involves viewing the $G_2 \times \mathrm{SU}(2)$ invariant twist as a further deformation of the minimal twist [47]. This minimal twist can be placed on backgrounds of the form $\mathbb{R} \times \mathbb{C}^\times \times \mathbb{C}^4$ and has the feature that dimensional reduction along $S^1 \subset \mathbb{C}^\times$ is indeed the $\mathrm{SU}(4)$ invariant twist of type IIA. The fields of the theory are given by resolutions of holomorphic divergence free vector fields and co-closed 1-forms on $\mathbb{C}^\times \times \mathbb{C}^5$ – the deformation to the $G_2 \times \mathrm{SU}(2)$ twist on $\mathbb{R}^7 \times \mathbb{C}^\times \times \mathbb{C}$ is given by a certain linear coefficient 1-form on $\mathbb{C}^\times \times \mathbb{C}$. Under dimensional reduction, this linear coefficient 1-form is identified with a linear superpotential on $\mathbb{C}$ as expected.

## 4.2 T-duality

In this subsection, we explain a version of the usual T-duality between type IIA and type IIB superstring theories that holds in the protected sectors we have defined.

The idea of T-duality is simple; a string cannot detect a difference between a circle of radius $r$ and a circle of radius $1/r$. Therefore, when one considers a string theory defined on a spacetime manifold with a factor of a circle $S_r^1$ of radius $r$, it may be identified with a seemingly different theory defined on a spacetime manifold with the circle replaced by a circle $S_{1/r}^1$ of radius $1/r$.

The main claim is that a shadow of the physically well-known T-duality between IIA and IIB string theories holds at the level of twisted supergravity. We begin by discussing T-duality for topological string theory.

### 4.2.1 T-duality in topological string theory

We claim that topological string theory on $(M^{2k} \times \mathbb{R} \times S^1)_A \times X_B^{4-k}$ is equivalent to topological string theory on $M_A^{2k} \times (\mathbb{C}^\times \times X^{4-k})_B$. This follows directly from an equivalence between A-model on $\mathbb{R} \times S^1$ and B-model on $\mathbb{C}^\times$ (see [68]):

$$\mathrm{Fuk}_{\mathcal{W}}(T^*S^1) \simeq \mathrm{Coh}(\mathbb{C}^\times).$$

This is the underlying input for T-duality between topological string theories as we need. This equivalence of categories will induce an identification between cyclic cochains. We will use this identification to produce a map between the subcomplexes that we have identified as describing twists of supergravity.

Now we formulate T-duality between the closed string field theories we consider. As mentioned, an equivalence of DG categories induces an identification between Hochschild chains

---

[3]There is a sense in which quantum cohomology is a misnomer because as a deformation of de Rham cohomology $H_{\mathrm{dR}}(M)$ of $M$, it is from a stringy effect. Hence string cohomology (or perhaps F1 cohomology) may have been a better name for quantum cohomology. However, string cohomology now referes to a different construction so we are stuck with established terminology. In view of this remark, we opt to call the corresponding deformation of de Rham cohomology of a $G_2$ manifold M2 cohomology.

that intertwines the natural $S^1$ actions. We summarize this at the level of homology in the following table:

Table 6: Identification of homology from closed T-duality.

|  | A-model on $T^*S^1$ | B-model on $\mathbb{C}^\times$ |
|---|---|---|
| $HH_\bullet$ | $\mathrm{SH}^{-\bullet}(T^*S^1) \cong H_\bullet(LS^1)$ $\cong \mathbb{C}[z,z^{-1}][\varepsilon]$ | $HH_\bullet(\mathbb{C}^\times) \cong \Omega^{-\bullet}(\mathbb{C}^\times)$ $\cong \mathbb{C}[z,z^{-1}][\frac{dz}{z}]$ |
| $\Delta$ | $[S^1]: H_0(LS^1) \to H_1(LS^1)$ $f \mapsto (z\partial_z f)\varepsilon$ | $\partial: \Omega^0(\mathbb{C}^\times) \to \Omega^{-1}(\mathbb{C}^\times)$ $f \mapsto (z\partial_z f) \cdot \frac{dz}{z}$ |

where Hochschild homology of wrapped Fukaya category of the cotangent bundle $T^*S^1$ is identified as symplectic cohomology $\mathrm{SH}^{-\bullet}(T^*S^1)$, or equivalently, homology $H_\bullet(LS^1)$ (see [69]). Here $LS^1 \cong S^1 \times \mathbb{Z}$ is the free loop space where $\mathbb{Z}$ encodes the winding number, $S^1$ encodes the initial position, and we write $\varepsilon = [S^1] \in H_1(S^1)$. Now identifying Hochschild homology and Hochschild cohomology using the Calabi–Yau structure, we obtain an isomorphism between $\mathbb{C}[z,z^{-1}][\varepsilon]$, where we abuse the notation to still write $\varepsilon$ for the odd variable, and $\mathrm{PV}_{\mathrm{hol}}(\mathbb{C}^\times) \cong \mathbb{C}[z,z^{-1}][z\partial_z]$ such that the $S^1$-actions are still preserved.

**Remark 4.7.** T-duality is supposed to exchange momentum and winding modes. Understanding this will guide us in finding a T-duality statement in the twisted supergravity setting below.

In the A-model, those closed string states in $H^\bullet(LS^1) \cong H^\bullet(S^1) \otimes H^\bullet(\mathbb{Z})$ with low energy (or without any non-perturbative effect) can be thought of as where the winding number is zero. This is exactly the summand $H^\bullet(S^1) \cong \mathbb{C}[\varepsilon]$. On the other hand, in the B-model, the $\overline{\partial}$-cohomology of $\mathrm{PV}(\mathbb{C}^\times)$ is $\mathbb{C}[z,z^{-1}] \otimes \mathbb{C}[z\partial_z]$. Here the fields in $\mathrm{PV}^{1,\bullet}(\mathbb{C}^\times)$ do not propagate, so the $\overline{\partial}$ cohomology of the fields in the supergravity approximation (as described by minimal BCOV) is exactly $\mathbb{C}[z,z^{-1}]$. The decomposition of closed string states into momentum and winding modes is summarized in the following table:

Table 7: Exchange of momentum and winding modes.

|  | A-model on $T^*S^1$ | B-model on $\mathbb{C}^\times$ |
|---|---|---|
| $HH^\bullet$ | $\mathbb{C}[z,z^{-1}] \otimes \mathbb{C}[\varepsilon]$ | $\mathrm{PV}_{\mathrm{hol}}(\mathbb{C}^\times) \cong \mathbb{C}[z,z^{-1}] \otimes \mathbb{C}[z\partial_z]$ |
| low-energy | $\mathbb{C}[\varepsilon]$ | $\mathrm{PV}^0_{\mathrm{hol}}(\mathbb{C}^\times) \cong \mathbb{C}[z,z^{-1}]$ |
| winding | $\mathbb{C}[z,z^{-1}]$ | $\mathbb{C}[z\partial_z]$ |

The T-dual image of the low energy fields in the A-model must therefore have constant coefficient in the direction where we apply T-duality. In other words, we cannot find non-constant functions as the T-dual image of fields in the supergravity approximation of the A-model.

### 4.2.2 T-duality in twisted supergravity

We now wish to formulate T-duality between supergravity theories on particular backgrounds, namely, $\mathrm{IIA}_{\mathrm{SUGRA}}[(\mathbb{R}^4 \times T^*S^1)_A \times X_B^2]$ and $\mathrm{IIB}_{\mathrm{SUGRA}}[\mathbb{R}^4_A \times (\mathbb{C}^\times \times X^2)_B]$. Note that this does not immediately follow from our description in the previous subsection involving closed string field theory and its description in terms of BCOV theory – our model for supergravity theories is based on the theories with potentials. Still one would want to find a map **T** fitting in the

following diagram

$$\begin{array}{ccc}
\text{IIA}_{\text{cl}}[(\mathbb{R}^4 \times T^*S^1)_A \times X_B^2] & \xrightarrow{\ \simeq\ } & \text{IIB}_{\text{cl}}[\mathbb{R}_A^4 \times (\mathbb{C}^\times \times X^2)_B] \\
\Psi \Big\uparrow & & \Big\uparrow \Phi \\
\text{IIA}_{\text{SUGRA}}[(\mathbb{R}^4 \times T^*S^1)_A \times X_B^2] & \xrightarrow{\ \mathbf{T}\ } & \text{IIB}_{\text{SUGRA}}[\mathbb{R}_A^4 \times (\mathbb{C}^\times \times X^2)_B]
\end{array} \quad .$$

Here $\Phi$ is a natural map induced from $\widetilde{\mathcal{E}}_{\text{mBCOV}} \to \mathcal{E}_{\text{mBCOV}} \to \mathcal{E}_{\text{BCOV}}$, but as noted in Remark 3.9 $\Psi$ is a bit subtle to define. However, upon restricting to those field configurations that have zero winding and zero momentum around $S^1$, the inclusion is easier to describe. That is, we will define a map

$$\Psi' \colon H^\bullet(\text{IIA}_{\text{SUGRA}}[(\mathbb{R}^4 \times T^*S^1)_A \times X_B^2]; d_{T^*S^1} + t\Delta) \to H^\bullet(\text{IIA}_{\text{cl}}[(\mathbb{R}^4 \times T^*S^1)_A \times X_B^2]; d_{T^*S^1} + t\Delta).$$

The notation is meant to suggest that this is a map that is expected to admit a cochain level lift.

First of all, we note

$$H^\bullet(\text{IIA}_{\text{SUGRA}}[(\mathbb{R}^4 \times T^*S^1)_A \times X_B^2]; d_{T^*S^1} + t\Delta) \simeq H^\bullet(\text{IIA}_{\text{SUGRA}}[(\mathbb{R}^4 \times T^*S^1)_A \times X_B^2]; d_{T^*S^1}),$$

because the spectral sequence immediately degenerates for a degree reason. Then the desired map should be induced by a natural map

$$\Psi' \colon H^\bullet(\Omega^\bullet(T^*S^1) \otimes \widetilde{\mathcal{E}}_{\text{mBCOV}}(X), d_{T^*S^1}) \to H^\bullet(C^\bullet(LS^1)^{S^1} \otimes_{H^\bullet(BS^1)} \mathcal{E}_{\text{BCOV}}(X), d_{T^*S^1} + t\Delta),$$

which we abuse the notation to still denote by $\Psi'$ and is defined as follows. To describe the map, observe that using quasi-isomorphisms $(\Omega^\bullet(T^*S^1), d_{T^*S^1}) \simeq (\Omega^\bullet(S^1), d_{S^1}) \simeq \mathbb{C}[\varepsilon]$, the cohomology $H^\bullet(\Omega^\bullet(T^*S^1) \otimes \widetilde{\mathcal{E}}_{\text{mBCOV}}(X), d_{T^*S^1})$ is given by

$$\begin{array}{ccc}
\underline{-2} & \underline{-1} & \underline{0} \\[4pt]
\text{PV}_X^{0,\bullet} & & \\[8pt]
& \text{PV}_X^{2,\bullet} \oplus \text{PV}_X^{0,\bullet}\,\varepsilon & \\[8pt]
& & \text{PV}_X^{2,\bullet}\,\varepsilon
\end{array} \quad .$$

On the other hand, $H^\bullet(C^\bullet(LS^1)^{S^1} \otimes_{H^\bullet(BS^1)} \mathcal{E}_{\text{BCOV}}(X); d_{T^*S^1} + t\Delta)$, which we know should be isomorphic to $H^\bullet(\text{PV}^{\bullet,\bullet}(\mathbb{C}^\times \times X^2)[\![t]\!], \overline{\partial}_{\mathbb{C}^\times} + t\partial_{\mathbb{C}^\times})$ from the previous subsection, is given by

$$\begin{array}{cccc}
\underline{-2} & \underline{-1} & \underline{0} & \underline{1} \\[4pt]
\text{PV}_X^{0,\bullet}[z^{\pm 1}][\![t]\!] & & & \\[8pt]
& \text{PV}_X^{1,\bullet}[z^{\pm 1}][\![t]\!] \oplus \text{PV}_X^{0,\bullet}[z^{\pm 1}][\![t]\!]\varepsilon & & \\[8pt]
& & \text{PV}_X^{2,\bullet}[z^{\pm 1}][\![t]\!] \oplus \text{PV}_X^{1,\bullet}[z^{\pm 1}][\![t]\!]\varepsilon & \\[8pt]
& & & \text{PV}_X^{2,\bullet}[z^{\pm 1}][\![t]\!]\varepsilon
\end{array} \quad ,$$

where we use the Künneth formula $\text{PV}^{k,\bullet}(X \times Y) \cong \bigoplus_{i+j=k} \text{PV}^{i,\bullet}(X) \otimes \text{PV}^{j,\bullet}(Y)$ and the identification

$$H^\bullet(\text{PV}^{i,\bullet}(\mathbb{C}^\times), \overline{\partial}) \cong \begin{cases} \mathbb{C}[z, z^{-1}], & \text{if } i = 0, \\ \mathbb{C}[z, z^{-1}]z\partial_z, & \text{if } i = 1. \end{cases}$$

Then $\Psi'$ is the natural map defined by

$$
\Psi' = \begin{cases} \mathrm{Id}\,, & \text{on } \mathrm{PV}_X^{0,\bullet} \oplus \mathrm{PV}_X^{0,\bullet}\,\varepsilon\,, \\ \partial\,, & \text{on } \mathrm{PV}_X^{2,\bullet}\,, \\ \partial \otimes 1\,, & \text{on } \mathrm{PV}_X^{2,\bullet}\,\varepsilon\,. \end{cases}
$$

With this in hand, we proceed to describe the following diagram.

$$
\begin{array}{ccc}
H^\bullet(\mathrm{IIA}_{\mathrm{cl}}[(\mathbb{R}^4 \times T^*S^1)_A \times X_B^2]; d_{T^*S^1} + t\Delta) & \xrightarrow{\ \simeq\ } & H^\bullet(\mathrm{IIB}_{\mathrm{cl}}[\mathbb{R}_A^4 \times (\mathbb{C}^\times \times X^2)_B]; \overline{\partial}_{\mathbb{C}^\times} + t\partial_{\mathbb{C}^\times}) \\
\Big\uparrow{\scriptstyle \Psi'} & & \Big\uparrow{\scriptstyle \Phi'} \\
H^\bullet(\mathrm{IIA}_{\mathrm{SUGRA}}[(\mathbb{R}^4 \times T^*S^1)_A \times X_B^2]; d_{T^*S^1}) & \xrightarrow{\ \mathbf{T}\ } & H^\bullet(\mathrm{IIB}_{\mathrm{SUGRA}}[\mathbb{R}_A^4 \times (\mathbb{C}^\times \times X^2)_B]; \overline{\partial}_{\mathbb{C}^\times} + t\partial_{\mathbb{C}^\times})
\end{array}
$$

Figure 2: T-duality diagram.

Such a diagram follows from the following:

**Definition/Proposition 4.8.** There is a map

$$
\mathbf{T}\colon \widetilde{\mathcal{E}}_{\mathrm{mBCOV}}(X)[\varepsilon] \to H^\bullet\left(\widetilde{\mathcal{E}}_{\mathrm{mBCOV}}(\mathbb{C}^\times \times X); \overline{\partial}_{\mathbb{C}^\times} + t\partial_{\mathbb{C}^\times}\right),
$$

of cochain complexes such that the following diagram is commutative

$$
\begin{array}{ccc}
H^\bullet\left(C^\bullet(LS^1)^{S^1} \otimes_{H^\bullet(BS^1)} \mathcal{E}_{\mathrm{BCOV}}(X); d_{T^*S^1}\right) & \xleftarrow{\ \simeq\ } & H^\bullet\left(\mathcal{E}_{\mathrm{BCOV}}(\mathbb{C}^\times \times X); \overline{\partial}_{\mathbb{C}^\times} + t\partial_{\mathbb{C}^\times}\right) \\
\Big\uparrow & & \Big\uparrow{\scriptstyle \Phi'} \\
\widetilde{\mathcal{E}}_{\mathrm{mBCOV}}(X)[\varepsilon] & \dashrightarrow{\ \mathbf{T}\ } & H^\bullet\left(\widetilde{\mathcal{E}}_{\mathrm{mBCOV}}(\mathbb{C}^\times \times X); \overline{\partial}_{\mathbb{C}^\times} + t\partial_{\mathbb{C}^\times}\right)
\end{array}.
$$

Moreover, the induced map on $\partial$-cohomology is a map of DG Lie algebras.

*Proof.* We begin by describing the maps. The entries of the top row are already identified as

$$
\begin{array}{cccc}
\underline{-2} & \underline{-1} & \underline{0} & \underline{1} \\
\mathrm{PV}_X^{0,\bullet}[z^{\pm 1}][\![t]\!] & & & \\
& \mathrm{PV}_X^{1,\bullet}[z^{\pm 1}][\![t]\!] \oplus \mathrm{PV}_X^{0,\bullet}[z^{\pm 1}][\![t]\!]z\partial_z & & \\
& & \mathrm{PV}_X^{2,\bullet}[z^{\pm 1}][\![t]\!] \oplus \mathrm{PV}_X^{1,\bullet}[z^{\pm 1}][\![t]\!]z\partial_z & \\
& & & \mathrm{PV}_X^{2,\bullet}[z^{\pm 1}][\![t]\!]z\partial_z
\end{array}.
$$

Let us turn to the map $\Phi'$ which is the map on cohomology induced from the natural map from IIB supergravity to IIB closed string field theory. To describe the source, we can compute cohomology using the spectral sequence associated to the filtration by powers of $t$. The spectral sequence degenerates at the first page for degree reasons. Using the Künneth formula and the computation of $H^\bullet(\mathrm{PV}^{i,\bullet}(\mathbb{C}^\times), \overline{\partial})$ as before, we find that the cohomology $H^\bullet(\widetilde{\mathcal{E}}_{\mathrm{mBCOV}}(\mathbb{C}^\times \times X), \overline{\partial}_{\mathbb{C}^\times})$ is given by

$$
\begin{array}{ccc}
\underline{-2} & \underline{-1} & \underline{0} \\
\mathrm{PV}_X^{0,\bullet}[z^{\pm 1}] & & \\
& \mathrm{PV}_X^{1,\bullet}[z^{\pm 1}] \oplus \mathrm{PV}_X^{0,\bullet}[z^{\pm 1}]z\partial_z & t\,\mathrm{PV}_X^{0,\bullet}[z^{\pm 1}] \\
& & \mathrm{PV}_X^{2,\bullet}[z^{\pm 1}]z\partial_z
\end{array},
$$

and hence $H^\bullet\left(\widetilde{\mathcal{E}}_{\mathrm{mBCOV}}\left(\mathbb{C}^\times \times X\right), \overline{\partial}_{\mathbb{C}^\times} + t\partial_{\mathbb{C}^\times}\right)$ is given by taking cohomology with respect to the differential

$$t\partial_{\mathbb{C}^\times}: \ \mathrm{PV}_X^{0,\bullet}[z^{\pm 1}]z\partial_z \longrightarrow t\,\mathrm{PV}_X^{0,\bullet}[z^{\pm 1}],$$

which yields the complex

$$
\begin{array}{ccc}
\underline{-2} & \underline{-1} & \underline{0} \\[4pt]
\mathrm{PV}_X^{0,\bullet}[z^{\pm 1}] & & \\[8pt]
& \mathrm{PV}_X^{1,\bullet}[z^{\pm 1}] \xrightarrow{\ t\partial_X\ } & \\[-2pt]
& \oplus & t\,\mathrm{PV}_X^{0,\bullet} \\[-2pt]
& \mathrm{PV}_X^{0,\bullet}z\partial_z & \\[8pt]
& & \mathrm{PV}_X^{2,\bullet}[z^{\pm 1}]z\partial_z
\end{array}
\ \ .
$$

Then the map

$$\Phi': \ H^\bullet\left(\widetilde{\mathcal{E}}_{\mathrm{mBCOV}}\left(\mathbb{C}^\times \times X\right), \overline{\partial}_{\mathbb{C}^\times} + t\partial_{\mathbb{C}^\times}\right) \to H^\bullet\left(\mathcal{E}_{\mathrm{BCOV}}\left(\mathbb{C}^\times \times X\right), \overline{\partial}_{\mathbb{C}^\times} + t\partial_{\mathbb{C}^\times}\right),$$

is given by the identity on the summands in degrees $-2$ and $-1$ in the decomposition above, given by zero on $t\,\mathrm{PV}_X^{0,\bullet}$, and is given by the divergence operator $\partial$ on the summand $\mathrm{PV}_X^{2,\bullet}[z,z^{-1}]z\partial_z$. However, not all components of the divergence operator act nontrivially in cohomology. Decomposing $\partial = \partial_{\mathbb{C}^\times}\otimes 1 + 1\otimes\partial_X$, we see that the second term only acts nontrivially on those elements in $\mathrm{PV}_X^{2,\bullet}z\partial_z$. Indeed, the image of any element outside of this subspace under the $\partial_X$ operator will be a vector field with nonzero modes along $\mathbb{C}^\times$, which are trivial in cohomology. Moreover, on $\mathrm{PV}_X^{2,\bullet}z\partial_z$, the term $\partial_{\mathbb{C}^\times}\otimes 1$ acts as zero. Thus we have described the map $\Phi'$ as

$$\Phi' = \begin{cases} \mathrm{Id}\,, & \text{on } \mathrm{PV}_X^{0,\bullet}[z^{\pm 1}] \oplus \mathrm{PV}_X^{1,\bullet}[z^{\pm 1}] \oplus \mathrm{PV}_X^{0,\bullet}z\partial_z\,, \\[4pt] \partial_X\,, & \text{on } \mathrm{PV}_X^{2,\bullet}z\partial_z\,, \\[4pt] 0\,, & \text{on } t\,\mathrm{PV}_X^{0,\bullet} \text{ and } \mathrm{PV}_X^{2,\bullet}z^{k+1}\partial_z,\ k\neq 0\,. \end{cases}$$

Finally, we define the dashed arrow $\mathbf{T}$ to be the map of cochain complexes given by

$$
\begin{array}{ccccccc}
\underline{-2} & \underline{-1} & \underline{0} & \underline{-2} & \underline{-1} & & \underline{0} \\[6pt]
\mathrm{PV}_X^{0,\bullet} & \cdots\cdots\cdots\cdots & \!\!\!\rightarrow \mathrm{PV}_X^{0,\bullet}[z^{\pm 1}] & & & & \\[10pt]
\mathrm{PV}_X^{2,\bullet} & \xrightarrow{\ \partial\ } & & \mathrm{PV}_X^{1,\bullet}[z^{\pm 1}] & \xrightarrow{\ t\partial_X\ } & & \\[-2pt]
\oplus & & & \oplus & & t\,\mathrm{PV}_X^{0,\bullet} & \\[-2pt]
\mathrm{PV}_X^{0,\bullet}\varepsilon & \xrightarrow{\ \varepsilon\mapsto z\partial_z\ } & & \mathrm{PV}_X^{0,\bullet}z\partial_z & & & \\[10pt]
\mathrm{PV}_X^{2,\bullet}\varepsilon & \xrightarrow{\qquad\ \varepsilon\mapsto z\partial_z\qquad} & & & & \mathrm{PV}_X^{2,\bullet}[z^{\pm 1}]z\partial_z &
\end{array}
\ \ ,
$$

where all the non-labeled maps are natural inclusion possibly using the identification of $\varepsilon$ and $z\partial_z$. It is easy to see that this is indeed a cochain map.

Next we show that the above map induces a map of DG Lie algebras on $\partial_X$-cohomology after shifting. We check this explicitly – there are several cases:

- Let $\alpha_1 \in \mathrm{PV}_X^{0,\bullet}$ and $\alpha_2 \in \mathrm{PV}_X^{0,\bullet}$. We have $\mathbf{T}[\alpha_1, \alpha_2] = \mathbf{T}(0) = 0$. On the other hand, $[\mathbf{T}(\alpha_1), \mathbf{T}(\alpha_2)]$ is the Schouten bracket between constants on $\mathbb{C}^\times$, which vanishes. Similarly, it holds for $\alpha_1 \in \mathrm{PV}_X^{0,\bullet}$ and $\alpha_2\varepsilon \in \mathrm{PV}_X^{0,\bullet}\varepsilon$; we have $\mathbf{T}[\alpha_1, \alpha_2\varepsilon] = \mathbf{T}(0) = 0$, while $[\mathbf{T}(\alpha_1), \mathbf{T}(\alpha_2\varepsilon)]$ is the Schouten bracket between constants on $\mathbb{C}^\times$ and $z\partial_z$, which vanishes.

- Let $\alpha \in \mathrm{PV}_X^{0,\bullet}$ and $\beta \in \mathrm{PV}_X^{2,\bullet}$. We have $\mathbf{T}[\alpha, \beta] = (-1)^{|\alpha|-1}\partial_X[\alpha, \beta]_{\mathrm{SN}}$. On the other hand, $[\mathbf{T}(\alpha), \mathbf{T}(\beta)] = [\alpha, \partial_X \beta] = [\alpha, \partial_X \beta]_{\mathrm{SN}}$. This equals $(-1)^{|\alpha|-1}\partial_X[\alpha, \beta]_{\mathrm{SN}}$ using the fact that the divergence operator $1 \otimes \partial_X$ is a shifted derivation of the Schouten bracket. Similarly, it holds for $\alpha \in \mathrm{PV}_X^{0,\bullet}$ and $\beta \varepsilon \in \mathrm{PV}_X^{2,\bullet}\varepsilon$; we have $\mathbf{T}[\alpha, \beta \varepsilon] = (-1)^{|\alpha|-1}\partial_X[\alpha, \beta]_{\mathrm{SN}}z\partial_z$, while $[\mathbf{T}(\alpha), \mathbf{T}(\beta \varepsilon)] = [\alpha, \beta z\partial_z] = (-1)^{|\alpha|-1}\partial_X[\alpha, \beta]_{\mathrm{SN}}z\partial_z$ as $\alpha$ and $\beta$ are both constants on $\mathbb{C}^\times$. A similar argument works for $\alpha \varepsilon \in \mathrm{PV}_X^{0,\bullet}\varepsilon$ and $\beta \in \mathrm{PV}_X^{2,\bullet}$ as well.

- Let $\beta_1 \in \mathrm{PV}_X^{2,\bullet}$ and $\beta_2 \in \mathrm{PV}_X^{2,\bullet}$. We have $\mathbf{T}[\beta_1, \beta_2] = \partial[\beta_1, \partial \beta_2]_{\mathrm{SN}}$. On the other hand, $[\mathbf{T}\beta_1, \mathbf{T}\beta_2] = [\partial \beta_1, \partial \beta_2]_{\mathrm{SN}}$. They coincide because $\partial$ is a (shifted) derivation. Similarly, it holds for $\beta_1 \in \mathrm{PV}_X^{2,\bullet}$ and $\beta_2\varepsilon \in \mathrm{PV}_X^{2,\bullet}\varepsilon$.

- Other cases either trivially vanish or follow from graded skew-symmetry of the Lie brackets.

$\square$

Finally, we can extend T-duality to the relevant twists of supergravity and closed string field theory by taking it to act as the identity on all other tensor factors. This yields the desired commuting diagram in Figure 2.

## 4.3 Twisted S-duality from eleven dimensions

In the previous two subsections, we have constructed maps as indicated in the following diagram:

$$
\begin{array}{ccc}
& H^\bullet(\mathrm{IIA}_{\mathrm{cl}}[(\mathbb{R}^5 \times S^1)_A \times X_B^2]) \xrightarrow{\;\simeq\;} H^\bullet(\mathrm{IIB}_{\mathrm{cl}}[\mathbb{R}_A^4 \times (\mathbb{C}^\times \times X^2)_B]) \\
\end{array}
$$

$$
H^\bullet(11\mathrm{d}[(\mathbb{R}^5 \times T^2)_A \times X_B]) \xrightarrow[\simeq]{\mathrm{red}_M} H^\bullet(\mathrm{IIA}_{\mathrm{SUGRA}}[(\mathbb{R}^5 \times S^1)_A \times X_B^2]) \xrightarrow{\mathbf{T}} H^\bullet(\mathrm{IIB}_{\mathrm{SUGRA}}[\mathbb{R}_A^4 \times (\mathbb{C}^\times \times X^2)_B]),
$$

where the cohomology is taken along the direction of $T^2$ for the 11d supergravity theory, $T^*S^1$ for IIA theory, and $\mathbb{C}^\times$ for IIB theory, respectively.

The composition $\mathbf{T} \circ \mathrm{red}_M$ of the two maps in the bottom row gives a map relating a twist of 11d supergravity on a torus to a twist of IIB supergravity on a "circle $S^1 \subset \mathbb{C}^\times$"; we denote this by $\Phi_{11\mathrm{d}\to\mathrm{IIB}} = \mathbf{T} \circ \mathrm{red}_M$. Note that there are natural $\mathrm{SL}(2, \mathbb{Z})$ actions on both the source and the $\partial_X$ cohomology of the target – the action on the source comes from the natural action on $H^\bullet(T^2)$ and the map on the target is the $\mathrm{SL}(2, \mathbb{Z})$ induced from the one defined in Subsection 2.3.2. Our main result of this section is the following:

**Theorem 4.9.** The composition $\Phi_{11\mathrm{d}\to\mathrm{IIB}} = \mathbf{T} \circ \mathrm{red}_M$ given by

$$
H^\bullet(11\mathrm{d}[(\mathbb{R}^5 \times T^2)_A \times X_B]; d_{T^2}) \xrightarrow{\mathrm{red}_M} H^\bullet(\mathrm{IIA}[(\mathbb{R}^5 \times S^1)_A \times X_B]; d_{S^1})
$$

$$
\xdownarrow{\Phi_{11\mathrm{d}\to\mathrm{IIB}}} \qquad \Big\downarrow \mathbf{T}
$$

$$
H^\bullet(\mathrm{IIB}[\mathbb{R}_A^4 \times (\mathbb{C}^\times \times X)_B]; \overline{\partial}_{\mathbb{C}^\times} + t\partial_{\mathbb{C}^\times}),
$$

induces an $\mathrm{SL}(2, \mathbb{Z})$ equivariant map of DG Lie algebras on $\partial_X$ cohomology.

*Proof.* Note that it is automatic that the map $\Phi_{11\mathrm{d}\to\mathrm{IIB}}$ induces a map of DG Lie algebras on $\partial_X$-cohomology. We need only check that equivariance. We begin by explicitly computing the map $\Phi_{11\mathrm{d}\to\mathrm{IIB}}$. Note that we have a quasi-isomorphism $H^\bullet(T^2) \cong \mathbb{C}[\varepsilon_M, \varepsilon]$, where $\varepsilon_M, \varepsilon$ are an ordered basis of $H^1(T^2)$ in which the generators $S, T$ of $\mathrm{SL}(2, \mathbb{Z})$ act according to Definition

2.19. Here $\varepsilon_M$ is from the circle of the M-theory, so Lemma 4.3 is non-trivially applied to $\varepsilon_M$, but not $\varepsilon$. By way of this quasi-isomorphism, we may write

$$H^\bullet\left(11\mathrm{d}[(\mathbb{R}^5 \times T^2)_A \times X]; d_{T^2}\right) \cong \Omega^\bullet(\mathbb{R}^5) \otimes \Omega^{0,\bullet}(X)[\varepsilon, \varepsilon_M].$$

The map $\Phi_{11\mathrm{d}\to\mathrm{IIB}}$ acts as the identity on $\Omega^\bullet(\mathbb{R}^5)$, so we deal with the other tensor factors. Fix $\alpha \in \Omega^{0,\bullet}(X)$.

From the definition of the maps $\mathrm{red}_M$ and $\mathbf{T}$, we see that:

$$\Phi_{11\mathrm{d}\to\mathrm{IIB}}(\alpha) = \mathbf{T} \circ \mathrm{red}_M(\alpha) = \mathbf{T}(\alpha \wedge \Pi_X) = \partial(\alpha \wedge \Pi_X),$$
$$\Phi_{11\mathrm{d}\to\mathrm{IIB}}(\alpha\varepsilon) = \mathbf{T} \circ \mathrm{red}_M(\alpha\varepsilon) = \mathbf{T}(\alpha\varepsilon \wedge \Pi_X) = \alpha \wedge \Pi_X \wedge z\partial_z,$$
$$\Phi_{11\mathrm{d}\to\mathrm{IIB}}(\alpha\varepsilon_M) = \mathbf{T} \circ \mathrm{red}_M(\alpha\varepsilon_M) = \mathbf{T}(\alpha) = \alpha,$$
$$\Phi_{11\mathrm{d}\to\mathrm{IIB}}(\alpha\varepsilon\varepsilon_M) = \mathbf{T} \circ \mathrm{red}_M(\alpha\varepsilon\varepsilon_M) = \mathbf{T}(\alpha\varepsilon) = \alpha \wedge z\partial_z.$$

Therefore, writing $\Omega^{-1}_{X\times\mathbb{C}^\times} = \Pi_X \wedge z\partial_z$, we compute that:

$$\mathbb{S}(\Phi_{11\mathrm{d}\to\mathrm{IIB}}(\alpha)) = \partial(\alpha \wedge \Pi_X), \qquad\qquad \mathbb{T}(\Phi_{11\mathrm{d}\to\mathrm{IIB}}(\alpha)) = \partial(\alpha \wedge \Pi_X),$$
$$\mathbb{S}(\Phi_{11\mathrm{d}\to\mathrm{IIB}}(\alpha\varepsilon)) = -\alpha, \qquad\qquad \mathbb{T}(\Phi_{11\mathrm{d}\to\mathrm{IIB}}(\alpha\varepsilon)) = \alpha + \alpha \wedge \Pi_X \wedge z\partial_z,$$
$$\mathbb{S}(\Phi_{11\mathrm{d}\to\mathrm{IIB}}(\alpha\varepsilon_M)) = \alpha \wedge \Pi_X \wedge z\partial_z, \qquad \mathbb{T}(\Phi_{11\mathrm{d}\to\mathrm{IIB}}(\alpha\varepsilon_M)) = \alpha,$$
$$\mathbb{S}(\Phi_{11\mathrm{d}\to\mathrm{IIB}}(\alpha\varepsilon\varepsilon_M)) = \alpha \wedge z\partial_z, \qquad\quad \mathbb{T}(\Phi_{11\mathrm{d}\to\mathrm{IIB}}(\alpha\varepsilon\varepsilon_M)) = \alpha \wedge z\partial_z.$$

On the other hand, we have that $S$ acts on $\mathbb{C}[\varepsilon_M, \varepsilon]$ by $\varepsilon_M \mapsto \varepsilon, \varepsilon \mapsto -\varepsilon_M$ and $T$ acts by $\varepsilon_M \mapsto \varepsilon_M$ and $\varepsilon \mapsto \varepsilon + \varepsilon_M$. Therefore, we see that:

$$\Phi_{11\mathrm{d}\to\mathrm{IIB}}(S(\alpha)) = \Phi_{11\mathrm{d}\to\mathrm{IIB}}(\alpha) = \partial(\alpha \wedge \Pi_X),$$
$$\Phi_{11\mathrm{d}\to\mathrm{IIB}}(S(\alpha\varepsilon)) = -\Phi_{11\mathrm{d}\to\mathrm{IIB}}(\alpha\varepsilon_M) = -\alpha,$$
$$\Phi_{11\mathrm{d}\to\mathrm{IIB}}(S(\alpha\varepsilon_M)) = \Phi_{11\mathrm{d}\to\mathrm{IIB}}(\alpha\varepsilon) = \alpha \wedge \Pi_X \wedge z\partial_z,$$
$$\Phi_{11\mathrm{d}\to\mathrm{IIB}}(S(\alpha\varepsilon\varepsilon_M)) = \Phi_{11\mathrm{d}\to\mathrm{IIB}}(\alpha\varepsilon\varepsilon_M) = \alpha \wedge z\partial_z,$$
$$\Phi_{11\mathrm{d}\to\mathrm{IIB}}(T(\alpha)) = \Phi_{11\mathrm{d}\to\mathrm{IIB}}(\alpha) = \partial(\alpha \wedge \Pi_X),$$
$$\Phi_{11\mathrm{d}\to\mathrm{IIB}}(T(\alpha\varepsilon)) = \Phi_{11\mathrm{d}\to\mathrm{IIB}}(\alpha\varepsilon + \alpha\varepsilon_M) = \alpha \wedge \Pi_X \wedge z\partial_z + \alpha,$$
$$\Phi_{11\mathrm{d}\to\mathrm{IIB}}(T(\alpha\varepsilon_M)) = \Phi_{11\mathrm{d}\to\mathrm{IIB}}(\alpha\varepsilon_M) = \alpha,$$
$$\Phi_{11\mathrm{d}\to\mathrm{IIB}}(T(\alpha\varepsilon\varepsilon_M)) = \Phi_{11\mathrm{d}\to\mathrm{IIB}}(\alpha\varepsilon\varepsilon_M) = \alpha \wedge z\partial_z.$$

This completes the proof. $\qquad\qquad\qquad\qquad\qquad\qquad\qquad\qquad\qquad\qquad\qquad\qquad\square$

**Remark 4.10.** In the above, we recovered the action of S-duality on $\mathrm{IIB}_{\mathrm{SUGRA}}[M^4_A \times (\mathbb{C}^\times \times X^2)_B]$ by establishing an equivalence with $11\mathrm{d}[(M^4 \times \mathbb{R} \times T^2)_A \times X^2_B]$ and considering the natural action of $\mathrm{SL}(2,\mathbb{Z})$ on the $T^2$ factor of the latter. It is natural to wonder whether one may obtain similar dualities from considering the $G_2 \times \mathrm{SU}(2)$ invariant twist of M-theory on other geometries that likewise include torus factors. In this remark, we will offer some speculation that $11\mathrm{d}[(S^1 \times X^3)_A \times \mathrm{TN}_B]$, where $X^3$ is a Calabi–Yau 3-fold and TN denotes Taub-NUT, deformed by a particular closed string field recovers the S-duality between topological strings $X^3$ developed in [17, 70].

The closed string field in question will exist in $11\mathrm{d}[(S^1 \times X^3)_A \times Q_B]$ where $Q$ is a holomorphic symplectic surface with a Hamiltonian $\mathbb{C}^\times$-action. Letting $H$ denote the Hamiltonian function for this action, we consider a closed string field of the form $d\theta \wedge H$ where $d\theta$ denotes a harmonic representative for the volume form on $S^1$. We claim that this closed string field deforms the background geometry into a twisted product $S^1 \tilde{\times} Q$ which is $Q$-bundle over $S^1$ with monodromy given by the Hamiltonian $\mathbb{C}^\times$-action on $Q$. Indeed, turning on this closed string

field has the effect of deforming the differential $\ell_1$ by $\ell_2(d\theta \wedge H, -)$. Explicitly, paying attention to just the components of the fields in $\Omega^\bullet(S^1) \otimes \Omega^{0,\bullet}(Q)$, the restriction of the deformed action reads

$$
\begin{aligned}
S(\alpha) &= \int \left( \frac{1}{2}\alpha\ell_1\alpha + \alpha\ell_2(d\theta \wedge H, \alpha) + \frac{1}{6}\alpha\ell_2(\alpha,\alpha) \right) \wedge \omega \\
&= \int \left( \frac{1}{2}(\alpha(d+\bar\partial)\alpha + \alpha d\theta \wedge \{H,\alpha\}) + \frac{1}{6}\alpha\{\alpha,\alpha\} \right) \wedge \omega \\
&= \int \left( \frac{1}{2}(\alpha(d + d\theta \wedge \{H,-\})\alpha + \alpha\bar\partial\alpha) + \frac{1}{6}\alpha\{\alpha,\alpha\} \right) \wedge \omega,
\end{aligned}
$$

where $\omega$ denotes the volume form on $Q$. Thus we see that this closed string field acts like a background gauge field which looks like a connection whose parallel transport will rotate $Q$ according to the vector field $\{H, -\}$ as one goes around the base $S^1$.

Now we specialize to the case $Q = \mathrm{TN}$. Identifying $\mathrm{TN} \cong \mathbb{C}^2$ as complex manifolds, we can pick holomorphic Darboux coordinates $z_1, z_2$. There is a Hamiltonian $\mathbb{C}^\times$ action with Hamiltonian given by $H(z_1, z_2) = z_1 z_2$. Then, the restriction of the deformed action to the components of the fields in $\Omega^\bullet(S^1) \otimes \Omega^{0,\bullet}(\mathrm{TN})$ reads

$$
S(\alpha) = \int \left( \frac{1}{2}(\alpha(d + d\theta \wedge (z_1\partial_{z_1} - z_2\partial_{z_2}))\alpha + \alpha\bar\partial\alpha) + \frac{1}{6}\alpha\{\alpha,\alpha\} \right) \wedge \omega,
$$

where $\omega$ now denotes the volume form on $\mathrm{TN}$.

We claim that changing the roles of the topological $S^1$ and the circle $S^1 \subset \mathrm{TN}$ is responsible the famous correspondence between the Gromov–Witten invariants and Donaldson–Thomas invariants [13] when we add an M2 brane on $S^1 \times Y$ to the picture where $Y \subset X^3$ is a holomorphic curves:

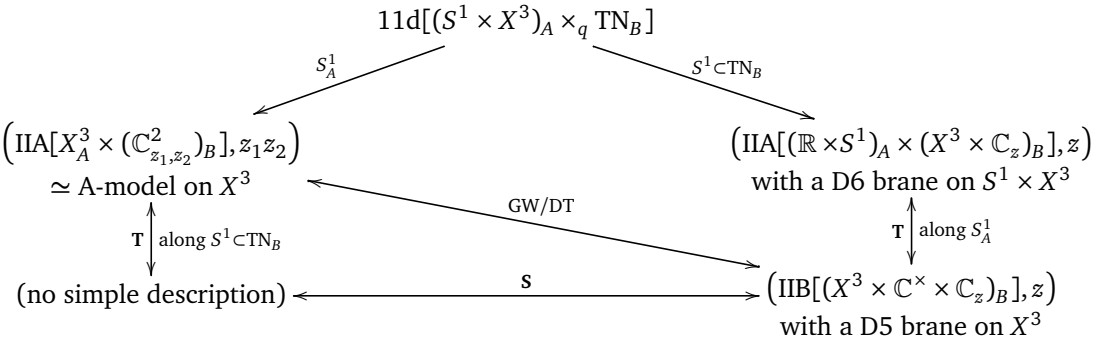

Let us discuss this in more detail. First, reducing $11\mathrm{d}[(S^1 \times X^3)_A \times \mathrm{TN}_B]$ on the topological $S^1$ yields $\mathrm{IIA}[X_A^3 \times \mathrm{TN}_B]$, under which the closed string field $d\theta \wedge z_1 z_2$ maps to $z_1 z_2$. This closed string field has the effect of localizing the degrees of freedom to the critical locus of $z_1 z_2$, which is the origin. This yields the A-model on $X^3$ – considering an M2 brane wrapping $S^1 \times Y$, where $Y \subset X^3$ is a holomorphic curve, yields an F1 wrapping $Y$ upon dimensional reduction. Counting the possible $Y \subset X^3$ that the F1 may wrap is supposed to yield the Gromov–Witten invariants of $X^3$. For completeness, let us say a few words about its T-dual image as well. Recall that TN admits a natural tri-Hamiltonian $U(1)$ action with fixed point the origin – the hyperkähler moment map $\mathrm{TN} \to \mathbb{R}^3$ exhibits TN as an $S^1$ fibration over $\mathbb{R}^3$ with the fiber shrinking to a point at $\{0\} \in \mathbb{R}^3$. T-dualizing along this $S^1$ yields an NS5 wrapping $X^3$ and leaves the F1 untouched.

Now let us reverse the roles of the two circles in M-theory. In view of Remark 4.6 we have that reducing the $G_2 \times \mathrm{SU}(2)$ invariant twist of M-theory on a holomorphic circle will yield

the SU(4) invariant twist of IIA deformed by a linear superpotential. It is expected that the $S^1$-reduction of $\text{TN}_B$ gives $\mathbb{R}_A \times \mathbb{C}_B$ with a D6 brane in a transversal direction. The discussion in Remark 4.6 additionally tells us that retaining higher momentum modes when reducing will yield additional D0 branes bound to the D6 brane. Therefore reducing along the $S^1$ fiber of Taub-NUT yields $\text{IIA}[(\mathbb{R} \times S^1)_A \times (X^3 \times \mathbb{C})_B]$, with a linear superpotential supported along $\mathbb{C}$ and a D6 brane wrapping $S^1 \times X^3$ with D0 branes bound to it. An additional M2 brane wrapping $S^1 \times Y$ will reduce to a D2 brane wrapping the same locus. The linear superpotential renders closed string field theory trivial, so we expect that the additional closed string field $d\theta \wedge z_1 z_2$ does not further deform the dimensionally reduced theory. Of course, without explicit formulae for the dimensional reduction in Remark 4.6, we are unable to check this claim. Granting this claim however, since the closed string field theory is trivial in perturbation theory, the partition function of the dimensional reduction must simply count states where D0 and D2 branes are bound to a single D6 brane wrapping $X^3 \times S^1$. It is known that such counts recover the DT invariants of $X^3$ (see, for instance [71]).

To make contact with the work of [17, 70] we perform an additional T-duality along the topological $S^1$. Doing so takes us to the $\text{IIB}[(X^3 \times \mathbb{C}^\times \times \mathbb{C})_B]$ with a linear superpotential along $\mathbb{C}_B$ and configuration of D$(-1)$ and D1 branes bound to a single D5 wrapping $X^3$. Once again the closed string field theory and open string field theory are rendered perturbatively trivial by the linear superpotential. One can readily see that the space of solutions to the equations of motion for the world-volume theory deformed by the linear superpotential is given by $T^*[-1]\left(\underline{\text{Map}}(X^3, B\,\text{GL}_1)_{\text{dR}}\right)$. The partition function of this theory is simply a generating function for counts of instantons of different charges. A configuration of $N$ D$(-1)$ branes and a D1 brane wrapping $Y$ gives us instantons in this theory with charges given by $\text{ch}_3 = N$, $\text{ch}_2 = [Y]$. Therefore, we indeed find the DT invariants of $X^3$ as they appear in [17]. Thus, this version of S-duality recovers the GW/DT correspondence.

# 5 Applications

In this section, we present some applications of our constructions. We focus on the case of flat spaces for technical reasons. We argue that several interesting deformations of 4-dimensional $\mathcal{N} = 4$ supersymmetric gauge theory are S-dual to each other. The deformations in question will be further deformations of the holomorphic-topological twist (also known as Kapustin twist after [72]). For $G = \text{GL}(N)$ this is precisely the theory living on a stack of $N$ D3 branes wrapping $\mathbb{R}^2 \times \mathbb{C}$ in topological string theory on $\mathbb{R}_A^4 \times \mathbb{C}_B^3$ (see Example 2.4). The deformations of interest are:

- The HT(A) and HT(B) deformations of 4d $\mathcal{N} = 4$ gauge theory. These are deformations of the Kapustin twist that are relevant for the geometric Langlands theory as analyzed in [14–16].

- Different types of superconformal deformations of 4d $\mathcal{N} = 4$ gauge theory as is realized in the supergravity setting via AdS/CFT correspondence. We also explain how these arise as homotopies trivializing the action of certain rotations on the background in the flavor of the $\Omega$-deformation.

- A quadratic superpotential transverse to the world-volume of the D3 branes. This deforms holomorphic-topological twist of 4-dimensional $\mathcal{N} = 4$ theory with gauge group $\text{GL}(N)$ into 4-dimensional Chern–Simons theory with gauge group $\text{GL}(N|N)$.

The desired claims will follow in two steps:

1. We first check that the holomorphic-topological twist is preserved under S-duality. To do so, we check that the stack of D3 branes is mapped to itself, which is as expected from physical string theory. This amounts to checking that the field sourced by the D3 branes in the sense of Definition 2.10 is preserved by S-duality. This is also an expected result from the work of Kapustin [72] but we argue it from a stringy perspective.

2. We then check that the claimed deformations are exchanged under S-duality. This amounts to checking that the preimages of these deformations under the closed-open map are mapped to each other under S-duality.

## 5.1 S-duality of a D3 brane

In this subsection, we argue that a stack of D3 branes wrapping $\mathbb{R}^2 \times \mathbb{C}$ in type IIB supergravity theory on $\mathbb{R}_A^4 \times \mathbb{C}_B^3$ is preserved under S-duality and conclude that this is related to a version of the geometric Langlands correspondence.

### 5.1.1 S-duality on a field sourced by D3 branes

As indicated above, the argument will proceed by analyzing the equation for the field sourced by the D3 branes.

**Proposition 5.1.** The field sourced by D3 branes wrapping $\mathbb{R}^2 \times \mathbb{C} \subset \mathbb{R}_A^4 \times \mathbb{C}_B^3$ is preserved under S-duality.

*Proof.* Let us first consider the minimal twist of type IIB supergravity theory on $\mathbb{C}^5$. For future reference, we say that $\mathbb{C}^5$ has coordinates $u, v, z, w_1, w_2$ and fix a holomorphic volume form $\Omega_{\mathbb{C}^5} = du \wedge dv \wedge dz \wedge dw_1 \wedge dw_2$. Suppose we have a stack of $N$ D3 branes supported at $v = w_1 = w_2 = 0$. By Definition 2.10, the field $F$ sourced by the $N$ D3 branes is a $(2,2)$-polyvector $F \in \ker \partial$ satisfying the equation

$$\bar{\partial} F \vee \Omega_{\mathbb{C}^5} = N \delta_{v=w_1=w_2=0} \, .$$

A solution of this equation is given by the Bochner–Martinelli kernel

$$F_{\text{sol}} = N \frac{\partial_u \wedge \partial_z}{(|v|^2 + |w_1|^2 + |w_2|^2)^3} (\bar{v} d\bar{w}_1 \wedge d\bar{w}_2 + \bar{w}_1 d\bar{w}_2 \wedge d\bar{v} + \bar{w}_2 d\bar{v} \wedge d\bar{w}_2) .$$

We are omitting an overall factor to get the same normalization as in Definition 2.10, as our argument holds regardless of the normalization. The reader may find formulas with the correct normalization in [39].

Now consider the further twist of IIB supergravity theory $\text{IIB}_{\text{SUGRA}}[\mathbb{R}_A^4 \times \mathbb{C}_B^3]$. We may think of it as gotten by making $\mathbb{C}_{u,v}^2$ noncommutative, i.e. by turning on the Poisson bivector $\partial_u \wedge \partial_v$. Then the field sourced by the D3 branes now should satisfy the deformed equation

$$(\bar{\partial} + [\partial_u \wedge \partial_v, -]_{\text{SN}}) F \vee \Omega_{\mathbb{C}^5} = N \delta_{v=w_1=w_2=0} \, ,$$

where $[-,-]_{\text{SN}}$ denotes the Schouten–Nijenhuis bracket on $\text{PV}^{\bullet,\bullet}(\mathbb{C}^2)$ up to a sign. We claim that $F_{\text{sol}}$ is in the kernel of $[\partial_u \wedge \partial_v, -]_{\text{SN}}$ so that it is still a solution to the deformed equation. To show this, note

$$[\partial_u \wedge \partial_v, -]_{\text{SN}} = \partial_u \wedge \partial_v(-) \pm \partial_u(-) \wedge \partial_v \, ,$$

and then $F_{\text{sol}}$ is evidently annihilated by both of these terms separately. This establishes the claim.

Under the isomorphism $\Omega^\bullet(\mathbb{R}^4) \cong \mathrm{PV}^{\bullet,\bullet}(\mathbb{C}^2_{u,v})$ induced by the standard symplectic form on $\mathbb{C}^2$, the above becomes

$$
\begin{aligned}
F_{\mathrm{sol}} = N &\frac{dv \wedge \partial_z}{(|v|^2 + |w_1|^2 + |w_2|^2)^3}(\bar{v}d\bar{w}_1 \wedge d\bar{w}_2 + \bar{w}_1 d\bar{w}_2 \wedge d\bar{v} + \bar{w}_2 d\bar{v} \wedge d\bar{w}_1) \\
&\in (\Omega^1(\mathbb{R}^4) \oplus \Omega^2(\mathbb{R}^4)) \otimes \mathbb{C}\langle \partial_z \rangle \otimes \mathrm{PV}^{0,\bullet}(\mathbb{C}^2_{w_1,w_2}).
\end{aligned}
$$

Now, by Definition 2.19, $\mathbb{S}$ acts as the identity on $F_{\mathrm{sol}}$. $\qquad\square$

### 5.1.2 Dolbeault geometric Langlands correspondence

We now explain the relation of the above with the so-called Dolbeaut geometric Langlands correspondence. A large part of what follows is a summary of the second author's joint work with C. Elliott [14]. For more details and contexts, one is advised to refer to the original article.

Let $C$ be a smooth projective curve and $G$ be a reductive group over $\mathbb{C}$ (and we write $\check{G}$ for its Langlands dual group). The best hope version of the geometric Langlands duality asserts an equivalence of DG categories

$$
\mathrm{D}(\mathrm{Bun}_G(C)) \simeq \mathrm{QCoh}(\mathrm{Flat}_{\check{G}}(C)),
$$

where $\mathrm{D}(\mathrm{Bun}_G(C))$ is the category of D-modules on the space $\mathrm{Bun}_G(C)$ of $G$-bundles on $C$ and $\mathrm{QCoh}(\mathrm{Flat}_{\check{G}}(C))$ is of quasi-coherent sheaves on the space $\mathrm{Flat}_{\check{G}}(C)$ of flat $\check{G}$-connections on $C$. This is the *de Rham geometric Langlands* correspondence.

Kapustin and Witten [9] studied certain $\mathbb{CP}^1$-family of topological twists of 4-dimensional $\mathcal{N} = 4$ supersymmetric gauge theory. To find the relation with the geometric Langlands correspondence, they realized that two twists should play particularly important roles. These are what are called the *A-twist* and *B-twist*, because they become A-model and B-model after certain compactification. Namely, if the 4-dimensional spacetime manifold $X$ is of the form $X = \Sigma \times C$, then compactification along $C$ leads to A-model on $\Sigma$ with target moduli space of Higgs bundles on $C$ and B-model on $\Sigma$ with target the moduli space of flat connections on $C$. Then studying the categories of boundary conditions of S-dual theories leads to a version of the geometric Langlands correspondence. However, this is most naturally seen as depending only on the topology of $C$ (which led to the exciting program of *Betti geometric Langlands* correspondence [73]), as opposed to the algebraic structure of $C$ which the original program is about.

In [14], a framework was introduced to capture the algebraic structure of the moduli spaces of solutions to the equations of motion. Moreover, it was suggested that when $X = \Sigma \times C$ one can study *holomorphic-topological twist* where the dependence is topological on $\Sigma$ and holomorphic on $C$ and the following theorem was proven. Here we write $\mathrm{EOM}(M) = \mathrm{EOM}^G(M)$ for the moduli space of solutions of a twisted gauge theory with group $G$ on a spacetime manifold $M$ and use subscripts to denote which twist we have used.

**Theorem 5.2.** [14]

$$
\mathrm{EOM}_{\mathrm{HT}}(\Sigma \times C) = T^*_{\mathrm{form}}[-1]\underline{\mathrm{Map}}(\Sigma_{\mathrm{dR}} \times C_{\mathrm{Dol}}, BG) = T^*_{\mathrm{form}}[-1]\underline{\mathrm{Map}}(\Sigma_{\mathrm{dR}}, \mathrm{Higgs}_G(C)),
$$

where $\mathrm{Higgs}_G(Y) := \underline{\mathrm{Map}}(Y_{\mathrm{Dol}}, BG)$ is the moduli space of $G$-Higgs bundles on $Y$, where $Y_{\mathrm{Dol}} := T_{\mathrm{form}}[1]Y$ is the Dolbeault stack of $Y$, $Y_{\mathrm{dR}}$ is the de Rham stack of $Y$, and $T^*_{\mathrm{form}}[-1](-)$ stands for the formal completion of the $(-1)$-shifted cotangent bundle along the zero section.

In this language, a B-model with target $X$ would be described by $T^*_{\mathrm{form}}[-1]\underline{\mathrm{Map}}(\Sigma_{\mathrm{dR}}, X)$, so the result can be summarized as stating that compactifying the holomorphic-topological twist along $C$ yields the B-model with target $\mathrm{Higgs}_G(C)$. Categorified geometric quantization

of B-model is studied in the sequel [15] where its relation with the moduli space of vacua is also investigated in detail. For the purpose of this paper, one can take the category of boundary conditions of the B-model with target $X$ to be $\mathrm{QCoh}(X)$ as is common in the context of homological mirror symmetry.

Now note that the holomorphic-topological twist is exactly what we see by putting D3 branes on $\mathbb{R}^2 \times \mathbb{C} \subset \mathbb{R}^4_A \times \mathbb{C}^3_B$. The globalization data needed to consider a theory on a non-flat space $\Sigma \times C$ comes from a twisting homomorphism. Then the fact that the field sourced by D3 branes on $\mathbb{R}^2 \times \mathbb{C}$ is preserved under S-duality suggests that the holomorphic-topological twist must be self-dual under S-duality for $G = \mathrm{GL}(N)$. In other words, our S-duality result expects a nontrivial conjectural equivalence $\mathrm{QCoh}(\mathrm{Higgs}_G(C)) \simeq \mathrm{QCoh}(\mathrm{Higgs}_G(C))$ for $G = \mathrm{GL}(N)$. This is compatible with the conjectural equivalence $\mathrm{QCoh}(\mathrm{Higgs}_G(C)) \simeq \mathrm{QCoh}(\mathrm{Higgs}_{\check{G}}(C))$ for a general reductive group $G$, which is what is known as the *classical limit of geometric Langlands* or *Dolbeault geometric Langlands* correspondence of Donagi and Pantev [74].

## 5.2 S-duality and De Rham geometric Langlands correspondence

We claim that our S-duality map in fact predicts the de Rham geometric Langlands correspondence in a way different from the suggestion of Kapustin and Witten [9].

To see this, recall that $N$ D3-branes on $\mathbb{R}^2 \times \mathbb{C} \subset \mathbb{R}^4_A \times \mathbb{C}^3_B$ yields the holomorphic-topological twist of $\mathrm{GL}(N)$ gauge theory described by

$$\mathcal{E}^{\mathrm{HT}}_{\mathrm{D3}} = \Omega^\bullet(\mathbb{R}^2) \otimes \Omega^{0,\bullet}(\mathbb{C})[\varepsilon_1, \varepsilon_2] \otimes \mathfrak{gl}(N)[1], \qquad \text{with differential} \qquad d_{\mathbb{R}^2} \otimes 1 + 1 \otimes \bar{\partial}_{\mathbb{C}}.$$

By definition, HT(A)-deformation is given by $\partial_{\varepsilon_1}$ and HT(B)-deformation is given by $\varepsilon_2 \partial_z$ where $z$ is a coordinate of $\mathbb{C}$. Again, the way we globalize to consider the spacetime of the form $\Sigma \times C$ amounts to fixing a twisting homomorphism. This globalization process is carefully discussed in the previous work of the second-named author with C. Elliott [14]:

**Theorem 5.3.** [14]

$$\mathrm{EOM}_{\mathrm{HT(A)}}(\Sigma \times C) = \underline{\mathrm{Map}}(\Sigma_{\mathrm{dR}} \times C_{\mathrm{Dol}}, BG)_{\mathrm{dR}} = T^*_{\mathrm{form}}[-1]\underline{\mathrm{Map}}(\Sigma_{\mathrm{dR}}, \mathrm{Bun}_G(C)_{\mathrm{dR}}),$$

$$\mathrm{EOM}_{\mathrm{HT(B)}}(\Sigma \times C) = T^*_{\mathrm{form}}[-1]\underline{\mathrm{Map}}(\Sigma_{\mathrm{dR}} \times C_{\mathrm{dR}}, BG) = T^*_{\mathrm{form}}[-1]\underline{\mathrm{Map}}(\Sigma_{\mathrm{dR}}, \mathrm{Flat}_G(C)).$$

**Note 5.4.** The comparison is summarized in the following table:

Table 8: Comparison of perturbative and non-perturbative descriptions for different deformations of 4d $N = 4$ gauge theory.

| theory | perturbative local | non-perturbative global |
|---|---|---|
| $\mathcal{E}^{\mathrm{HT}}_{\mathrm{D3}}$ | $\Omega^\bullet(\mathbb{R}^2) \otimes \Omega^{0,\bullet}(\mathbb{C})[\varepsilon_1, \varepsilon_2] \otimes \mathfrak{g}$ | $T^*_{\mathrm{form}}[-1]\underline{\mathrm{Map}}(\Sigma_{\mathrm{dR}}, \mathrm{Higgs}_G(C))$ |
| $\mathcal{E}^{\mathrm{HT(A)}}_{\mathrm{D3}}$ | $\Omega^\bullet(\mathbb{R}^2) \otimes \Omega^{0,\bullet}(\mathbb{C})[\varepsilon_1, \varepsilon_2] \otimes \mathfrak{g}$ with $\partial_{\varepsilon_1}$ | $T^*_{\mathrm{form}}[-1]\underline{\mathrm{Map}}(\Sigma_{\mathrm{dR}}, \mathrm{Bun}_G(C)_{\mathrm{dR}})$ |
| $\mathcal{E}^{\mathrm{HT(B)}}_{\mathrm{D3}}$ | $\Omega^\bullet(\mathbb{R}^2) \otimes \Omega^{0,\bullet}(\mathbb{C})[\varepsilon_1, \varepsilon_2] \otimes \mathfrak{g}$ with $\varepsilon_2 \partial_z$ | $T^*_{\mathrm{form}}[-1]\underline{\mathrm{Map}}(\Sigma_{\mathrm{dR}}, \mathrm{Flat}_G(C))$ |

Heuristically speaking, one can rewrite

$$T^*_{\mathrm{form}}[-1]\underline{\mathrm{Map}}(\Sigma_{\mathrm{dR}}, \mathrm{Higgs}_G(C)) \simeq \underline{\mathrm{Map}}(\Sigma_{\mathrm{dR}}, \underline{\mathrm{Map}}(C_{\mathrm{Dol}}, BG))_{\mathrm{Dol}},$$

where $\varepsilon_i$ is responsible for each Dolbeault stack. With our choice of convention, $\varepsilon_1$ is for the outer Dol and $\varepsilon_2$ is for the inner Dol. Then $\partial_{\varepsilon_1}$ is exactly what deforms the outer Dol to dR. Moreover, the twisting homomorphism makes $\varepsilon_2$ as $dz$ so $\varepsilon_2 \partial_z$ becomes the $\partial$-operator, which deforms $C_{\mathrm{Dol}}$ to $C_{\mathrm{dR}}$. This explains why the HT(A) and HT(B) deformations realize the desired global descriptions.

**Remark 5.5.** Here the notation is different from the one of [14]. We use the notation HT(A) and HT(B) to emphasize that those are realized as further deformations of the holomorphic-topological twist. By considering the category of boundary conditions or taking the category of quasi-coherent sheaves of the target of the B-model, we obtain $D(\text{Bun}_G(C)) := \text{QCoh}(\text{Bun}_G(C)_{\text{dR}})$ and $\text{QCoh}(\text{Flat}_G(C))$ respectively for $G = \text{GL}(N)$. Note that these are the main protagonists of the de Rham geometric Langlands correspondence and we did not need to invoke A-model or analytic dependence of the moduli space. Indeed, this was one of the main points of [14] on realizing the physical context of the de Rham geometric Langlands correspondence.

On the other hand, that there is a sense in which these two deformations are S-dual to each other is a new proposal that is different from the work of Kapustin and Witten [9]; if we thought of HT(A) and HT(B) deformations in terms of supercharges and considered the corresponding twists of the 4d $\mathcal{N} = 4$ gauge theory, then they would not have been S-dual to each other, because those supercharges do not even have the same rank. However, here we think of HT(A) and HT(B) as deformations of the HT twist, which in particular does not depend on the coupling constant of the gauge theory. Because twisted theories by supercharges possibly depend on the coupling constant, the theories deformed from the HT twist can be rather different from the ones directly realized from the 4d gauge theory by twists, although they have the same moduli spaces of solutions to the equations of motion at the classical level.

Now we argue that these two deformations of the HT twisted theory are indeed S-dual pairs, thereby recovering the de Rham geometric Langlands correspondence from our framework. Since globalization process is explained in the original paper [14], it remains to explain that those two deformations are S-dual to each other on a flat space, which is very easy:

**Proposition 5.6.** The HT(A) and HT(B) deformations are mapped to each other under S-duality.

*Proof.* Under the closed-open map, the preimages of the deformations $\partial_{\varepsilon_1}$ and $\varepsilon_2 \partial_z$ are superpotentials $w_1$ and the Poisson tensor $\partial_{w_2} \wedge \partial_z$, where $w_i$ denote holomorphic coordinates transverse to the world-volume of the D3 branes. Now, by Definition 2.19, we see that

$$\mathbb{S}(w_1) = \partial_{w_2} \wedge \partial_z.$$

The overall sign doesn't matter for the twists, so we are done. $\square$

**Remark 5.7.** A recent paper [16] by the second-named author with C. Elliott argues that the twisted S-duality in our sense also recovers the quantum geometric Langlands correspondence in a simpler way than the one of Kapustin and Witten [9].

## 5.3 S-duality on superconformal symmetries

In this subsection, we will study how S-duality acts on certain closed string fields that are quadratic polynomials. These will turn out to come from superconformal symmetries of the world-volume theory of a D3 brane, but realized as closed string fields in twisted IIB supergravity.

In [10, Section 9] Costello and Li identified certain deformations of the holomorphic twist of 4d $\mathcal{N} = 4$ theory that come from an action of the 4d $\mathcal{N} = 4$ superconformal algebra. This was done in the context of a proposal for a fully holomorphically twisted version of the AdS/CFT correspondence between the holomorphic twist of 4d $\mathcal{N} = 4$ theory and the SU(5)-invariant twist of type IIB supergravity. Concretely, the proposal posits a relationship between the world-volume theory of a large number of D3 branes wrapping $\mathbb{C}^2 \subset \mathbb{C}^5$, and

$\mathrm{IIB}_{\mathrm{SUGRA}}\big[\big(\mathbb{C}^5\setminus\mathbb{C}^2\big)_B\big]$ deformed by the flux sourced by the D3 branes. To formulate such a correspondence, a natural first step is to match the symmetries of either object. The holomorphic twist of the 4d $\mathcal{N}=4$ superconformal algebra naturally acts on the former theory, and Costello and Li construct the corresponding action on the latter theory.

Explicitly, this is established through the following results:

**Theorem 5.8** ( [10]).

1. The holomorphic twist of the 4d $\mathcal{N}=4$ superconformal algebra is equivalent to $\mathfrak{psl}(3|3)$ as a Lie superalgebra.

2. There is a map of Lie algebras

$$\mathfrak{psl}(3|3)\to\mathcal{E}_{\mathrm{mBCOV}}(\mathbb{C}^5)\,.$$

3. Consider $N$ D3 branes wrapping $\mathbb{C}^2\subset\mathbb{C}^5$ and let $F$ denote the flux they source. There exist $N$-dependent corrections to the above map to yield a map of Lie algebras

$$\mathfrak{psl}(3|3)\to\big(\mathrm{PV}^{\bullet,\bullet}(\mathbb{C}^5\setminus\mathbb{C}^2)[\![t]\!],\ \ell_1=\bar\partial+t\partial+N[F,-]_{\mathrm{SN}},\ \ell_2=[-,-]_{\mathrm{SN}}\big)\,.$$

Let us focus our attention on a collection of odd elements in the image of the map $\mathfrak{psl}(3|3)\to\mathcal{E}_{\mathrm{mBCOV}}(\mathbb{C}^5)$, which come from those superconformal symmetries that survive the holomorphic twist. Concretely, choosing coordinates $\mathbb{C}^2_{u,z}\subset\mathbb{C}^5_{u,v,z,w_1,w_2}$, the relevant closed string fields are given by elements

$$uv,\quad uw_1,\quad uw_2,\quad zv,\quad zw_1,\quad zw_2\quad\in\quad \mathrm{PV}^0_{\mathrm{hol}}(\mathbb{C}^5)\subset\mathrm{PV}^0_{\mathrm{hol}}(\mathbb{C}^5\setminus\mathbb{C}^2)\,,$$

and another type of three elements of $\mathrm{PV}^2_{\mathrm{hol}}(\mathbb{C}^5)\subset\mathrm{PV}^2_{\mathrm{hol}}(\mathbb{C}^5\setminus\mathbb{C}^2)$ given by

$$\partial_v(u\partial_u+z\partial_z-w_1\partial_{w_1}-w_2\partial_{w_2}),\quad \partial_{w_1}(u\partial_u+z\partial_z-v\partial_v-w_2\partial_{w_2}),\quad \partial_{w_2}(u\partial_u+z\partial_z-v\partial_v-w_1\partial_{w_1})\,.$$

We wish to analyze a subset of these deformations that survive to the further twist $\mathrm{IIB}_{\mathrm{SUGRA}}[\mathbb{R}^4_A\times\mathbb{C}^3_B]$. To properly do so, we should consider the cohomology of $\mathfrak{psl}(3|3)$ with respect to an SU(3)-invariant supercharge; we will present the details of this calculation elsewhere. The residual superconformal symmetries which are nontrivial in cohomology are $zw_1,\ zw_2\in\mathrm{PV}^0_{\mathrm{hol}}(\mathbb{C}^3)$ and $\partial_{w_1}(z\partial_z-w_2\partial_{w_2}),\ \partial_{w_2}(z\partial_z-w_1\partial_{w_1})\in\mathrm{PV}^2_{\mathrm{hol}}(\mathbb{C}^3)$.

The next proposition says that these residual symmetries are precisely S-dual to each other in the following way:

**Proposition 5.9.** Under the twisted S-duality map $\mathbb{S}$, we obtain the following correspondence

$$\begin{aligned}
zw_1&\quad\longleftrightarrow\quad\partial_{w_2}(z\partial_z-w_1\partial_{w_1}),\\
zw_2&\quad\longleftrightarrow\quad-\partial_{w_1}(z\partial_z-w_2\partial_{w_2}).
\end{aligned}$$

*Proof.* It follows from the following simple computations:

$$\begin{aligned}
\mathbb{S}(zw_1)&=w_1\partial_{w_1}\partial_{w_2}-z\partial_z\partial_{w_2}=\partial_{w_2}(z\partial_z-w_1\partial_{w_1}),\\
\mathbb{S}(zw_2)&=w_2\partial_{w_1}\partial_{w_2}+z\partial_z\partial_{w_1}=-\partial_{w_1}(z\partial_z-w_2\partial_{w_2}).
\end{aligned}$$

$\square$

**Remark 5.10.** In this remark, we provide another interpretation of these deformations.[4] The following claims are conjectural at the moment and discussions of the precise mathematical framework needed to articulate the nature of relevant objects is beyond the scope of the current paper.

In [75], the authors computed the holomorphic twist of the 4d $\mathcal{N} = 2$ superconformal algebra and identified certain odd elements that correspond to the superconformal deformation of [76] used to construct chiral algebras associated to 4d $\mathcal{N} = 2$ SCFT. Since the holomorphic twist of the 4d $\mathcal{N} = 2$ superconformal algebra acts on the holomorphic twist of a 4d $\mathcal{N} = 2$ SCFT, this furnishes a construction of these chiral algebras via the holomorphic twist. Let us sketch this construction in the case of the holomorphic twist of 4d $\mathcal{N} = 4$ theory. Recall that the holomorphic twist is described by $\Omega^{0,\bullet}(\mathbb{C}^2)[\varepsilon_1, \varepsilon_2, \varepsilon_3] \otimes \mathfrak{gl}_N$ in the perturbative BV formalism. Fixing coordinates $u, z$ on $\mathbb{C}^2$, desired superconformal deformation takes the form $u\partial_{\varepsilon_i}$ or $z\partial_{\varepsilon_i}$. For instance, the element $u\partial_{\varepsilon_3}$ turns $\Omega^{0,\bullet}(\mathbb{C}_u)[\varepsilon_3]$ into the Koszul complex for the origin in $\mathbb{C}_u$ so the resulting theory is quasi-isomorphic to $\Omega^{0,\bullet}(\mathbb{C})[\varepsilon_1, \varepsilon_2] \otimes \mathfrak{gl}_N$. Then from the result of [76], we know Lie algebra cochains of this should be the semiclassical limit of the chiral algebra associated to 4d $\mathcal{N} = 4$ SCFTs. Note that this set-up can be recovered from considering $N$ D3 branes wrapping $\mathbb{C}^2 \subset \mathbb{C}^5$ in IIB$[\mathbb{C}^5_B]$ and turning on a closed string field of the form identified above.

On the other hand, the same chiral algebra was recently understood [77–79] in terms of a certain $\Omega$-background in the holomorphic-topological twist of the 4d $\mathcal{N} = 2$ theory [72]. For comparison with the above, let us explain how this works in the example of the holomorphic-topological twist of 4d $\mathcal{N} = 4$ theory. The fields of the theory are given by $\Omega^\bullet(\mathbb{R}^2) \otimes \Omega^{0,\bullet}(\mathbb{C})[\varepsilon_1, \varepsilon_2] \otimes \mathfrak{gl}_N$. We subject the theory to a so-called B-type $\Omega$-background on $\mathbb{R}^2$ in the sense of [80]. This corresponds to working equivariantly with respect to a rotation about the origin in $\mathbb{R}^2$. This amounts to replacing $\Omega^\bullet(\mathbb{R}^2)$ with the Cartan model for $S^1$-equivariant cohomology of $\mathbb{R}^2$, that is, we end up with $\Omega^\bullet(\mathbb{R}^2)[t]^{S^1} \otimes \Omega^{0,\bullet}(\mathbb{C})[\varepsilon_1, \varepsilon_2] \otimes \mathfrak{gl}_N$ where $t$ is a parameter of degree 2 and $\ell_1 = d + t\iota_{\partial_\theta} + \bar{\partial}$. This is a family of complexes over $\mathbb{C}[t]$; the localization theorem tells us that the generic fiber is quasi-isomorphic to $\Omega^{0,\bullet}(\mathbb{C})[\varepsilon_1, \varepsilon_2] \otimes \mathfrak{gl}_N$ with differential $\ell_1 = \bar{\partial}$.

The above example illustrates the following phenomenon. A superconformal deformation of the holomorphic twist has the same effect as subjecting the theory to a further twist, and performing an $\Omega$-background construction along the newly topological directions. Moreover, the superconformal deformation plays the role as a homotopy trivializing the complexified infinitesimal action of the rotation we work equivariantly with respect to when performing an $\Omega$-deformation. It is natural to ask whether a similar phenomena occurs for those superconformal deformations that correspond to the ones we have identified above under a closed-open map.

Our starting point is the holomorphic-topological twist, not the holomorphic twist. Still we will see the similar relation between superconformal deformations and $\Omega$-background. One claim is that the HT(B)-twist $\varepsilon_2\partial_z$ from the previous example and the deformation $z\partial_{\varepsilon_2}$ combine to give an $\Omega$-background in the B-twist. More precisely, the complexification of the infinitesimal action of rotations in the $z$-plane acts on the fields of the HT(B)-twist via the Lie derivative $\mathcal{L}_{z\partial_z}$. This action can be made null-homotopic for the differential in the HT(B)-twist by way of the homotopy $z\partial_{\varepsilon_2}$ – this is precisely the superconformal deformation. One can show that working equivariantly with respect to the above rotation yields a family of complexes over $H^\bullet(B\,\mathbb{C}^\times)$ whose generic fiber is quasi-isomorphic to the deformation of $\mathcal{E}^{\mathrm{HT}}_{\mathrm{D3}}$ by $z\partial_{\varepsilon_2}$.

---

[4]We are grateful to Dylan Butson and Brian Williams for very enlightening discussions on the topic of this remark.

Somewhat surprisingly, the same pattern persists after S-duality. We claim that the HT(A)-twist $\partial_{\varepsilon_1}$ and the deformation $\varepsilon_1\varepsilon_2\partial_{\varepsilon_2} + \varepsilon_1 z\partial_z$ similarly combine to give an $\Omega$-background on the A-twist in the original sense of Nekrasov [81]. Once again, the action of complexified rotations on the space of fields is made homotopically trivial by way of a map that corresponds to a superconformal deformation under the closed open map. The situation is summarized in the following table:

Table 9: A-type and B-type twists in $\Omega$-background.

| | A-type | B-type |
|---|---|---|
| twist of $\mathcal{E}_{\text{D3}}^{\text{HT}}$ | $\Omega^\bullet(\mathbb{R}^2)[\varepsilon_2] \otimes \left( \Omega^{0,\bullet}(\mathbb{C})\varepsilon_1 \xrightarrow{\partial_{\varepsilon_1}} \Omega^{0,\bullet}(\mathbb{C}) \right)$ | $\Omega^\bullet(\mathbb{R}^2)[\varepsilon_1] \otimes \left( \Omega^{0,\bullet}(\mathbb{C}) \xrightarrow{\varepsilon_2\partial_z} \Omega^{0,\bullet}(\mathbb{C})\varepsilon_2 \right)$ |
| $\Omega$-background | $\mathcal{L}_{z\partial_z} = z\partial_z + \varepsilon_2\partial_{\varepsilon_2}$ | $\mathcal{L}_{z\partial_z} = z\partial_z + \varepsilon_2\partial_{\varepsilon_2}$ |
| deformation as | $\varepsilon_1\varepsilon_2\partial_{\varepsilon_2} + \varepsilon_1 z\partial_z$ on $\mathbb{C}[z,\varepsilon_1,\varepsilon_2]$ | $z\partial_{\varepsilon_2}$ on $\mathbb{C}[z,\varepsilon_2]$ |
| homotopy for $\Omega$ | $[\partial_{\varepsilon_1}, \varepsilon_1\varepsilon_2\partial_{\varepsilon_2} + \varepsilon_1 z\partial_z] = z\partial_z + \varepsilon_2\partial_{\varepsilon_2}$ | $[\varepsilon_2\partial_z, z\partial_{\varepsilon_2}] = z\partial_z + \varepsilon_2\partial_{\varepsilon_2}$ |

Let us comment on the nature of the claim we are making. The $\Omega$-background in a twist of a supersymmetric gauge theory is implemented by an equivariant structure on topological directions of spacetime. This yields a family of theories over the equivariant cohomology $H^\bullet(B\mathbb{C}^\times)$ whose generic fiber describes a theory that lives at the fixed points of the $\mathbb{C}^\times$-action. Similarly, $\Omega$-background constructions in twists of type II supergravity theories can be implemented by replacing the de Rham complex that appeared in the A-twisted directions of our supergravity theories by the Cartan model. Moreover, one might expect an implementation of this procedure at the level of topological string theory. Starting with the data of a Calabi–Yau category with a categorical $S^1$-action, one may hope to produce $\Omega$-deformed world-volume theories from ext computations and $\Omega$-deformed supergravity as a subset of the cyclic cochains. We will return to a discussion of the $\Omega$-background in supergravity in these terms elsewhere.

Remark 5.10 is an ingredient of a derivation from twisted string theory, of why the Yangian appears in the algebra of monopole operators in the A-twist of 3-dimensional $\mathcal{N} = 4$ theory as in [82, 83]. We first learned this perspective from Costello in his talk [84] at the 2017 String-Math conference, and works that expand this perspective further include [85, 86] Here the A-twist means the twist where the algebra of local operators parametrizes the Coulomb branch. A discussion on related topics is also given in the work of Costello and Yagi [87].

We recall Costello's derivation, adapted to our context. Consider $\text{IIB}[\mathbb{R}_A^4 \times \mathbb{C}_B^3]$ with the following brane configuration:

Table 10: D-brane configuration.

| | 0 | 1 | 2 | 3 | 4 | 5 | 6 | 7 | 8 | 9 |
|---|---|---|---|---|---|---|---|---|---|---|
| | $u$ | | $v$ | | $z$ | | $w_1$ | | $w_2$ | |
| $K$ D5 | | $\times$ | $\times$ | | $\times$ | $\times$ | $\times$ | $\times$ | | |
| $N$ D3 | $\times$ | $\times$ | | | $\times$ | $\times$ | | | | |

We also turn on a closed string field of the form $zw_2$.

Let us first consider the stack of D5 branes. The world-volume theory of the stack of D5 branes is a holomorphic-topological twist of 6-dimensional $\mathcal{N} = (1,1)$ supersymmetric gauge theory. Computing the self-Ext of this brane, we find that the space of fields is given by

$$\mathcal{E} = \Omega^\bullet(\mathbb{R}^2) \otimes \Omega^{0,\bullet}(\mathbb{C}^2_{z,w_1})[\varepsilon_2] \otimes \mathfrak{gl}(K)[1].$$

As the image of the closed string field under the closed-open map is the deformation $z\partial_{\varepsilon_2}$, turning this on leads to

$$\Omega^\bullet(\mathbb{R}^2)\otimes\Omega^{0,\bullet}(\mathbb{C}_{w_1})\otimes\left(\Omega^{0,\bullet}(\mathbb{C}_z)\varepsilon_2\xrightarrow{z\partial_{\varepsilon_2}}\Omega^{0,\bullet}(\mathbb{C}_z)\right)\otimes\mathfrak{gl}(K)[1]\cong\Omega^\bullet(\mathbb{R}^2)\otimes\Omega^{0,\bullet}(\mathbb{C}_{w_1})\otimes\mathfrak{gl}(K)[1].$$

In sum, the D-brane gauge theory on the D5 branes becomes 4d Chern–Simons theory on $\mathbb{R}^2\times\mathbb{C}_{w_1}$ with gauge group $\mathrm{GL}(K)$. In fact, what we have described is exactly the construction of [87] in our chosen protected sector of type IIB theory.

Now let us consider what happens to the D3 branes. Of course, the D-brane gauge theory of the D3 branes is precisely $\mathcal{E}_{\mathrm{D3}}^{\mathrm{HT}}=\Omega^\bullet(\mathbb{R}^2)\otimes\Omega^{0,\bullet}(\mathbb{C}_z)[\varepsilon_1,\varepsilon_2]\otimes\mathfrak{gl}(N)[1]$. Now turning on the deformation $z\partial_{\varepsilon_2}$ yields the complex

$$\Omega^\bullet(\mathbb{R}^2)\otimes\left(\Omega^{0,\bullet}(\mathbb{C}_z)\varepsilon_2\xrightarrow{z\partial_{\varepsilon_2}}\Omega^{0,\bullet}(\mathbb{C}_z)\right)\otimes\mathbb{C}[\varepsilon_1]\otimes\mathfrak{gl}(N)[1]\cong\Omega^\bullet(\mathbb{R}^2)[\varepsilon_1]\otimes\mathfrak{gl}(N)[1].$$

Thus, we find 2-dimensional BF theory with gauge group $\mathrm{GL}(N)$.

A similar calculation shows that the bi-fundamental strings stretched between the two stacks of branes yields free fermions as a 1-dimensional defect living on the line, corresponding to the direction 1 in the table. Thus, we find exactly the topological string set-up of [88], where it is shown that the operators of the coupled system that live on the line generate a quotient of the Yangian of $\mathfrak{gl}(K)$.

We wish to analyze the effect of S-duality on the above setup. Physically, it is known that D5 branes become NS5 branes after S-duality. Therefore, acting on the set-up by S-duality yields a Hanany–Witten brane cartoon whose low-energy dynamics is described by a 3-dimensional $\mathcal{N}=4$ linear quiver gauge theory.

Checking the claim of [84] then amounts to checking that the local operators of this 3d $\mathcal{N}=4$ theory, deformed by the closed string field $\varepsilon_1\varepsilon_2\partial_{\varepsilon_2}+\varepsilon_1 z\partial_z$, give the same quotient of the Yangian of $\mathfrak{gl}(K)$. We intend to return to this question, and a broader analysis of 3-dimensional $\mathcal{N}=4$ quiver gauge theories in our context, elsewhere.

## 5.4 S-duality on 4d Chern–Simons theory

As another quadratic polynomial, we consider

$$\mathbb{S}(w_1 w_2)=w_1\partial_z\partial_{w_1}-w_2\partial_z\partial_{w_2},$$

so that we know $\partial_{\varepsilon_1}\partial_{\varepsilon_2}$ is S-dual to $\pi=\varepsilon_1\partial_{\varepsilon_1}\partial_z-\varepsilon_2\partial_{\varepsilon_2}\partial_z$.

First, to understand the consequence of adding the deformation $\partial_{\varepsilon_1}\partial_{\varepsilon_2}$ to the holomorphic-topological twist $\Omega^\bullet(\mathbb{R}^2)\otimes\Omega^{0,\bullet}(\mathbb{C})[\varepsilon_1,\varepsilon_2]\otimes\mathfrak{gl}(N)[1]$, we note that $(\mathbb{C}[\varepsilon_1,\varepsilon_2],\partial_{\varepsilon_1}\partial_{\varepsilon_2})$ is the Clifford algebra $\mathrm{Cl}(\mathbb{C}^2)\cong\mathrm{End}(\mathbb{C}^{1|1})$, and hence the deformed theory is

$$\mathcal{E}=\Omega^\bullet(\mathbb{R}^2)\otimes\Omega^{0,\bullet}(\mathbb{C})\otimes\mathfrak{gl}(N|N)[1],$$

also known as the 4d Chern–Simons theory with gauge group given by the supergroup $\mathrm{GL}(N|N)$.

On the other hand, $\pi$ gives a peculiar noncommutative deformation of the twisted theory. The original category $\mathrm{Perf}(\mathrm{Higgs}_G(C))$ of boundary conditions of the 4-dimensional theory with $G=\mathrm{GL}(N)$ reduced along $C$ is now deformed to $\mathrm{Perf}(\mathrm{Higgs}_G(C),\pi)$.

**Remark 5.11.** Costello suggested that this deformation can be explicitly constructed in terms of difference modules. In particular, he suggested that when $C=E$ is an elliptic curve, then the deformed category is the category of coherent sheaves on the moduli space $\mathrm{Higgs}_\lambda(E)$ of rank $N$ vector bundles $F$ on $E$ together with a homomorphism $F\to T_\lambda^*F$ where $T_\lambda\colon E\to E$ is the translation by $\lambda$. Indeed, when $\lambda=0$, this recovers the moduli space $\mathrm{Higgs}_G(E)$ of $G$-Higgs bundles on $E$, explaining the notation.

A general principle tells us that the category of line defects of a theory acts on the category of boundary conditions. In this case, we understand the category of line defects of 4-dimensional Chern–Simons theory on a Calabi–Yau curve $C$, namely, $\mathbb{C}$, $\mathbb{C}^{\times}$, or $E$, [89–91] as monoidal category of representations of the Yangian, the quantum loop group, the elliptic quantum group for $GL(N|N)$, respectively.

Hence from duality one can conjecture that the category of line defects of 4d Chern–Simons theory for $GL(N|N)$ acts on the category of boundary conditions $\mathrm{Perf}(\mathrm{Higgs}_G(C), \pi)$. It would be interesting to make this conjecture more precise along the line of suggestion of Costello and investigate it further.

# Acknowledgments

We thank Kevin Costello, Davide Gaiotto, and Si Li for their guidance as we learned aspects of string theory. As mathematicians, entering this area posed many challenges, and their support was invaluable. This project would not have taken off without Costello's lectures on a related topic and the extensive discussions that followed. We are also grateful to Dylan Butson, Jacques Distler, Nick Rozenblyum, Brian Williams, and Edward Witten for critical questions and insightful discussions. The paper went through several rounds of revision, during which we benefited from the support of various institutions. S.R. would like to acknowledge both the Perimeter Institute for Theoretical Physics and the University of Edinburgh for support while various drafts of the current paper were being prepared.

**Funding information** This work was first announced at Aspen Center for Physics, which is supported by National Science Foundation grant PHY-1607611. This work was supported by the National Research Foundation of Korea (NRF) grant funded by the Korea government(MSIT) (No. 2022R1F1A107114213) and by the LAMP Program of the National Research Foundation of Korea (NRF) grant funded by the Ministry of Education (No. RS-2023-00301976).

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
