# Peer review of "Twisted S-Duality"

_SciPost Physics, doi:SciPost Phys. 19, 049 (2025)_

## Round 1 · Referee Report · Anonymous (Referee 1) · 2025-6-15

Report

This paper studies S-duality in type IIB supergravity from the point of view of twisted supergravity (BCOV theory), working within the approach via topological string theory initiated in the work of Costello and Li. It is a major revision of a preprint that first appeared in late 2019.

The paper makes substantial contributions: Its main results are a rigorous construction of the S-duality map as an automorphism of a particular "theory of potentials" for minimal BCOV theory on threefolds (Theorem 2.20) and a proof at the level of formal moduli problems (Theorem 4.9) that the geometric action of $SL(2,\mathbb{Z})$ on the torus factor in the spacetime geometry $\mathbb{R}^5 \times T^2 \times X$ induces an action on eleven-dimensional supergravity that intertwines with S-duality for BCOV theory (again with potentials) on $\mathbb{R}^4 \times \mathbb{C}^\times \times X$, under the map given by circle compactification followed by T-duality. Here $X$ is a Calabi-Yau surface. Both results are expected and well-established in the physics literature, but are upgraded to the level of rigorous theorems.

Beyond this, the techniques of the paper contribute as much or more than the results. The use of such "theories of potentials" is of essential importance in connecting results from the topological-string approach to the supergravity literature; the authors offer a careful and general discussion, which (as far as this reviewer is aware) does not appear anywhere else in the literature. Section 3 offers a clear, comprehensive, and in places novel overview of general ideas in twisted supergravity; while it is schematic at times, and refers more to expected general features of supergravity than to any specific theory, this overview (especially the table on Page 29) will be of great use to a broader audience hoping to make inroads into the twisted supergravity literature.

The paper is carefully written and is well-structured. I am happy to recommend it for publication in SciPost, and do not think it requires any changes from the present version prior to its publication. That being said, I will mention a couple of small typos and broader suggestions here, which the authors can choose to incorporate or not, as they see fit.

Minor typos/remarks: 1- At the bottom of page 3: the de Rham complex. 2- On page 4: a certain category 3- On page 15: ask for the gauge-fixing condition. Also, in the first sentence of 2.3.1: Recall that BCOV theory. 5- Definition 2.12: Remove the indefinite article: "Minimal BCOV theory." 6- Though it is standard, perhaps a citation for the Palatini action is appropriate here?

Other comments: 1- On page 16, potentials are initially introduced only as a replacement of a divergence-free $(d-1)$-polyvector field by a $d$-polyvector field. But the intention is to use them in more generality, as Remark 2.18 makes clear. I feel this could be structured a little more clearly, and a quick remark on the isomorphism with partial de Rham complexes, or on the physical intuition (connections versus curvatures), would go a long way towards helping readers who have not seen such a maneuver before. 2- In the discussion starting on page 23: Perhaps one is really talking about a Cartan connection here? And would it be helpful to think of a map in conceiving of the "partial theory" we are talking about: something like "Whatever supergravity is, it should come with a map to $T^*[-1]C$, where $C$ is the formal moduli problem of $\mathfrak{siso}$ connections (of whatever appropriate type)"?

Recommendation

Publish (surpasses expectations and criteria for this Journal; among top 10%)

---

## Round 1 · Referee Report · Anonymous (Referee 2) · 2025-7-13

Report

This paper makes significant contributions to: 1) discovery of an $SL(2,\mathbb{Z})$ symmetry in the minimal BCOV theory with potentials (Theorem 2.20), in particular, a mathematical formulation of S-duality in the twisted type IIB supergravity; 2) providing a rigorous proof of the expectation from physics literatures that S-duality arises from the mapping class group of the M-theory torus (Theorem 4.9). This paper also contains interesting applications of the main results: various S-duality statements in deformations of 4d N=4 gauge theories are presented, for example, a physical perspective on the de Rham geometric Langlands correspondence.

The presentation of this paper is clear and comprehensive. I recomment it to publish the current version on SciPost.

In addition to the Referee 1's comments, I will make two more here. 1) In the middle of page 10, perhaps it is better to say "by HKR theorem" instead of "by way of the Hodge decomposition", 2) typo in the middle of page 35, should be "Künneth formula".

Recommendation

Publish (surpasses expectations and criteria for this Journal; among top 10%)

---

## Editorial Decision

published